# Learning Unanimously Acceptable Lotteries via Queries

Davin Choo [1]   Paul W. Goldberg [2]   Nicholas Teh [2] [*]

## Abstract

Many high-stakes AI deployments proceed only if every stakeholder deems the system acceptable relative to their own minimum standard. With randomization over a finite menu of options, this becomes a feasibility question: does there exist a lottery over options that clears all stakeholders' acceptability bars? We study a query model where the algorithm proposes lotteries and receives only binary accept/reject feedback. We give deterministic and randomized algorithms that either find a unanimously acceptable lottery or certify infeasibility; adaptivity can avoid eliciting many stakeholders' constraints, and randomization further reduces the expected elicitation cost relative to full elicitation. We complement these upper bounds with worst-case lower bounds (in particular, linear dependence on the number of stakeholders and logarithmic dependence on precision are unavoidable). Finally, we develop learning-augmented algorithms that exploit natural forms of advice (e.g., likely binding stakeholders or a promising lottery), improving query complexity when predictions are accurate while preserving worst-case guarantees.

## 1. Introduction

In high-stakes AI deployments, before release, systems are checked against a collection of requirements: internal safety review, policy and compliance constraints, domain expert criteria, and often structured human evaluation (Gebru et al., 2021; Government of Canada, 2020; Mitchell et al., 2019; Raji et al., 2020; US Department of Commerce, 2023; UK Department for Science, Innovation and Technology, 2024). In many settings, these requirements act as *gates* rather than soft preferences: failure on a single critical check is enough

to block deployment (Anthropic, 2025; European Parliament and Council of the European Union, 2024; OpenAI, 2025). This kind of decision process is conservative by design: it is meant to ensure that no required standard is violated, even when different teams (or affected stakeholders) focus on different risks (Government of Canada, 2020; US Department of Commerce, 2023).

A second, often overlooked aspect is that a deployment choice determines a *distribution* of outcomes rather than a single outcome (Sutton & Barto, 2018). Even a fixed model run under a fixed policy produces variable behavior due to environmental uncertainty and stochastic execution (Ovadia et al., 2019; Quiñonero-Candela et al., 2008). Crucially, considering distributions (lotteries) rather than deterministic choices convexifies the feasible space. In many governance settings, no single monolithic model configuration may satisfy every stakeholder's constraints simultaneously. However, a randomized mixture (e.g., routing 90% of traffic to a high-performance but high-risk model and 10% to a highly robust fallback) may satisfy the aggregate risk thresholds of all parties. Modern deployments frequently include such explicit mixing mechanisms (e.g., routing, fallback modes) (Amazon Web Services, 2023). From a governance perspective, the problem therefore becomes selecting a *distribution* over options that clears all safety bars.

In such settings, it is natural that different stakeholders impose different minimum standards. This can often be understood in practice as *operational risk thresholds* reflecting different risk appetites (US Department of Commerce, 2023). A compliance group may insist on a hard constraint; a safety team may require a baseline level of robustness; an impacted population may care about guarantees that do not align with average-case performance. We formalize this by letting each stakeholder evaluate candidate deployments through their own scalar yardstick (which one can read as a utility, a safety score, or an inverted risk score), together with a personal acceptability threshold that encodes their minimum bar. Crucially, these standards are typically assessed through coarse feedback (Government of Canada, 2020; 2025). Approval processes do not ask each stakeholder to provide a complete utility function over all possible outcomes. They ask whether a proposed deployment is *acceptable*. It is also common that acceptability is judged relative to a reference point: the current system

---

[*]Primary author; authors ordered alphabetically [1]Harvard University, Cambridge, Massachusetts, United States [2]University of Oxford, United Kingdom. Correspondence to: Nicholas Teh <nicholas.teh@cs.ox.ac.uk>.

*Proceedings of the 43rd International Conference on Machine Learning*, Seoul, South Korea. PMLR 306, 2026. Copyright 2026 by the author(s).

in production, a previously approved policy, or a conservative baseline. This view provides a simple interpretation of a stakeholder's "minimum bar": approval means the proposal meets or exceeds what would otherwise be deployed, according to that stakeholder's criterion.

Our work studies the resulting feasibility problem: given a menu of candidate deployments and stakeholders who respond only with accept/reject judgments, can we find a deployment choice that everyone approves (i.e., a *unanimously acceptable* choice), or certify that no such choice exists? A certificate of infeasibility is meaningful here: it says the current menu is inadequate to satisfy all thresholds simultaneously, so the correct response is to modify the design space (e.g., add mitigations or introduce safer operating modes) rather than to search longer for a nonexistent compromise (Anthropic, 2025; European Parliament and Council of the European Union, 2024; OpenAI, 2025; US Department of Commerce, 2023).

Our focus is the *information cost* of reaching such a conservative decision. We work in a query-based model in which the algorithm proposes candidate randomized deployment choices and observes only yes/no acceptability feedback from stakeholders. The goal is to minimize the number of such queries needed to either produce a universally approved choice or correctly conclude infeasibility.

## 1.1. Our Results

We study the following query feasibility problem. There are $n$ agents and $m$ alternatives. The algorithm proposes lotteries $\mathbf{x} \in \Delta(S)$ and may query any agent $i$ with a query $\text{Query}(i, \mathbf{x}) \in \{\texttt{True}, \texttt{False}\}$ indicating whether $\langle \mathbf{u}_i, \mathbf{x} \rangle \geq \tau_i$, where $(\mathbf{u}_i, \tau_i)$ are unknown to the algorithm. The goal is to either (i) output a unanimously acceptable lottery $\mathbf{x} \in \bigcap_{i=1}^{n} \mathcal{A}_i$, or (ii) correctly certify infeasibility.

**Single agent elicitation.** We give an exact procedure LearnHyperplane($i$) that recovers an agent's acceptability halfspace on the simplex using only accept/reject queries using $\mathcal{O}(m \log(1/\varepsilon))$ membership queries and returns either ACCEPTALL, REJECTALL, or a linear inequality equivalent to $\langle \mathbf{u}_i, \mathbf{x} \rangle \geq \tau_i$ for all $\mathbf{x} \in \Delta(S)$ (Lemmas 3.1 and 3.2).

**Deterministic multi-agent algorithm.** Building on LearnHyperplane, we give an adaptive deterministic algorithm that learns an agent's constraint only when it is violated by the current candidate lottery. It always outputs a unanimously acceptable lottery when one exists, and otherwise outputs NULL, using $\mathcal{O}\left(n^2 + nm \log(1/\varepsilon)\right)$ queries in the worst case (Theorem 3.3).

**Randomization reduces hyperplane learning.** We adapt a Clarkson-style sampling/reweighting approach for low-dimensional linear programming to our query model. The resulting randomized algorithm is always correct and uses

$\mathbb{E}[\mathcal{O}(nm \log n + \min\{n, m^3 \log n\} \cdot m \log(1/\varepsilon))]$ queries (Theorem 3.4). In particular, when $n \gg m^3 \log n$, it learns only a vanishing fraction of agents' constraints in expectation, while preserving correctness.

**Lower bounds.** We show that any (deterministic or randomized) algorithm that is always correct must make $\Omega((n - \min\{n, m\}) + (\min\{n, m\} - 1) \log(1/\varepsilon))$ queries in the worst case, even for binary utilities (Theorem 4.1). Additionally, even for a single agent ($n = 1$), $\Omega(m)$ queries are necessary in the worst case (Theorem 4.2). These bounds explain the unavoidable linear dependence on $n$ and logarithmic dependence on $1/\varepsilon$.

**Learning-augmented elicitation.** Finally, we give learning-augmented variants that exploit natural forms of advice (e.g., likely binding stakeholders or a promising lottery), improving query complexity when predictions are accurate while preserving worst-case guarantees. With a predicted permutation $\hat{\sigma}$ over agents, the deterministic algorithm uses $\mathcal{O}((n + m \log(1/\varepsilon)) R(\hat{\sigma}))$ queries, where $R(\hat{\sigma})$ is the number of *record* agents encountered under that order (Theorem 5.1). For the randomized algorithm, we bias the initial sampling weights toward earlier agents in $\hat{\sigma}$ and obtain an expected query complexity of $\mathcal{O}(nm\mu + \min\{n, m^3\mu\} \cdot m \log(1/\varepsilon))$, where $\mu = \log E(\hat{\sigma}) + \log \log n$ and $E(\hat{\sigma})$ is the smallest prefix length that contains a valid witness set (Theorem 5.2). Additionally, given a predicted lottery $\hat{\mathbf{x}}$, we warm-start the single agent turning point searches inside LearnHyperplane, reducing the cost of learning an agent's halfspace to $\mathcal{O}(m + m \log(1 + \delta_{\max,i}(\hat{\mathbf{x}})/\varepsilon^2))$ queries (Theorem 5.3), while preserving worst-case bounds when the prediction is poor. We provide accompanying lower bounds on the minimum number of queries required even in the presence of each of these predictions.

## 1.2. Related Work

**Preference elicitation and query learning.** Our setup is closely related to preference elicitation, which studies how to ask as few questions as possible to make a decision without fully learning preferences (e.g., in auctions or voting). A classic connection between elicitation and computational learning is that elicitation can be viewed as querying multiple "concepts" with the goal of producing an *optimal example* rather than reconstructing each concept (Blum et al., 2004). Subsequent work develops a variety of query types and policies for eliciting complex preferences and utilities (e.g., Boutilier, 2002; Conen & Sandholm, 2001). Our model aligns in this spirit: our goal is not to recover $(\mathbf{u}_i, \tau_i)$, but to find any lottery satisfying all agents' constraints. The technical distinction is that we study a structured continuous decision space (the simplex over $m$ alternatives) with *binary accept/reject queries* about lotteries, and we provide worst-case query bounds with explicit dependence on the

required numerical precision $\varepsilon$.

**Query learning of halfspaces and geometric concepts.** At a geometric level, each agent induces a halfspace over the $(m-1)$-simplex via an acceptability constraint $\langle \mathbf{u}_i, \mathbf{x} \rangle \geq \tau_i$. Our single-agent subroutine that recovers an agent's acceptability boundary is related to classical query learning of geometric concepts, including exact learning on discretized domains and membership-query learning of polytopes where the *precision* of query points is treated as a resource (Angluin, 1988; Bshouty et al., 1998a;b; Goldberg & Kwek, 2000). The key difference is the learning objective and the multi-agent structure: most prior work targets *identifying* an unknown concept for a single oracle, whereas we have $n$ distinct (unknown) halfspaces and only need to find *one point* in their intersection (or prove emptiness). This distinction is what enables algorithms that learn only a small subset of "constraining" agents' hyperplanes in the worst case, rather than learning all $n$ constraints, at least in expectation.

**Learning-augmented algorithms.** Since the seminal work of Lykouris & Vassilvitskii (2021), there has been a surge of interest in incorporating unreliable advice into algorithm design and analyzing performance as a function of advice quality across various areas of computer science. This framework has been especially successful in online optimization, where the core challenge lies in handling uncertainty about future inputs. In this context, advice can serve as a useful proxy for the unknown future. Most previous works in this setting are in the context of online algorithms, e.g. for the ski-rental problem (Gollapudi & Panigrahi, 2019; Wang et al., 2020; Angelopoulos et al., 2024), non-clairvoyant scheduling (Purohit et al., 2018), scheduling (Lattanzi et al., 2020; Bamas et al., 2020a; Antoniadis et al., 2022), augmenting classical data structures with predictions (e.g. indexing (Kraska et al., 2018) and Bloom filters (Mitzenmacher, 2018)), online selection and matching problems (Antoniadis et al., 2023; Dütting et al., 2021; Choo et al., 2024; 2025), online TSP (Bernardini et al., 2022; Gouleakis et al., 2023), and a more general framework of online primal-dual algorithms (Bamas et al., 2020b). However, there have been some recent applications to other areas, e.g. graph algorithms (Chen et al., 2022; Dinitz et al., 2021), causal learning (Choo et al., 2023), mechanism design (Gkatzelis et al., 2022; Agrawal et al., 2022), distribution learning (Bhattacharyya et al., 2025a;b), and fair division (Choo et al., 2026). For an overview of this growing area, we refer the reader to the survey by Mitzenmacher & Vassilvitskii (2022).[1] Learning-augmented algorithms (also called algorithms with predictions/advice) aim to exploit imperfect predictions without sacrificing worst-case guarantees, typically establishing *consistency* (good predictions

help), *robustness* (bad predictions do not hurt asymptotically), and often *smoothness* (graceful degradation with prediction error). In this work, we adapt this framework to elicitation for unanimous acceptability by considering predictions natural to governance pipelines—an ordering of "most constraining" agents, or a good candidate lottery—and analyzing how these predictions reduce the number of expensive hyperplane learning steps. Compared to the dominant learning-augmented literature (which is largely online/metric/optimization focused), our setting highlights a different prediction target: advice about which *constraints* will be active (or which candidate point is near the feasible region) in a feasibility with queries problem.

## 2. Model and Preliminaries

For any positive integer $z$, denote $[z] := \{1, \ldots, z\}$. Let $N = [n]$ be the set of $n$ *agents*,[2] and $S = \{s_1, \ldots, s_m\}$ be the set of $m$ *alternatives*. A *lottery* over $S$ is a probability vector $\mathbf{x} = (x_1, \ldots, x_m) \in \Delta(S)$, where $\Delta(S) := \{\mathbf{x} \in \mathbb{R}_{\geq 0}^m : \sum_{j=1}^m x_j = 1\}$ is the $(m-1)$-simplex. Let $\mathbf{e}_j$ denote the *pure* lottery placing unit mass on $s_j$.

Each agent $i \in N$ has an (unknown) *utility* (or *value*) $u_i(s_j) \in [0, 1]$ for each alternative $s_j \in S$,[3] and an (unknown) threshold $\tau_i \in (0, 1]$.[4] The *expected utility* of lottery $\mathbf{x}$ for agent $i$ is $U_i(\mathbf{x}) = \sum_{j=1}^m x_j\, u_i(s_j)$. Agent $i$ *accepts* $\mathbf{x}$ if and only if $U_i(\mathbf{x}) \geq \tau_i$. Define the acceptable region/set $\mathcal{A}_i := \{\mathbf{x} \in \Delta(S) : \langle \mathbf{u}_i, \mathbf{x} \rangle \geq \tau_i\}$, i.e., the intersection of the simplex with a halfspace. A lottery is *unanimously acceptable* if and only if $\mathbf{x} \in \bigcap_{i \in N} \mathcal{A}_i$.[5]

*Example* 2.1. Let $m = 2$. Suppose the agents have $u_1 = (1, 0)$, $\tau_1 = 0.6$ and $u_2 = (0, 1)$, $\tau_2 = 0.6$, so $\mathcal{A}_1 = \{\mathbf{x} \in \Delta(S) : x_1 \geq 0.6\}$ and $\mathcal{A}_2 = \{\mathbf{x} \in \Delta(S) : x_2 \geq 0.6\} = \{\mathbf{x} \in \Delta(S) : x_1 \leq 0.4\}$. Then $\mathcal{A}_1 \cap \mathcal{A}_2 = \varnothing$, and no unanimously acceptable lottery exists.

**Expected utility assumption.** Within the expected utility model, risk attitudes over deterministic outcomes can be encoded in the utility coefficients assigned to alternatives, and many hard option-level constraints can be encoded as linear threshold constraints or by removing infeasible options from the menu. Non-expected utility or tail-sensitive criteria fall outside this halfspace model; we discuss this limitation in Section 6 and Appendix B.1.

**Query model.** The algorithm does not observe $(\mathbf{u}_i, \tau_i)$.

---

[1]See also `https://algorithms-with-predictions.github.io/`.

[2]These are the *stakeholders* as mentioned in the introduction.

[3]We can also denote an agent's *utility vector* as $\mathbf{u}_i$, where the $j$-th entry is $u_i(s_j)$.

[4]We assume without loss of generality that no agent has threshold 0, otherwise they can trivially be ignored.

[5]We model preferences over lotteries via expected utility, so each agent's acceptance set is a halfspace on $\Delta(S)$; see Appendix B.1 for motivation and standard foundations.

It may issue *membership queries* of the form $\text{Query}(i, \mathbf{x})$ where $i \in N$ and $\mathbf{x} \in \Delta(S)$. The oracle returns `True` if and only if $\mathbf{x} \in \mathcal{A}_i$, and `False` otherwise. The *query complexity* is the number of oracle calls.

---

**Problem (Unanimous Acceptability via Queries)**

Given $n, m$, and query access to agents, output either:
  (i) a lottery $\mathbf{x} \in \Delta(S)$ such that $U_i(\mathbf{x}) \geq \tau_i$ for all $i \in N$, or
  (ii) NULL, certifying that $\bigcap_{i \in N} \mathcal{A}_i = \varnothing$.

---

Deterministic algorithms must be correct on all instances. Our randomized algorithms are always correct; randomness only affects the number of queries. We analyze the expected number of queries they use.

*Remark* 2.2. A natural relaxation to the unanimity objective is to maximize the *number of accepting agents*. However, we can show that even with an unlimited query budget (i.e., to obtain explicit access to $(\mathbf{u}_i, \tau_i)$), this optimization problem is NP-hard (see Appendix B.4 for a discussion).

We now illustrate the setting with a short, concrete example.

*Example* 2.3. Fix $m = 3$ alternatives $S = \{s_1, s_2, s_3\}$ and $\varepsilon = 0.1$. Consider three agents with utilities and thresholds:

| $i$ | $u_i(s_1)$ | $u_i(s_2)$ | $u_i(s_3)$ | $\tau_i$ |
|---|---|---|---|---|
| 1 | 1.0 | 0.6 | 0.2 | 0.6 |
| 2 | 0.2 | 1.0 | 0.5 | 0.7 |
| 3 | 0.2 | 0.2 | 1.0 | 0.3 |

For a lottery $\mathbf{x} = (x_1, x_2, x_3) \in \Delta(S)$, the acceptability sets $\mathcal{A}_i = \{\mathbf{x} \in \Delta(S) : \langle \mathbf{u}_i, \mathbf{x} \rangle \geq \tau_i\}$ are the following halfspaces intersected with the simplex: $\mathcal{A}_1$ ($x_1 + 0.6x_2 + 0.2x_3 \geq 0.6$); $\mathcal{A}_2$ ($0.2x_1 + x_2 + 0.5x_3 \geq 0.7$); $\mathcal{A}_3$ ($0.2x_1 + 0.2x_2 + x_3 \geq 0.3$).

Observe that no pure lottery is unanimously acceptable (e.g., $\mathbf{e}_1$ is rejected by agent 2, $\mathbf{e}_2$ by agent 3, and $\mathbf{e}_3$ by agent 1), but the lottery $\mathbf{x}^* = (0.25, 0.60, 0.15)$ is unanimously acceptable since $U_1(\mathbf{x}^*) = 0.64 \geq 0.6$, $U_2(\mathbf{x}^*) = 0.725 \geq 0.7$, and $U_3(\mathbf{x}^*) = 0.32 \geq 0.3$. Figure 1 visualizes the three halfspaces and their intersection.

**Finite precision.** Fix $\varepsilon \in (0, 1/2]$ with $1/\varepsilon \in \mathbb{Z}_{>0}$. We assume $\varepsilon$-quantized parameters: $u_i(s_j) \in \{0, \varepsilon, 2\varepsilon, \ldots, 1\}$ and $\tau_i \in \{\varepsilon, 2\varepsilon, \ldots, 1\}$. This is necessary for finite-query exactness and can be viewed as $\Theta(\varepsilon)$-robust unanimity (Appendix B.2 and B.3). For some counting-based lower bounds we additionally assume $m \leq (1/\varepsilon)^{1-\delta}$ for a fixed constant $\delta \in (0, 1]$ (see Appendices B.2 and B.3 for more details).

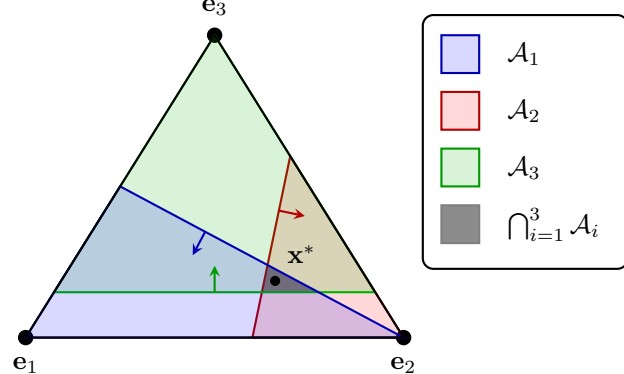

*Figure 1.* Example 2.3 ($m = 3$): halfspace acceptability regions $\mathcal{A}_1, \mathcal{A}_2, \mathcal{A}_3$ on the probability simplex. The vertices $\mathbf{e}_1, \mathbf{e}_2, \mathbf{e}_3$ are pure lotteries, arrows indicate the acceptable side of each constraint, and $\mathbf{x}^*$ is a feasible lottery in $\bigcap_{i=1}^{3} \mathcal{A}_i$ (gray).

# 3. Algorithms for Learning Unanimously Acceptable Lotteries

We begin by investigating elicitation algorithms that can give us an upper bound on the worst-case number of queries needed to either find a unanimously acceptable lottery, or certify that none exists.

We first focus on how we can (efficiently) elicit the acceptable region for a single agent. For a fixed agent $i$, acceptability is a single linear inequality: $U_i(\mathbf{x}) = \langle \mathbf{u}_i, \mathbf{x} \rangle \geq \tau_i$. Thus, the acceptable region is exactly a halfspace intersected with the simplex $\Delta(S)$. Our goal is to recover an equivalent (normalized) halfspace description using as few membership queries $\text{Query}(i, \mathbf{x})$ as possible.

Algorithm 1 (LearnHyperplane) learns an explicit halfspace description of agent $i$'s acceptable set. It begins by querying the vertices $\mathbf{e}_1, \ldots, \mathbf{e}_m$ to partition indices into $L^{\text{acc}}$ and $L^{\text{rej}}$. If $L^{\text{rej}} = \varnothing$ (resp., $L^{\text{acc}} = \varnothing$), then the agent accepts (resp., rejects) every lottery and we return ACCEPTALL (resp., REJECTALL). Otherwise, fix any rejected index $r \in L^{\text{rej}}$. For any edge between a rejected vertex $\mathbf{e}_k$ and an accepted vertex $\mathbf{e}_{k'}$, consider the edge lotteries $\mathbf{x}_{k,k',\alpha} := (1-\alpha)\mathbf{e}_k + \alpha\mathbf{e}_{k'}$ for $\alpha \in [0, 1]$. Along such an edge, $U_i(\mathbf{x}_{k,k',\alpha})$ is affine and strictly increasing in $\alpha$, so the accept/reject behavior changes exactly once. We call $\alpha^*_{i;k,k'} := \inf\{\alpha \in [0, 1] : \text{Query}(i, \mathbf{x}_{k,k',\alpha}) = \text{True}\} \in (0, 1]$ the *turning point* on edge $(\mathbf{e}_k, \mathbf{e}_{k'})$; equivalently, it is the smallest $\alpha$ that is accepted, and (since the boundary is the equality $\langle u_i, \mathbf{x} \rangle = \tau_i$) it satisfies $U_i(\mathbf{x}_{k,k',\alpha^*_{i;k,k'}}) = \tau_i$.

LearnHyperplane recovers $m - 1$ turning points on selected simplex edges (via ExactThreshold)—first on each edge $(\mathbf{e}_r, \mathbf{e}_j)$ for $j \in L^{\text{acc}}$, and then (if needed) on edges $(\mathbf{e}_k, \mathbf{e}_a)$ from the remaining rejected vertices to a fixed accepted $a$ with $\alpha^*_{i;r,a} < 1$. In the $(m-1)$-dimensional affine hull of $\Delta(S)$, these $m - 1$ edge intersections determine

---

**Algorithm 1** LearnHyperplane($i$)

---

**Input:** Agent $i \in N$

**Output:** ACCEPTALL, REJECTALL, or a vector $\mathbf{c}_i \in \mathbb{R}^m$ whose induced halfspace $\{\mathbf{x} \in \Delta(S) : \langle \mathbf{c}_i, \mathbf{x} \rangle \geq 1\}$ coincides with agent $i$'s acceptable region on $\Delta(S)$.

1: Query all pure lotteries $\mathbf{e}_1, \ldots, \mathbf{e}_m$ and let $L^{\mathrm{acc}} := \{j : j \in [m], \mathrm{Query}(i, \mathbf{e}_j) = \texttt{True}\}$, $L^{\mathrm{rej}} := [m] \setminus L^{\mathrm{acc}}$.

2: **if** $L^{\mathrm{rej}} = \varnothing$ **then return** ACCEPTALL

3: **else if** $L^{\mathrm{acc}} = \varnothing$ **then return** REJECTALL

4: **end if**

5: Choose any $r \in L^{\mathrm{rej}}$.

6: For distinct $k, k' \in [m]$ and $\alpha \in [0, 1]$, define $\mathbf{x}_{k,k',\alpha} := \alpha \mathbf{e}_{k'} + (1 - \alpha)\mathbf{e}_k$.

7: **for** each $j \in L^{\mathrm{acc}}$ **do**

8:     Find turning point $\alpha_{r,j} \leftarrow \mathrm{ExactThreshold}(i, r, j)$ on edge $(\mathbf{e}_r, \mathbf{e}_j)$ (i.e., $\alpha_{r,j} = \alpha^*_{i;r,j}$).

9: **end for**

10: **if** $\alpha_{r,j} = 1$ for all $j \in L^{\mathrm{acc}}$ **then**

11:     Construct $\mathbf{c}_i \in \mathbb{R}^m$ by setting $c_{i,j} = 1$ for $j \in L^{\mathrm{acc}}$ and $c_{i,k} = 0$ for $k \in L^{\mathrm{rej}}$.

12:     **return** $c_i$

13: **end if**

14: Choose any $a \in L^{\mathrm{acc}}$ such that $\alpha_{r,a} < 1$.

15: **for** each $k \in L^{\mathrm{rej}} \setminus \{r\}$ **do**

16:     Find turning point $\alpha_{k,a} \leftarrow$ $\mathrm{ExactThreshold}(i, k, a)$ on edge $(\mathbf{e}_k, \mathbf{e}_a)$ (i.e., $\alpha_{k,a} = \alpha^*_{i;k,a}$)

17: **end for**

18: Construct $\mathbf{c}_i = (c_{i,1}, \ldots, c_{i,m})$ as follows:
    Set $c_{i,r} = 0$.
    For each $j \in L^{\mathrm{acc}}$, set $c_{i,j} = 1/\alpha_{r,j}$.
    For each $k \in L^{\mathrm{rej}} \setminus \{r\}$, set $c_{i,k} = \dfrac{1 - \alpha_{k,a}\, c_{i,a}}{1 - \alpha_{k,a}}$.

19: **return** $\mathbf{c}_i$

---

the boundary hyperplane up to scaling; the algorithm normalizes to produce a vector $\mathbf{c}_i$ such that for all $\mathbf{x} \in \Delta(S)$, $\langle \mathbf{c}_i, \mathbf{x} \rangle \geq 1$ if and only if agent $i$ accepts $\mathbf{x}$. The subroutine ExactThreshold is a one-dimensional search along an edge and uses $\mathcal{O}(\log(1/\varepsilon))$ queries per turning point (Appendix C.1). We also provide a geometric intuition of the algorithm in Appendix C.2.

Formalizing this idea, we present LearnHyperplane, which elicits the supporting hyperplane of a single agent's acceptable region as described above, and by using only $\mathcal{O}(m \log(1/\varepsilon))$ queries.

We now establish two key properties of LearnHyperplane. The first concerns query complexity: we show that the algorithm identifies all necessary threshold points using only $\mathcal{O}(\log(1/\varepsilon))$ queries. Lemma 3.1 formalizes this bound.

**Lemma 3.1.** *Algorithm 1 makes $\mathcal{O}(m \log(1/\varepsilon))$ queries.*

Our next step is to show that the resulting vector exactly encodes the boundary of the agent's acceptable region.

**Lemma 3.2.** *Fix an agent $i \in N$. Algorithm 1 returns one of the following: (i) ACCEPTALL, in which case $U_i(\mathbf{x}) \geq \tau_i$ for all $\mathbf{x} \in \Delta(S)$; (ii) REJECTALL, in which case $U_i(\mathbf{x}) < \tau_i$ for all $\mathbf{x} \in \Delta(S)$; or (iii) a vector $\mathbf{c}_i \in \mathbb{R}^m$ such that for all $\mathbf{x} \in \Delta(S)$, $\langle \mathbf{c}_i, \mathbf{x} \rangle \geq 1$ if and only if $U_i(\mathbf{x}) \geq \tau_i$. In case (iii), the induced halfspace $\{\mathbf{x} \in \Delta(S) : \langle \mathbf{c}_i, \mathbf{x} \rangle \geq 1\}$ coincides with agent $i$'s acceptable set.*

These two lemmas show that an agent's acceptance region can be elicited and represented as a single linear constraint. This reduces the multi-agent elicitation problem to determining whether the intersection of the learned halfspaces is nonempty. Intersecting the learned halfspaces across agents and solving a feasibility linear program (LP) allows us to find a unanimously acceptable lottery or certify that none exists (without requiring further queries).

A simple baseline is to run LearnHyperplane($i$) for *every* agent $i \in N$ and then solve the resulting feasibility LP. This full elicitation strategy makes $\mathcal{O}(nm \log(1/\varepsilon))$ queries. However, aside from the trivial early termination case where some agent rejects all lotteries, it always learns all $n$ hyperplanes, so its *best-case* query complexity on feasible instances is still $\Theta(nm \log(1/\varepsilon))$. In many instances this is unnecessarily pessimistic. Intuitively, feasibility (and the particular lottery we end up returning) might be "certified" by only a small subset of constraints, so learning the remaining agents' hyperplanes provides no additional benefit. This motivates an adaptive deterministic algorithm (Algorithm 2) that pays for hyperplane elicitation only when it is forced by an observed violation.

Fix a deterministic tie-breaking rule: define $\mathrm{Select}(C)$ to return the lexicographically maximum feasible lottery $\mathbf{x}$ among all $\mathbf{x} \in \Delta(S)$ satisfying $\langle \mathbf{c}, \mathbf{x} \rangle \geq 1$ for all $\mathbf{c} \in C$, and return NULL if infeasible.[6]

Intuitively, Algorithm 2 maintains a set $C$ of learned acceptability halfspaces and proposes a candidate $\mathbf{x} \leftarrow \mathrm{Select}(C)$. It then queries the remaining (unlearned) agents. If every unlearned agent accepts $\mathbf{x}$, then $\mathbf{x}$ is unanimously acceptable (all previously learned agents accept $\mathbf{x}$ by construction). Otherwise, upon encountering the first unlearned agent $i$ that rejects $\mathbf{x}$, the algorithm invokes LearnHyperplane($i$) to elicit $i$'s acceptability halfspace, adds the resulting constraint to $C$, and immediately restarts with a new candidate lottery. The algorithm reports infeasibility either when $\mathrm{Select}(C) = $ NULL or when LearnHyperplane($i$) returns REJECTALL. Our result is as follows.

---

[6] Equivalently, Select can be implemented by solving a sequence of LPs that maximize $x_1$, then $x_2$, etc., over the feasible region. This means it can be implemented in polynomial time. Our focus is the number of membership queries, so we treat this offline computation cost as secondary.

**Algorithm 2** Deterministic algorithm that returns a unanimously acceptable lottery if one exists, or NULL otherwise

**Input:** Set of agents $N = \{1, \ldots, n\}$ and set of alternatives $S = \{s_1, \ldots, s_m\}$
**Output:** Lottery $\mathbf{x} = (x_1, \ldots, x_m) \in \Delta(S)$ such that $U_i(\mathbf{x}) \geq \tau_i$ for all $i \in N$, or NULL
  1: $C \leftarrow \varnothing$      ▷ learned hyperplane constraints
  2: $N' \leftarrow N$    ▷ agents whose hyperplanes not learned
  3: **while** True **do**
  4:    $\mathbf{x} \leftarrow \text{Select}(C)$
  5:    **if** $\mathbf{x} = $ NULL **then return** NULL
  6:    **end if**
  7:    $j \leftarrow $ None    ▷ keep track of the first violator
  8:    **for** $i = 1, \ldots, n$ **do**
  9:      **if** $i \notin N'$ **then continue**
 10:      **else if** $\text{Query}(i, \mathbf{x}) = $ False **then**
 11:        $j \leftarrow i$; break     ▷ violator
 12:      **end if**
 13:    **end for**
 14:    **if** $j = $ None **then return** $\mathbf{x}$
 15:    **end if**
 16:    $\mathbf{c} \leftarrow \text{LearnHyperplane}(j)$; $N' \leftarrow N' \setminus \{j\}$
 17:    **if** $\mathbf{c} = $ REJECTALL **then return** NULL
 18:    **else** $C \leftarrow C \cup \{\mathbf{c}\}$
 19:    **end if**
 20: **end while**
 21: **return** $\mathbf{x}$

**Theorem 3.3.** *Algorithm 2 returns a unanimously acceptable lottery when one exists, and outputs* NULL *otherwise, using* $\mathcal{O}\left(n^2 + nm \log(1/\varepsilon)\right)$ *queries.*

Note that the extra $n^2$ term arises solely from verification queries in the worst case: each time a new hyperplane is learned, the new candidate $\mathbf{x} \leftarrow \text{Select}(C)$ must be checked against the remaining unlearned agents. Note that if the number of agents on which Algorithm 2 calls LearnHyperplane on is small, then we can avoid learning most agents' halfspaces and can be substantially cheaper than full elicitation.

A natural follow-up question is whether one can do better than eliciting the hyperplane of *every* agent. In many governance deployments, $m$ (the menu size) is modest while $n$ (stakeholders) can be very large. In such settings, algorithms that avoid learning all $n$ halfspaces are attractive, motivating the randomized approach below. Specifically, we show that randomization (via Algorithm 3, details in Appendix C.6) gives an improvement on the *expected* number of hyperplane learning we have to do. As with any randomized guarantee, the worst case over the internal randomness can still be as large as $n$; our improvement is in expectation.

**Theorem 3.4.** *There exists a randomized algorithm that returns a unanimously acceptable lottery when*

one exists, and outputs NULL *otherwise, using* $\mathcal{O}\left(nm \log n + \min\{n, m^3 \log n\} \cdot m \log(1/\varepsilon)\right)$ *queries in expectation.*

*Proof idea (informal).* Our algorithm adapts the Clarkson (1995) low-dimensional LP sampling approach to our model: each round samples a small (weight-biased) set of agents, learns only their halfspaces, solves the sampled LP to get a candidate lottery $\mathbf{x}$, then verifies $\mathbf{x}$ against all agents and multiplicatively upweights the violators. The geometric reason this works is that in dimension $m - 1$ there is always a small set of $\mathcal{O}(m)$ agents that certifies feasibility/infeasibility (and pins down the tie-broken optimum when feasible); any failed verification must violate at least one witness agent, so repeated up-weighting quickly forces the sample to include all witnesses, after which verification certifies correctness. The novel consideration here is our *oracle strength*: we do not get violated constraints for free from a separation oracle, so we must explicitly elicit (and cache) sampled agents' constraints via LearnHyperplane. $\square$

Thus, the randomized algorithm reduces the number of expensive hyperplane learning occurrences to $\mathcal{O}(\min\{n, m^3 \log n\})$ in expectation, at the cost of $\mathcal{O}(m \log n)$ global verification rounds. Intuitively, if $n \gg m^3 \log n$, then the expected number of queries is $\mathcal{O}(nm \log n + m^4(\log n) \log(1/\varepsilon))$; whereas if $n \leq m^3 \log n$, the second term becomes $\mathcal{O}(nm \log(1/\varepsilon))$.

*Remark* 3.5 (Settings where Theorem 3.4 brings about improvement). Theorem 3.4 is most useful when the menu of deployment choices is small but the pool of users, evaluators, or stakeholders is large. The following examples illustrate settings in which this separation between $m$ and $n$ can arise. One example is canary rollout, where the choice set is typically very small. Google Cloud Deploy defines canary as splitting traffic between an already-deployed version and a new version, and Istio (Google, n.d.) explicitly supports A/B testing, canary rollouts, and staged rollouts with percentage-based traffic splits such as 75/25 or "20% of calls go to the new version". A second example is online experimentation, where the same geometry is standard: A/B/n tests involve only a small number of variants, while each experiment may expose very large user populations, often numbering in the millions (Kohavi, 2015; Deng et al., 2013). A third example is responsible AI evaluation, where the number of candidate models or deployment modes may again be small, while the number of evaluators can be very large. NIST's Generative AI Profile (NIST, 2024) explicitly recommends structured public feedback and large-group AI red-teaming; Chatbot Arena (Chiang et al., 2024) reports over 240K crowdsourced votes; and the DEF CON public Generative AI red-team event involved 2,244 participants and more than 17,000 conversations. These are precisely settings in which $m$ is small but $n$ can be very large. More

broadly, Theorem 3.4 is aimed at settings with a curated menu of rollout or model choices and a large pool of users, stakeholders, or evaluators.

We also remark that finding explicit infeasibility witnesses comes at no additional query cost (see Appendix C.7), which may be useful in governance/audit applications.

# 4. Lower Bounds on Query Complexity

The previous section establishes query-efficient algorithms for either finding a unanimously acceptable lottery or certifying infeasibility under membership-query access. We now ask how far one can improve these bounds in the worst case. Our lower bounds apply to both deterministic and randomized algorithms, and they already hold under binary utilities ($u_i(s_j) \in \{0, 1\}$).

**Theorem 4.1.** *Any (deterministic or randomized) algorithm that, on every instance, outputs a unanimously acceptable lottery when one exists (and correctly reports infeasibility otherwise), has a worst-case (expected) number of queries of $\Omega\left((n - \min\{n, m\}) + (\min\{n, m\} - 1)\log(1/\varepsilon)\right)$.*

Relative to Section 3, our algorithms have the correct qualitative dependence on $\varepsilon$ (via the $\Theta(\log(1/\varepsilon))$ threshold search cost). The remaining gaps are in how many times agents must be *re-queried* during verification. Algorithm 2 has an $\mathcal{O}(n^2)$ verification overhead in the worst case, while Algorithm 3 reduces the number of verification rounds to $\mathcal{O}(m \log n)$ in expectation (and thus $\mathcal{O}(nm \log n)$ verification queries). When $m$ is modest, this brings the dependence on $n$ close to the $\Omega(n)$ lower bound up to logarithmic factors, while simultaneously ensuring that only a small number of agents' halfspaces must be explicitly learned in expectation.

The next theorem isolates an orthogonal difficulty: even with a single agent, no algorithm can hope to achieve a polylogarithmic dependence on $m$ in our model.

**Theorem 4.2.** *Fix any $m \geq 2$. Even for $n = 1$, any correct algorithm must make $\Omega(m)$ queries in the worst case.*

# 5. Learning-Augmented Algorithms

The algorithms in Section 3 are designed for the worst case: they assume the algorithm has no prior information about which agents are likely to be constraining, or where a unanimously acceptable lottery might lie. In many applications, this is overly pessimistic. In particular, governance and safety evaluation pipelines often come with side information—historical data about which checks tend to bind, preliminary offline evaluations, domain expertise, or even heuristic rankings produced by other systems. While such advice can be imperfect, it can nevertheless capture meaningful structure that an elicitation algorithm can exploit to reduce the number of expensive stakeholder queries.

Learning-augmented algorithms (also called algorithms with advice/predictions) provide a framework for leveraging such information without sacrificing worst-case guarantees. The algorithm receives a prediction about some aspect of the instance, but must remain correct even if the prediction is arbitrarily wrong. We seek three standard properties: *consistency* (perfect advice results in fewer queries), *robustness* (arbitrarily bad advice does not worsen the asymptotic worst-case guarantees), and *smoothness* (performance degrades gradually as a function of a prediction quality parameter).

In our setting, advice is especially natural because the dominant query cost arises from learning agents' halfspaces: each call to LearnHyperplane costs $\mathcal{O}(m \log(1/\varepsilon))$ queries. Thus, even coarse predictions that reduce (i) the number of agents whose hyperplanes must be elicited, or (ii) the time until the algorithm queries the set of "constraining" agents, can translate into substantial query savings.

We consider two forms of predictions: *permutation predictions*, which provide an ordering of agents intended to place more constraining agents early, and *lottery predictions*, which provide a candidate lottery believed to be promising.[7]

## 5.1. Permutation Predictions

A natural and common kind of advice in practice is a ranking of stakeholders by "likely constraining"; for example, based on past deployment reviews or on preliminary screening. Algorithm 2 scans agents to find the first violator of the current candidate lottery; if a violator is found, the algorithm pays the full cost of eliciting that agent's halfspace. This suggests that a good scan order can substantially reduce the number of costly elicitation steps.

Let $\sigma$ be any permutation of the agents $N$. Define Algorithm 2$[\sigma]$ to be Algorithm 2 with the only change that, in each iteration, the algorithm scans unlearned agents in the order $\sigma(1), \ldots, \sigma(n)$ (instead of $1, \ldots, n$). Let $H(\sigma) \subseteq N$ denote the set of agents on which LearnHyperplane is invoked during the execution of Algorithm 2$[\sigma]$, and define $R(\sigma) := |H(\sigma)|$.[8] We call $R(\sigma)$ the number of *record agents* under order $\sigma$: it is exactly the number of times the algorithm is forced to pay for hyperplane elicitation. Smaller $R(\sigma)$ corresponds to better advice.

A permutation predictor outputs an ordering $\hat{\sigma}$ intended to place "more constraining" agents early. Importantly, there need not be a uniquely most constraining agent: which con-

---

[7]These two prediction types are illustrative rather than exhaustive; they simply demonstrate that natural, coarse advice can reduce query complexity while preserving worst-case guarantees.

[8]If $R(\sigma) = 0$, then the very first candidate lottery (computed from an empty constraint set) is already unanimously acceptable, and the algorithm terminates after only the $n$ verification queries. Thus, we assume $R(\sigma) \geq 1$ for notational simplicity.

straints become active depends on the sequence of candidate lotteries produced by Select. We therefore interpret "constraining" in an operational sense: an agent is constraining for order $\sigma$ if it is ever encountered as a violator before its halfspace is learned. The most direct use of a predicted permutation $\hat{\sigma}$ is to run Algorithm 2 with the agent order $\hat{\sigma}$ in the **for** loop on Line 8 (which we denote as Algorithm 2$[\hat{\sigma}]$). We treat $R(\hat{\sigma})$ as the prediction quality parameter.[9]

**Theorem 5.1.** *Let $\hat{\sigma}$ be a predicted permutation of the agents. Then Algorithm 2$[\hat{\sigma}]$ outputs a unanimously acceptable lottery if one exists, and* NULL *otherwise, using* $\mathcal{O}\left((n + m\log(1/\varepsilon))\,R(\hat{\sigma})\right)$ *queries.*

When advice is perfect (i.e., $R(\hat{\sigma}) = 1$), the algorithm of Theorem 5.1 uses only $\mathcal{O}(n + m\log(1/\varepsilon))$ queries; note that $\Omega(n + m)$ queries are necessary from Section 4. Meanwhile, even with arbitrarily bad advice, the query complexity never exceeds the advice-free worst-case guarantee of Algorithm 2 since $R(\hat{\sigma}) \leq n$ for any permutation. Finally, the linear dependence on $R(\hat{\sigma})$ gives us a smooth performance degradation as prediction quality worsens.

Permutation advice can also improve the randomized algorithm (Algorithm 3), which repeatedly samples agents according to a weight vector, solves the sampled subproblem, and multiplicatively upweights violators. In the advice-free analysis, the expected number of iterations scales as $\mathcal{O}(m\log n)$ because the algorithm must (in effect) discover a small witness set among $n$ agents. A predicted order allows us to bias sampling toward agents that appear early, reducing the effective $\log n$ dependence when a witness set is contained in a short prefix of $\hat{\sigma}$.

Let $\mathrm{rank}_{\hat{\sigma}}(i) \in \{1, \ldots, n\}$ denote the position of agent $i$ in $\hat{\sigma}$. Recall from the analysis of Algorithm 3 that correctness can be certified by a small *witness set*: in the feasible case, there is a basis of size at most $m - 1$ that determines the final (tie-breaking) solution; in the infeasible case, there is a *Helly witness* of size at most $m$, i.e., a set $B$ with $\bigcap_{i \in B} \mathcal{A}_i = \varnothing$. Let $\mathcal{W}$ denote the family of valid witness sets used in the analysis (bases in the feasible case, Helly witnesses in the infeasible case) and define the *prefix witness parameter* $E(\hat{\sigma}) := \min_{B \in \mathcal{W}} \max_{i \in B} \mathrm{rank}_{\hat{\sigma}}(i)$, i.e., $E(\hat{\sigma})$ is the smallest $k$ such that the first $k$ agents of $\hat{\sigma}$ contains some $B \in \mathcal{W}$. We define Algorithm 3$[\hat{\sigma}]$ as Algorithm 3 with an advice-biased initialization of the sampling weights: set the initial integer weight of agent $i$ to $w_i^{(1)} := \lceil n/\mathrm{rank}_{\hat{\sigma}}(i) \rceil$, and then run the same sampling and multiplicative updating procedure as in Algorithm 3.

---

[9]The prediction $\hat{\sigma}$ is a single global permutation; it is not a separate ranking recomputed for each candidate lottery. The algorithms remain adaptive despite this one-shot advice: Algorithm 2$[\hat{\sigma}]$ uses the order only when scanning unlearned agents, while Algorithm 3$[\hat{\sigma}]$ uses it only to initialize sampling weights before the usual multiplicative updates driven by observed violations.

**Theorem 5.2.** *Let $\hat{\sigma}$ be a predicted permutation of the agents. Then Algorithm 3$[\hat{\sigma}]$ outputs a unanimously acceptable lottery if one exists, and* NULL *otherwise, using* $\mathcal{O}(nm\mu + \min\{n, m^3\mu\} \cdot m\log(1/\varepsilon))$ *queries in expectation, where $\mu = \log E(\hat{\sigma}) + \log\log n$.*

The above result recovers the advice-free bound (up to constants) when the advice is arbitrary: since $E(\hat{\sigma}) \leq n$ always, we have $\mu = \mathcal{O}(\log n)$. When the advice is perfect, the dependence on $\log n$ in the advice-free analysis is replaced by $\log E(\hat{\sigma}) + \log\log n$. The bound is monotone in $E(\hat{\sigma})$, giving us smooth degradation as the prediction worsens.

We complement Theorems 5.1 and 5.2 by proving, in Appendix E.3, that even when the predicted permutation is perfect, any algorithm that must remain correct for arbitrary advice still requires $\Omega((n - \min\{n, m\}) + (\min\{n, m\} - 1)\log(1/\varepsilon))$ queries, as in the setting without advice.

### 5.2. Lottery Predictions

A second natural form of advice is a *lottery prediction*: the predictor provides a candidate lottery $\hat{\mathbf{x}} \in \Delta(S)$ that is believed to be informative for the instance.[10] The most direct use of $\hat{\mathbf{x}}$ is to verify it once: query each agent $i \in N$ on $\hat{\mathbf{x}}$. If $\mathrm{Query}(i, \hat{\mathbf{x}}) = \texttt{True}$ for all $i$, then $\hat{\mathbf{x}}$ is unanimously acceptable and the algorithm can terminate after exactly $n$ membership queries.

When $\hat{\mathbf{x}}$ is not unanimously acceptable, we can still leverage it to reduce the query cost of learning an individual agent's acceptability halfspace. Recall that $\mathrm{LearnHyperplane}(i)$ (Algorithm 1) identifies agent $i$'s boundary by locating $m - 1$ turning points along simplex edges, using $\mathrm{ExactThreshold}$ (Appendix C.1). Each such call performs a global bisection on an interval of length 1, costing $\Theta(\log(1/\varepsilon))$ queries.

Lottery advice provides a natural warm start for these one-dimensional searches. For distinct $k, k' \in [m]$, recall the edge lotteries $\mathbf{x}_{k,k',\alpha} := \alpha e_{k'} + (1 - \alpha)e_k$ for all $\alpha \in [0, 1]$. Given any $\mathbf{x} \in \Delta(S)$, define the pairwise projection coordinate $\widehat{\alpha}_{k,k'}(\mathbf{x})$, which takes on a value of $x_{k'}/(x_k + x_{k'})$ if $x_k + x_{k'} > 0$, and $1/2$ otherwise ($x_k + x_{k'} = 0$). Equivalently, $\widehat{\alpha}_{k,k'}(\mathbf{x})$ is the unique $\alpha \in [0, 1]$ such that the edge lottery $\mathbf{x}_{k,k',\alpha}$ has the same *relative* mass on $\{\mathbf{e}_k, \mathbf{e}'_k\}$ as $\mathbf{x}$ after renormalizing away the other $m - 2$ coordinates.

Fix an agent $i \in N$ and an ordered pair $(k, k')$ with $\mathrm{Query}(i, \mathbf{e}_k) = \texttt{False}$ and $\mathrm{Query}(i, \mathbf{e}_{k'}) = \texttt{True}$. As argued in Section 3, there is a unique turning point $\alpha^*_{i;k,k'} := \inf\{\alpha \in [0, 1] : \mathrm{Query}(i, \mathbf{x}_{k,k',\alpha}) = \texttt{True}\} \in (0, 1]$. Then, given a prediction $\hat{\mathbf{x}}$, define the (edge) turning point projection error $\delta_{i;k,k'}(\hat{\mathbf{x}}) := |\widehat{\alpha}_{k,k'}(\hat{\mathbf{x}}) - \alpha^*_{i;k,k'}|$. We illus-

---

[10]For example, $\hat{\mathbf{x}}$ may come from offline evaluation, a heuristic search procedure, or a previous deployment cycle. The algorithm must remain correct even if $\hat{\mathbf{x}}$ is arbitrary.

trate the defined quantities with Figure 2.

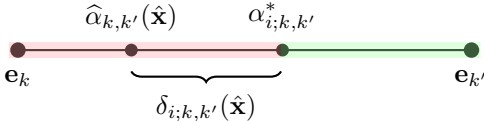

*Figure 2.* Each point on this line segment is a lottery. The red (resp., green) region indicates the set of rejected (resp., accepted) lotteries along this edge. A lottery prediction induces a projected turning point which can be used as a warm start $\widehat{\alpha}_{k,k'}(\hat{\mathbf{x}})$ to find the turning point $\alpha^*_{i;k,k'}$ faster.

Let $\mathcal{E}_i$ denote the set of ordered pairs $(k, k')$ on which LearnHyperplane invokes ExactThreshold when run on agent $i$ (i.e., the set of simplex edges whose turning points are actually queried); note that $|\mathcal{E}_i| \leq m - 1$. Define the per-agent prediction error $\delta_i^{\max}(\hat{\mathbf{x}}) := \max_{(k,k') \in \mathcal{E}_i} \delta_{i;k,k'}(\hat{\mathbf{x}})$.

We now define LearnHyperplane$[\hat{\mathbf{x}}]$ to be LearnHyperplane with a modification that every call to ExactThreshold$(i, k, k')$ is replaced by a warm-started routine ExactThresholdPred$(i, k, k', \widehat{\alpha}_{k,k'}(\hat{\mathbf{x}}))$ (Algorithm 4 in Appendix E.4).

**Theorem 5.3.** *Fix an agent $i \in N$ and a predicted lottery $\hat{\mathbf{x}} \in \Delta(S)$. Then* LearnHyperplane$[\hat{\mathbf{x}}](i)$ *returns one of the following: (i)* ACCEPTALL, *(ii)* REJECTALL, *or (iii) a vector $\mathbf{c}_i$ whose induced halfspace coincides with agent $i$'s acceptable set on $\Delta(S)$. Moreover, it has a query complexity of $\mathcal{O}(m + m \log(1 + \delta_i^{\max}(\hat{\mathbf{x}})/\varepsilon^2))$.*

The above result gives a learning-augmented bound for the subroutine that learns a single agent's acceptable halfspace. It is robust: for any prediction $\hat{\mathbf{x}}$, the query complexity is never worse (up to constants) than the baseline $\mathcal{O}(m \log(1/\varepsilon))$ bound. It is also consistent under perfect edge information: if $\delta_{i;k,k'}(\hat{\mathbf{x}}) = 0$ for all $(k, k') \in E_i$, then each warm-started threshold search uses only $\mathcal{O}(1)$ additional queries (no logarithmic factor), so the subroutine runs in $\mathcal{O}(m)$ queries total (including the $m$ pure lottery queries). More generally, the overhead above $m$ scales as $\sum_{(k,k') \in \mathcal{E}_i} \log(1 + \delta_{i;k,k'}(\hat{\mathbf{x}})/\varepsilon^2)$, giving us a smooth degradation with prediction error.

Then, Algorithms 2 and 3 can incorporate lottery advice by first verifying $\hat{\mathbf{x}}$ (early termination if unanimously accepted), and otherwise replacing every call to LearnHyperplane by LearnHyperplane$[\hat{\mathbf{x}}]$. This preserves correctness, and simply replaces the per-agent elicitation cost $\mathcal{O}(m \log(1/\varepsilon))$ by the refined bound in Theorem 5.3.[11]

Next, we show that the logarithmic dependence on the turning point projection error in Theorem 5.3 is necessary, even when $m = 2$, with the following lower bound.

**Proposition 5.4.** *Fix $\delta \in [\varepsilon^2, 1]$, and define $M :=$*

---

[11]Note that we do not require any relation between $\hat{\mathbf{x}}$ and feasibility; we only exploit whatever pairwise information it encodes.

$\min\{\lfloor 1/(2\varepsilon) \rfloor, \ \lfloor \delta/\varepsilon^2 \rfloor + 1\}$. *There exist $n = m = 2$, a lottery prediction $\widehat{\mathbf{x}} \in \Delta(S)$, and a family of instances $\{\mathcal{I}_t\}_{t=0}^{M-1}$ such that each $\mathcal{I}_t$ has a unique unanimously acceptable lottery $\mathbf{x}_t^* = \mathbf{x}_{1,2,\alpha_t}$, and for every $t$ the turning point projection error for agent 2 on edge $(\mathbf{e}_1, \mathbf{e}_2)$ in instance $\mathcal{I}_t$ satisfies $\delta_{2;1,2}(\widehat{x}) = |\alpha_{b1,2}(\widehat{x}) - \alpha_t| \leq \delta$. For this fixed $\widehat{\mathbf{x}}$, any correct query algorithm must make $\Omega\left(\log(1 + \delta/\varepsilon^2)\right)$ queries in the worst case (over the choice of $t$), in expectation for randomized algorithms.*

## 6. Conclusion and Future Work

We study the feasibility problem underlying conservative deployment gates: using only accept/reject feedback to proposed lotteries, the goal is to find a unanimously acceptable lottery or certify infeasibility. We give query-efficient deterministic, randomized, and learning-augmented elicitation algorithms that exploit the halfspace geometry of acceptability on the simplex. Complementary lower bounds show unavoidable linear dependence on the number of stakeholders and $\Theta(\log(1/\varepsilon))$ dependence on the target precision (and $\Omega(m)$ even for a single stakeholder).

Many interesting future work directions remain. In many governance settings, stakeholder judgments may be noisy, context-dependent, or costly to elicit at high precision. Extending our guarantees to stochastic or approximate oracles is an important future direction. Another direction is to relax strict unanimity: rather than requiring every agent to accept the same lottery, one could study approximation notions, or weighted stakeholders. Our model also does *not* capture non-expected utility or tail-sensitive criteria (e.g., CVaR, rank-dependent utility) (Rockafellar et al., 2000; Quiggin, 1982; Gilboa & Schmeidler, 1989). Since our techniques rely on halfspace structure, extending them to such criteria would require new geometric or oracle-based ideas.

### Acknowledgments

Paul Goldberg and Nicholas Teh are supported by the Advanced Research + Invention Agency (ARIA) as part of the ASPAI project.

### Impact Statement

This paper studies the information cost of satisfying multiple stakeholders' acceptability constraints when feedback is limited to binary accept/reject responses to proposed (randomized) deployment choices. A potential positive impact is that query-efficient elicitation procedures could reduce the burden of stakeholder evaluation and help surface infeasibility early, prompting designers to expand or revise the menu of deployment options rather than repeatedly testing non-viable compromises.

At the same time, the model is intentionally stylized for theoretical tractability (e.g., noise-free binary feedback and expected utility acceptability). As a result, the formal guarantees should be interpreted as guarantees *within* this model: applying the procedures in real governance settings would require careful attention to measurement noise, distribution shift, and risk-sensitive or tail-focused criteria that are not captured by expected utility theory. More broadly, randomized mixtures of deployment modes can interact in subtle ways with threshold-based approval processes. We view these considerations as motivating (i) clear communication of modeling assumptions and (ii) extensions to more realistic oracle and acceptability models (e.g., noisy feedback and risk-sensitive acceptance) as future work.

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

# Appendix

## A. Additional Related Work

**Pluralistic alignment and aggregation of diverse feedback.** A growing body of alignment work emphasizes that advanced AI systems must serve users and stakeholders with diverse values, and that aggregating feedback is not merely an engineering detail but a normative and algorithmic problem. Position papers argue explicitly for importing tools from social choice into modern alignment pipelines, e.g., when human feedback diverges across raters or populations (Conitzer et al., 2024; Sorensen et al., 2024). In practice, alignment procedures such as reinforcement learning from human feedback and related approaches elicit human judgments over candidate outputs and use them to shape model behavior (Ouyang et al., 2022; Bai et al., 2022). Our work is complementary in focus: rather than proposing a new aggregation rule or training method, we study the *query complexity* of reaching a conservative "deployment gate" style guarantee in which *every* stakeholder's acceptability threshold is met. This viewpoint captures settings where decisions must satisfy all relevant constraints (e.g., safety, compliance, or governance checks) and where feedback collection is costly, motivating algorithms that obtain just enough information to either (i) output a unanimously acceptable lottery, or (ii) certify infeasibility.

**Lotteries.** Lotteries are standard in decision theory and game theory (von Neumann & Morgenstern, 1944; Nash, 1950; Osborne & Rubinstein, 1994). Our use of lotteries is purely algorithmic: randomization enlarges the feasible set under linear (expected utility) acceptability constraints, but we do not study equilibrium selection or welfare optimality. From a mechanism design perspective, acceptability thresholds can be viewed as an individual participation or safety constraint, but we deliberately abstract away strategic reporting and incentive issues (Myerson, 1981; Nisan et al., 2007); our focus is the information cost (queries) required to find a unanimously feasible decision in the presence of unknown constraints.

**Query-efficient AI system selection.** Concurrent work by Neoh et al. (2026) studies a complementary query-efficient selection problem for AI deployment: given a set of candidate experts or models, the goal is to choose a small ensemble whose best member performs well on future tasks drawn from an unknown domain distribution. Their feedback models are binary correctness queries and pairwise preference queries over task-expert evaluations, and their algorithms exploit coverage-style structure to reduce evaluation cost. In contrast, our decision object is a lottery over alternatives, our feedback is stakeholder-level accept/reject feedback about proposed lotteries, and our objective is unanimous feasibility rather than approximate ensemble performance. Thus, both works study pre-deployment decision making from coarse feedback, but they address different feasibility/optimization goals and different geometric structures.

**Constrained decision-making and safe reinforcement learning.** Constrained MDPs and safe RL study decision-making under constraints, including feasibility and optimization of expected returns subject to safety constraints (Puterman, 1994; Altman, 1999; García & Fernández, 2015; Sutton & Barto, 2018). These works typically assume access to trajectories/samples and an explicit constraint specification (or a simulator), whereas our model isolates a different bottleneck: the cost of eliciting whether candidate (randomized) decisions satisfy each stakeholder's acceptability constraint. Our results therefore complement, rather than replace, the safe RL literature: they provide query-efficient methods for the approval/constraint checking component that can appear in governance loops.

**Query models in game theory.** A separate query-based line of work studies computing equilibria or best-response structure when payoffs are not explicitly given (e.g., via payoff queries or best-response queries). For example, Goldberg & Cossío (2021) study learning the partition of a mixed-strategy simplex induced by best responses, enabling equilibrium computation from such queries. Our oracle is strictly weaker (binary accept/reject rather than payoffs or best responses) and our goal is feasibility rather than equilibrium.

**Low-dimensional linear programming.** Once constraints are explicit (i.e., agents' preferences are fully known), unanimous acceptability reduces to feasibility of a low-dimensional linear program over the simplex. Classical convexity results such as Helly's theorem and its refinements imply that feasibility (or infeasibility) can be certified by a small witness set of constraints (of size $\mathcal{O}(m)$) (Helly, 1923; Danzer et al., 1963; Sharir & Welzl, 1992). Randomized algorithms for low-dimensional linear programming (e.g., randomized incremental and sampling/reweighting methods) exploit precisely this structure (Seidel, 1991; Clarkson, 1995; Matoušek et al., 1996). Our randomized elicitation algorithm is inspired by this line of work, but under a strictly weaker oracle model: we do not receive violated constraints (separating hyperplanes) for free, and instead must learn an agent's halfspace using accept/reject queries on lotteries. Our contribution is to show how this structure can still be leveraged in our query-limited setting, giving us an improved query complexity and accompanying lower bounds.

**Property testing/lower bounds for halfspaces.** Lower bounds for learning or testing halfspaces in various oracle models provide useful context for our $\varepsilon$-precision assumption and for understanding which tasks inherently require many queries. For example, Chen et al. (2025) prove lower bounds for testing halfspaces under a relative-error criterion in a property testing model. While their model is different from ours, it reinforces that seemingly modest changes in feedback/oracle access can qualitatively change the complexity of halfspace-related tasks.

**Maximum feasible subsystems.** Relaxing unanimity to maximize the number (or weight) of satisfied constraints connects to the *maximum feasible subsystem* problem, which is NP-hard and hard to approximate in general (Amaldi & Kann, 1995). Our paper shows that even under strong structural restrictions induced by lotteries over a simplex, optimizing the number of satisfied agents remains NP-hard, motivating unanimity as a clean baseline where the computational challenge lies primarily in elicitation rather than offline optimization.

**Portioning and budget aggregation.** Our model is also related to the portioning/budget aggregation literature, which studies how to aggregate opinions about how a divisible public resource (e.g., money or time) should be split across alternatives (Freeman et al., 2021; Airiau et al., 2023; Elkind et al., 2026). We refer the reader to the recent survey by Suksompong & Teh (2026) for a comprehensive overview of works in this area. At a geometric level, both settings choose a point in a simplex. The differences are semantic, informational, and objective-based: in portioning, the fractional vector is the implemented allocation itself and agents typically report richer information (e.g., ideal budget proposals, cardinal allocations, or ordinal rankings), with work focusing on truthfulness, proportionality, Pareto efficiency, or project fairness of aggregation rules (Freeman et al., 2021; Freeman & Schmidt-Kraepelin, 2024; Caragiannis et al., 2024; Elkind et al., 2026). In contrast, our simplex point is a lottery over alternatives, acceptability is evaluated via expected utility against agent-specific thresholds, and we focus on the query complexity of finding any unanimously acceptable lottery or certifying infeasibility, rather than selecting a normatively optimal aggregate allocation.

# B. Omitted Content from Section 2

## B.1. Justification for Expected Utility

**Expected utility and acceptability thresholds.** Given we are analyzing the output as *lotteries* over alternatives, we need a principled way to evaluate randomization. The standard modeling choice is the von Neumann-Morgenstern expected utility framework: under mild rationality axioms on preferences over lotteries, an agent's lottery preferences admit a cardinal utility representation that is *linear in probabilities*, i.e., the agent behaves as if maximizing expected utility (von Neumann & Morgenstern, 1944). This assumption is very common in algorithmic decision-making. For instance, Markov decision processes and reinforcement learning typically define the objective as maximizing *expected* cumulative reward/return under a (possibly stochastic) policy (Puterman, 1994; Sutton & Barto, 2018). In game theory, agents evaluate mixed strategies via expected utilities, and equilibrium notions (best responses, Nash equilibria) are defined in terms of expected payoffs (Nash, 1950; Osborne & Rubinstein, 1994).

We further adopt an agent-specific acceptability threshold $\tau_i$ and declare a lottery $\mathbf{x}$ acceptable to agent $i$ if and only if $U_i(\mathbf{x}) \geq \tau_i$. Thresholds capture a common "minimum standard" view of decision-making (Simon, 1955), and they match the way safety, compliance, and stakeholder constraints are modeled in modern ML/AI: one often maximizes expected performance subject to inequality constraints (e.g., constrained MDPs/safe or constrained RL), rather than folding all desiderata into a single scalar reward (Altman, 1999; García & Fernández, 2015). In multi-agent mechanism design, analogous inequalities appear as *individual rationality* (or participation) constraints requiring each agent's expected utility to exceed an outside option (Myerson, 1981; Nisan et al., 2007). In our setting, $\tau_i$ plays exactly this role of a personalized minimum bar, so unanimity corresponds to satisfying all agents' constraints without requiring interpersonal utility comparisons.

## B.2. Necessity of finite precision

We will explain why a finite-precision assumption is necessary to obtain finite query algorithms for *exact* unanimity. In the main body of the paper, we enforce this by assuming each $u_i(s_j)$ and $\tau_i$ is an integer multiple of $\varepsilon$, which implies that the relevant boundary/threshold locations are rationals with bounded denominators.

**Proposition B.1** (Necessity of finite precision). *If agent utilities and thresholds are allowed to be arbitrary real numbers in* $[0, 1]$ *(with no finite-precision restriction), then there is no algorithm that, using finitely many membership queries, always outputs a unanimously acceptable lottery when one exists (and correctly outputs* NULL *otherwise). This holds even for*

$n = m = 2$.

*Proof.* Consider $m = 2$ alternatives $S = \{s_1, s_2\}$. Any lottery can be identified with a single scalar $\alpha \in [0, 1]$, meaning $\mathbf{x}_\alpha = (\alpha, 1 - \alpha)$. For each $t \in (0, 1)$, define an instance $\mathcal{I}_t$ with two agents as follows:

- Agent 1 has $u_1(s_1) = 1$, $u_1(s_2) = 0$, and threshold $\tau_1 = t$. Then $U_1(\mathbf{x}_\alpha) = \alpha$, so agent 1 accepts exactly the interval $[t, 1]$.

- Agent 2 has $u_2(s_1) = 0$, $u_2(s_2) = 1$, and threshold $\tau_2 = 1 - t$. Then $U_2(\mathbf{x}_\alpha) = 1 - \alpha$, so agent 2 accepts exactly the interval $[0, t]$.

Hence, the unanimously acceptable set in $\mathcal{I}_t$ is exactly the singleton $\{\mathbf{x}_t\}$.

Now consider any (deterministic) algorithm $\mathcal{A}$ that is correct on every instance and that makes finitely many queries on every instance. Run $\mathcal{A}$ on the family $\{\mathcal{I}_t : t \in (0, 1)\}$. Each membership query made by $\mathcal{A}$ returns a single bit, so every execution produces a finite transcript over $\{0, 1\}$; therefore, across all $t$, the set of possible transcripts is countable, and consequently the set of possible outputs of $\mathcal{A}$ is also countable. However, correctness requires that on input $\mathcal{I}_t$, the algorithm outputs *exactly* $\mathbf{x}_t$ (it cannot output NULL, since $\mathcal{I}_t$ is feasible), and the map $t \mapsto x_t$ is injective over the uncountable set $(0, 1)$. This is impossible if $\mathcal{A}$ has only countably many possible outputs. Therefore no such finite-query algorithm exists.

The same impossibility immediately applies to any randomized algorithm that is required to be correct with probability 1: fixing the internal randomness gives us a deterministic algorithm, to which the above argument applies. $\qquad\square$

### B.3. Finite precision as $\varepsilon$-robust unanimity

Although our main model treats the $\varepsilon$-precision values as the ground truth, it is often useful to interpret the same assumption as capturing limited measurement/communication precision: agents may only be able to report utilities and thresholds up to granularity $\varepsilon$. Fix any underlying "continuous" instance with real-valued parameters $\bar{u}_i(s_j) \in [0, 1]$ and $\bar{\tau}_i \in (0, 1]$. Let $u_i(s_j)$ and $\tau_i$ be $\varepsilon$-quantized versions satisfying

$$\max_{j \in [m]} |u_i(s_j) - \bar{u}_i(s_j)| \leq \varepsilon \quad \text{and} \quad |\tau_i - \bar{\tau}_i| \leq \varepsilon \quad \text{for all } i \in N.$$

Write $\bar{U}_i(\mathbf{x}) = \sum_{j=1}^m x_j \bar{u}_i(s_j)$ and $U_i(\mathbf{x}) = \sum_j x_j u_i(s_j)$. Then, we have the following result.

**Lemma B.2.** *For any agent $i$ and any lottery $x \in \Delta(S)$, $|U_i(\mathbf{x}) - \bar{U}_i(\mathbf{x})| \leq \varepsilon$. Consequently, for any lottery $x \in \Delta(S)$:*

$$\left( \forall i \in N, \ \bar{U}_i(\mathbf{x}) \geq \bar{\tau}_i + 2\varepsilon \right) \implies \left( \forall i \in N, \ U_i(\mathbf{x}) \geq \tau_i \right) \implies \left( \forall i \in N, \ \bar{U}_i(\mathbf{x}) \geq \bar{\tau}_i - 2\varepsilon \right).$$

*Proof.* Since $x$ is a probability vector,

$$|U_i(\mathbf{x}) - \bar{U}_i(\mathbf{x})| = \left| \sum_{j=1}^m x_j \big( u_i(s_j) - \bar{u}_i(s_j) \big) \right| \leq \sum_{j=1}^m x_j |u_i(s_j) - \bar{u}_i(s_j)| \leq \varepsilon.$$

For the forward implication, if $\bar{U}_i(\mathbf{x}) \geq \bar{\tau}_i + 2\varepsilon$ then $U_i(\mathbf{x}) \geq \bar{U}_i(\mathbf{x}) - \varepsilon \geq \bar{\tau}_i + \varepsilon \geq \tau_i$. For the backward implication, if $U_i(\mathbf{x}) \geq \tau_i$ then $\bar{U}_i(\mathbf{x}) \geq U_i(\mathbf{x}) - \varepsilon \geq \tau_i - \varepsilon \geq \bar{\tau}_i - 2\varepsilon$. $\qquad\square$

**Interpretation.** Lemma B.2 shows that $\varepsilon$-precision on the agent parameters induces an $\Theta(\varepsilon)$-robust notion of unanimity: it rules out instances where feasibility relies on knife-edge comparisons that are unstable under perturbations of size $\mathcal{O}(\varepsilon)$ in the reported utilities/thresholds. In particular, if there exists a lottery that clears every agent's true threshold by a margin $2\varepsilon$, then it remains unanimously acceptable after $\varepsilon$-quantization; and any unanimously acceptable lottery in the quantized instance is within $2\varepsilon$ of being acceptable in the underlying continuous instance.

**Additional remarks on the $\varepsilon$-precision model.** Our algorithmic upper bounds in Section 3 do not rely on any further relationship between $m$ and $\varepsilon$. For some information-theoretic lower bounds, we assume $m \leq (1/\varepsilon)^{1-\delta}$ for a fixed constant $\delta \in (0, 1]$ (in particular, $m \leq 1/\varepsilon$), which avoids a degenerate setting where the positive $\varepsilon$-grid on the simplex is tiny and the resulting counting bounds become vacuous. Unless otherwise stated, we work throughout in the $\varepsilon$-precision model above. In particular, all constraints have rational coefficients, and whenever $\bigcap_{i \in N} \mathcal{A}_i \neq \varnothing$ it contains a lottery with rational coordinates. Thus, we may restrict all oracle queries and algorithm outputs to rational lotteries; for readability we write $\mathbf{x} \in \Delta(S)$ and suppress the qualifier $\mathbf{x} \in \Delta(S) \cap \mathbb{Q}^m$.

## B.4. Beyond Unanimity: Computational Intractability of Partial Acceptance Objectives

Our main goal in this paper is feasibility under a conservative consent constraint: either find a lottery that every agent deems acceptable, or certify that no such lottery exists. A natural relaxation is to ask for a lottery that is acceptable to *as many agents as possible*. Formally, define the (acceptability) welfare of a lottery $\mathbf{x}$ as $W(\mathbf{x}) := |\{i \in N : U_i(\mathbf{x}) \geq \tau_i\}|$. Then, the optimization problem (denote as MAX-ACCEPT-LOTTERY) would ask: find a lottery $\mathbf{x} \in \Delta(S)$ maximizing $W(\mathbf{x})$; whereas the decision variant (denote as $k$-ACCEPT-LOTTERY) would ask: given an integer $k \in \mathbb{Z}_+$, decide whether there exists $\mathbf{x} \in \Delta(S)$ such that $W(\mathbf{x}) \geq k$.

At a high level, $k$-ACCEPT-LOTTERY resembles the *maximum feasible subsystem* problem (MAX-FS), which asks to satisfy as many linear inequalities as possible and is NP-hard (Amaldi & Kann, 1995). Indeed, writing each agent's acceptability condition as a linear constraint $\langle \mathbf{u}_i, \mathbf{x} \rangle \geq \tau_i$ and adding the simplex constraints $x_1, \ldots, x_m \geq 0$ and $\sum_{j=1}^m x_j = 1$, $k$-ACCEPT-LOTTERY can be seen as a structured subclass of MAX-FS. Thus, MAX-FS hardness results do not immediately apply under our strong structural restrictions (nonnegative coefficients, a shared normalization constraint, etc.). The following theorem shows that the problem remains NP-hard even in this special case.

**Theorem B.3.** $k$-ACCEPT-LOTTERY *is NP-hard, even when* $u_i(s_j) \in \{0, 1\}$ *for every* $i \in N$ *and* $j \in [m]$.

*Proof.* We reduce from the classical NP-complete problem CLIQUE. An instance of CLIQUE is given by an undirected graph $G = (V, E)$ and an integer $\kappa \geq 2$; it is a yes-instance if $G$ contains a clique of size $\kappa \geq 2$, and a no-instance otherwise.

Given an instance $(G, \kappa)$ of CLIQUE with $|V| = m$ and $|E| = M$, we construct an instance of $k$-ACCEPT-LOTTERY with $m$ alternative and a polynomial number of agents.

Create one alternative per vertex, i.e., $S = \{s_v : v \in V\}$, where $|S| = m$.

Let $R := m + M + 1$. Note that $R$ is polynomial in the size of $G$.

Define three agent types; all utilities are in $\{0, 1\}$:

- **Type A: "Upper bound" agents.** For each vertex $v \in V$, create $R$ identical agents $a_{v,1}, \ldots, a_{v,R}$. For every agent $i \in N_A$, let
$$u_i(s_v) = 0 \text{ and } u_i(s_w) = 1 \text{ for all } w \neq v,$$
and threshold $\tau_i = 1 - 1/\kappa$. Then, for any lottery $\mathbf{x}$, $U_i(\mathbf{x}) = \sum_{w \neq v} x_w = 1 - x_v$, and thus
$$U_i(\mathbf{x}) \geq \tau_i \iff 1 - x_v \geq 1 - 1/\kappa \iff x_v \leq 1/\kappa.$$

  Intuitively, every Type A agent created from vertex $v$ only accepts a lottery if the probability mass placed on $v$ is at most $1/\kappa$.

- **Type B: "Lower bound" agents.** For each vertex $v \in V$, create one agent. For every agent $i \in N_B$, let
$$u_i(s_v) = 1 \text{ and } u_i(s_w) = 0 \text{ for all } w \neq v,$$
and threshold $\tau_i = 1/\kappa$. Then, for any lottery $\mathbf{x}$, $U_i(\mathbf{x}) = x_v$, and thus
$$U_i(\mathbf{x}) \geq \tau_i \iff x_v \geq 1/\kappa.$$

  Intuitively, every Type B agent created from vertex $v$ only accepts a lottery if the probability mass placed on $v$ is at least $1/\kappa$.

- **Type C: Edge agents.** For each edge $\{u, v\} \in E$, create one agent. For every agent $i \in N_C$ created from edge $\{u, v\}$, let

$$u_i(s_u) = u_i(s_v) = 1 \text{ and } u_i(s_w) = 0 \text{ for all } w \notin \{u, v\},$$

and threshold $\tau_i = 2/\kappa$. Then, for any lottery $\mathbf{x}$, $U_i(\mathbf{x}) = x_u + x_v$, and thus

$$U_i(\mathbf{x}) \geq \tau_i \iff x_u + x_v \geq 2/\kappa.$$

Now, the total number of agents is $n = |N_A| + |N_B| + |N_C| = Rm + m + M$, which is polynomial in $m + M$.

Now, set $k := Rm + \kappa + \binom{\kappa}{2}$. We claim that $(G, \kappa)$ is a yes-instance of CLIQUE if and only if the constructed instance has a lottery $\mathbf{x} \in \Delta(S)$ with $W(\mathbf{x}) \geq k$.

($\Rightarrow$) Assume $G$ contains a clique $C \subseteq V$ with $|C| = \kappa$. Define the lottery $\mathbf{x}$ as follows:

$$x_v = \begin{cases} 1/\kappa & \text{if } v \in C, \\ 0 & \text{otherwise.} \end{cases}$$

Now, all Type A agents will accept $\mathbf{x}$: For every vertex $v$, we have $x_v \leq 1/\kappa$, so each of the $R$ agents corresponding to each vertex $v$ will accept $\mathbf{x}$. Hence, we have $Rm$ accepting agents from Type A.

Exactly $\kappa$ Type B agents will accept $\mathbf{x}$: For $v \in C$, $x_v = 1/\kappa$, so any agent corresponding to a vertex in $C$ will accept; whereas for $v \notin C$, $x_v = 0$, so any agent corresponding to a vertex not in $C$ will reject. Thus, there is exactly $\kappa$ accepting agents from Type B.

Exactly $\binom{\kappa}{2}$ Type C agents will accept $\mathbf{x}$: For any edge $\{u, v\}$ with $u, v \in C$, we have $x_u + x_v = 2/\kappa$, so an agent corresponding to the edge $\{u, v\}$ will accept. Since $C$ is a clique, it has all $\binom{\kappa}{2}$ edges, so at least $\binom{\kappa}{2}$ Type C agents that accept.

Thus,

$$W(\mathbf{x}) \geq Rm + \kappa + \binom{\kappa}{2} = k.$$

($\Leftarrow$) Assume there exists a lottery $\mathbf{x} \in \Delta(S)$ such that $W(\mathbf{x}) \geq k$. We prove three claims.

**Claim 1: $x_v \leq 1/\kappa$ for every vertex $v \in V$.** Suppose for a contradiction that for some vertex $v$, $x_v > 1/\kappa$. Then every one of the $R$ Type $A$ agents associated with $v$ rejects $\mathbf{x}$ (since they accept $\mathbf{x}$ if and only if $x_v \leq 1/\kappa$). Thus, we lost at least $R$ acceptances from Type A agents. Even if all other agents (from Type B and Type C) accepted, then

$$\begin{aligned} W(\mathbf{x}) &\leq R(m-1) + m + M \\ &= Rm - R + m + M \\ &= Rm - (m + M + 1) + m + M \\ &= Rm - 1 \\ &< Rm + \kappa + \binom{\kappa}{2} = k, \end{aligned}$$

giving us a contradiction. Thus, $x_v \leq 1/\kappa$ for all $v \in V$.

**Claim 2: At most $\kappa$ Type B agents accept, and their corresponding vertices have probability mass exactly $1/\kappa$.** Let $T := \{v \in V : x_v \geq 1/\kappa\}$ be the set of vertices whose Type $B$ agents accept $\mathbf{x}$. Now, we know from Claim 1 that $x_v \leq 1/\kappa$ and together with $x_v \geq 1/\kappa$ imply that $x_v = 1/\kappa$ for all $v \in T$. Then,

$$1 = \sum_{v \in V} x_v \geq \sum_{v \in T} x_v = |T| \cdot \frac{1}{\kappa},$$

giving us $|T| \leq \kappa$. However, since $W(\mathbf{x}) \geq k$ and by Claim 1, all $Rm$ Type A agents accept, thus, among Types B and C agents combined, we must have at least

$$W(\mathbf{x}) - Rm = k - Rm = \kappa + \binom{\kappa}{2}$$

agents that accept $\mathbf{x}$.

**Claim 3: A Type C agent with corresponding edge $\{u, v\}$ can accept x only if both $u, v \in T$.** Note that in order for such an agent to accept $\mathbf{x}$, we must have that $x_u + x_v \geq 2/\kappa$, by definition. Then, since $x_u \leq 1/\kappa$ and $x_v \leq 1/\kappa$, we must necessarily have that $x_u = x_v = 1/\kappa$. Thus, $u, v \in T$.

Therefore, the number of Type C agents that accept $\mathbf{x}$ is exactly the number of edges of $G$ with both endpoints in $T$. For any undirected simple graph with $|T|$ vertices, the number of edges is at most $\binom{|T|}{2}$. Thus, the total number of agents which are Types B or C that accept $\mathbf{x}$ is at most $|T| + \binom{|T|}{2}$. However, we established earlier that we need at least $\kappa + \binom{\kappa}{2}$ agents among Types B and C to accept $\mathbf{x}$, so

$$|T| + \binom{|T|}{2} \geq \kappa + \binom{\kappa}{2}. \tag{1}$$

Now, let

$$f(y) := y + \binom{y}{2} = y + \frac{y(y-1)}{2} = \frac{y(y+1)}{2}.$$

Then, for any integer $y \geq 0$, it is clear that $f(y)$ is strictly increasing. Then, since $|T| \leq \kappa$ (as established in the proof of Claim 2), we have that $f(|T|) \leq f(\kappa)$. Combining this with (1) which gives us $f(\kappa) \leq f(|T|)$, we get that

$$f(\kappa) \leq f(|T|) \leq f(\kappa),$$

essentially giving us

$$f(\kappa) = f(|T|) \implies \kappa + \binom{\kappa}{2} = |T| + \binom{|T|}{2}. \tag{2}$$

By the strictly increasing property of $f$, we get that $|T| = \kappa$. Combining this with (2), we get that $\binom{|T|}{2} = \binom{\kappa}{2}$. Since $|T| = \kappa$, exactly $\kappa$ Type B agents accept $\mathbf{x}$ (namely, those corresponding to vertices in $T$). Moreover, because $W(\mathbf{x}) \geq k$ and all $Rm$ Type A agents accept by Claim 1, we need at least $\kappa + \binom{\kappa}{2}$ acceptances coming from Types B and C combined. Hence, at least $\binom{\kappa}{2}$ Type C agents must accept $\mathbf{x}$.

On the other hand, by Claim 3, a Type C agent can accept $\mathbf{x}$ only if its corresponding edge has both endpoints in $T$. There is one Type C agent per edge, and among $|T| = \kappa$ vertices there are at most $\binom{|T|}{2} = \binom{\kappa}{2}$ such edges. Therefore, at most $\binom{\kappa}{2}$ Type C agents can accept $\mathbf{x}$. Combining the two bounds, we conclude that exactly $\binom{\kappa}{2}$ Type C agents accept $\mathbf{x}$, which implies that every unordered pair of distinct vertices in $T$ forms an edge in $G$. Thus, the induced subgraph on $T$ is complete, i.e., $T$ is a clique of size $\kappa$ in $G$. $\qquad\square$

This hardness result complements our earlier conceptual motivation for unanimity. For unanimity, once each agent's acceptability halfspace is learned, the remaining task is simply to test feasibility (and select a witness) via linear programming, so the central bottleneck is elicitation via queries. In contrast, moving beyond unanimity to objectives based on maximizing $W(\mathbf{x})$ introduces computational intractability even when the full instance $(\mathbf{u}_i, \tau_i)_{i \in N}$ is given explicitly. For this reason, we treat unanimity as a clean baseline for conservative, safety-oriented collective decision making, and we leave approximation or structure-exploiting relaxations beyond unanimity to future work.

## C. Omitted Proofs from Section 3

### C.1. ExactThreshold **Subroutine**

We first establish some preliminaries. Fix an agent $i \in N$ and indices $k, k' \in [m]$ such that $\mathrm{Query}(i, \mathbf{e}_k) = \texttt{False}$ and $\mathrm{Query}(i, \mathbf{e}_{k'}) = \texttt{True}$. Recall the edge lotteries

$$\mathbf{x}_{k,k',\alpha} := \alpha \mathbf{e}_{k'} + (1 - \alpha)\mathbf{e}_k, \qquad \alpha \in [0, 1].$$

Along this edge,

$$U_i(\mathbf{x}_{k,k',\alpha}) = (1 - \alpha)u_i(s_k) + \alpha u_i(s_{k'}) = u_i(s_k) + \alpha(u_i(s_{k'}) - u_i(s_k)).$$

Since $\mathrm{Query}(i, \mathbf{e}_k) = \texttt{False}$ implies $u_i(s_k) < \tau_i$ and $\mathrm{Query}(i, \mathbf{e}_{k'}) = \texttt{True}$ implies $\tau_i \leq u_i(s_{k'})$, we have that $u_i(s_{k'}) - u_i(s_k) > 0$, so $U_i(x_{k,k',\alpha})$ is strictly increasing in $\alpha$. Thus, there is a unique

$$\alpha^* := \inf\{\alpha \in [0,1] : \mathrm{Query}(i, \mathbf{x}_{k,k',\alpha}) = \texttt{True}\} \in (0,1]$$

such that $\mathrm{Query}(i, \mathbf{x}_{k,k',\alpha}) = \texttt{True}$ if and only if $\alpha \geq \alpha^*$.

Solving $U_i(x_{k,k',\alpha^*}) = \tau_i$ gives

$$\alpha^* = \frac{\tau_i - u_i(s_k)}{u_i(s_{k'}) - u_i(s_k)}. \tag{3}$$

Under $\varepsilon$-precision, let $u_i(s_k) = a\varepsilon$, $u_i(s_{k'}) = b\varepsilon$, and $\tau_i = t\varepsilon$ for integers $a, b, t$. Then $b - a \in \{1, 2, \ldots, 1/\varepsilon\}$ and $t - a \in \{1, 2, \ldots, b - a\}$, so (3) equals $(t-a)/(b-a)$ and therefore can be written in lowest terms as $p/q$ with

$$1 \leq q \leq b - a \leq 1/\varepsilon. \tag{4}$$

Let $Q := 1/\varepsilon \in \mathbb{Z}_{>0}$. Any two distinct rationals $p/q \neq p'/q'$ with $1 \leq q, q' \leq Q$ satisfy

$$\left| \frac{p}{q} - \frac{p'}{q'} \right| = \frac{|pq' - p'q|}{qq'} \geq \frac{1}{qq'} \geq \frac{1}{Q^2} = \varepsilon^2.$$

Consequently, any interval of length $< \varepsilon^2/2$ contains *at most one* rational number whose reduced denominator is at most $Q$.

We now introduce the ExactThreshold subroutine.

ExactThreshold$(i, k, k')$    The subroutine uses membership queries only in the bisection phase.

1. Initialize $\ell \leftarrow 0$ and $u \leftarrow 1$. Note that $\mathrm{Query}(i, \mathbf{x}_{k,k',\ell}) = \mathrm{Query}(i, e_k) = \texttt{False}$ and $\mathrm{Query}(i, \mathbf{x}_{k,k',u}) = \mathrm{Query}(i, e_{k'}) = \texttt{True}$.

2. While $u - \ell \geq \varepsilon^2/2$:
   (a) Let $mid \leftarrow (\ell + u)/2$ and query $\mathrm{Query}(i, \mathbf{x}_{k,k',mid})$.
   (b) If the answer is $\texttt{True}$, set $u \leftarrow mid$; otherwise set $\ell \leftarrow mid$.

   Throughout the loop, the invariant $\mathrm{Query}(i, \mathbf{x}_{k,k',\ell}) = \texttt{False}$ and $\mathrm{Query}(i, \mathbf{x}_{k,k',u}) = \texttt{True}$ is maintained, and therefore $\ell < \alpha^* \leq u$.

3. Let $Q := 1/\varepsilon$. By (4), $\alpha^*$ has reduced denominator at most $Q$, and by the loop condition we have $u - \ell < \varepsilon^2/2$. Therefore $\alpha^*$ is the *unique* rational in $[\ell, u]$ with reduced denominator at most $Q$. Recover this unique rational and return it as $\alpha^*$. One concrete reconstruction routine (using no membership queries) is: iterate $q = 1, 2, \ldots, Q$, set $p := \lceil q\ell \rceil$, and return the first fraction $p/q$ satisfying $p/q \leq u$. Uniqueness implies the returned value equals $\alpha^*$.

We now prove correctness. Note that the monotonicity of $\alpha \mapsto \mathrm{Query}(i, \mathbf{x}_{k,k',\alpha})$ implies the bisection loop maintains $\ell < \alpha^* \leq u$. On termination, $u - \ell < \varepsilon^2/2$, so by the uniqueness argument above there is exactly one rational in $[\ell, u]$ with reduced denominator $\leq Q = 1/\varepsilon$, namely $\alpha^*$ itself. Hence the reconstruction step returns $\alpha^*$ exactly.

Finally, we prove the query complexity. Starting from an interval of length 1, each bisection query halves the bracket width. Thus the loop performs at most $\lceil \log(2/\varepsilon^2) \rceil = \mathcal{O}(\log(1/\varepsilon))$ queries, and the reconstruction phase uses none.

We conclude with a short remark.

**Why does $\varepsilon^2$ appear in our turning point searches?**    Under $\varepsilon$-precision, each utility $u_i(s_j)$ and threshold $\tau_i$ is an integer multiple of $\varepsilon$. Along any simplex edge between two pure lotteries, an agent's acceptance boundary occurs at a *turning point* $\alpha^*$ that is a rational number with reduced denominator at most $1/\varepsilon$. Any two distinct rationals with denominators at most $1/\varepsilon$ differ by at least $\varepsilon^2$. Consequently, once we bracket $\alpha^*$ into an interval of length $< \varepsilon^2/2$ using membership queries, $\alpha^*$ becomes the unique candidate rational in the bracket and can be recovered exactly via rational reconstruction. This is the origin of the $\mathcal{O}(\log(1/\varepsilon))$ query cost per one-dimensional threshold search.

## C.2. Geometric Intuition for LearnHyperplane

Recovering an agent's halfspace on the simplex relies on the convexity of linear acceptance regions. Along any edge connecting a rejected vertex $\mathbf{e}_k$ to an accepted vertex $\mathbf{e}_{k'}$, there exists exactly one "turning point" where the agent's utility crosses their threshold $\tau_i$. By finding these turning points on $m-1$ edges incident to a specific vertex, we identify $m-1$ points that lie strictly on the hyperplane boundary $\{\mathbf{x} : \langle \mathbf{u}_i, \mathbf{x} \rangle = \tau_i\}$. These points uniquely define the hyperplane (and thus the normal vector $\mathbf{c}_i$) in the $(m-1)$-dimensional affine hull of the simplex. Algorithm 1 implements this efficiently using binary search to locate these turning points.

## C.3. Proof of Lemma 3.1

The algorithm begins by querying each pure lottery $\mathbf{e}_j$ for $j \in [m]$, making exactly $m$ queries.

Each call to ExactThreshold performs bisection for $\mathcal{O}(\log(1/\varepsilon))$ steps (to obtain an interval of width $< \varepsilon^2/2$), followed by rational reconstruction (which uses no queries). Since Algorithm 1 calls ExactThreshold at most $m-1$ times, the total number of queries is $m + \mathcal{O}((m-1)\log(1/\varepsilon)) = \mathcal{O}(m\log(1/\varepsilon))$.

## C.4. Proof of Lemma 3.2

Fix an agent $i \in N$.

Algorithm 1 first queries all pure lotteries and partitions the alternatives as

$$L^{\mathrm{acc}} := \{j \in [m] : U_i(\mathbf{e}_j) \geq \tau_i\} = \{j \in [m] : u_i(s_j) \geq \tau_i\}, \quad L^{\mathrm{rej}} := [m] \setminus L^{\mathrm{acc}} = \{j \in [m] : u_i(s_j) < \tau_i\}.$$

If $L^{\mathrm{rej}} = \varnothing$ (i.e., Algorithm 1 returns ACCEPTALL), then $u_i(s_j) \geq \tau_i$ for all $j \in [m]$, and for every $\mathbf{x} \in \Delta(S)$,

$$U_i(\mathbf{x}) = \sum_{j=1}^{m} x_j u_i(s_j) \geq \sum_{j=1}^{m} x_j \tau_i = \tau_i,$$

and agent $i$ accepts every lottery, as claimed.

If $L^{\mathrm{acc}} = \varnothing$ (i.e., Algorithm 1 returns REJECTALL), then $u_i(s_j) < \tau_i$ for all $j \in [m]$, and for every $\mathbf{x} \in \Delta(S)$,

$$U_i(\mathbf{x}) = \sum_{j=1}^{m} x_j u_i(s_j) < \sum_{j=1}^{m} x_j \tau_i = \tau_i,$$

and agent $i$ rejects every lottery, as claimed.

In the remainder of this proof, we assume that $L^{\mathrm{acc}} \neq \varnothing$ and $L^{\mathrm{rej}} \neq \varnothing$. Then, Algorithm 1 chooses some $r \in L^{\mathrm{rej}}$, so $u_i(s_r) < \tau_i$.

For distinct $k, k' \in [m]$ and $\alpha \in [0,1]$, recall the definition $x_{k,k',\alpha} := \alpha e_{k'} + (1-\alpha)e_k$. In particular, for $j \in L^{\mathrm{acc}}$,

$$U_i(x_{r,j,\alpha}) = \alpha u_i(s_j) + (1-\alpha)u_i(s_r) = u_i(s_r) + \alpha(u_i(s_j) - u_i(s_r)).$$

Since $u_i(s_j) \geq \tau_i > u_i(s_r)$, we have $u_i(s_j) - u_i(s_r) > 0$, so $U_i(x_{r,j,\alpha})$ is strictly increasing in $\alpha$ and there is a unique $\alpha_{r,j} \in (0,1]$ such that $U_i(x_{r,j,\alpha_{r,j}}) = \tau_i$. Solving the equality gives

$$\alpha_{r,j}(u_i(s_j) - u_i(s_r)) = \tau_i - u_i(s_r) \implies \frac{1}{\alpha_{r,j}} = \frac{u_i(s_j) - u_i(s_r)}{\tau_i - u_i(s_r)}. \tag{5}$$

Algorithm 1 sets $c_{i,r} = 0$ and, for every $j \in L^{\mathrm{acc}}$, sets $c_{i,j} = 1/\alpha_{r,j}$.

We split our analysis into two cases.

**Case 1.** Suppose $\alpha_{r,j} = 1$ for all $j \in L^{\mathrm{acc}}$ (Line 10 of Algorithm 1). Then by (5), we have $u_i(s_j) = \tau_i$ for all $j \in L^{\mathrm{acc}}$. Since $u_i(s_k) < \tau_i$ for all $k \in L^{\mathrm{rej}}$, a lottery $\mathbf{x} \in \Delta(S)$ satisfies $U_i(\mathbf{x}) \geq \tau_i$ if and only if it assigns zero probability to rejected alternatives: indeed,

$$U_i(\mathbf{x}) = \sum_{j \in L^{\mathrm{acc}}} x_j \tau_i + \sum_{k \in L^{\mathrm{rej}}} x_k u_i(s_k) \leq \sum_{j \in L^{\mathrm{acc}}} x_j \tau_i + \sum_{k \in L^{\mathrm{rej}}} x_k \tau_i = \tau_i,$$

with strict inequality whenever $x_k > 0$ for some $k \in L^{\text{rej}}$ (because then $u_i(s_k) < \tau_i$ contributes strictly less). If $x_k = 0$ for all $k \in L^{\text{rej}}$, then $U_i(\mathbf{x}) = \sum_{j \in L^{\text{acc}}} x_j \tau_i = \tau_i$. Thus the acceptable set is $\{\mathbf{x} \in \Delta(S) : x_k = 0 \text{ for all } k \in L^{\text{rej}}\}$.

In this case, Algorithm 1 returns the vector $\mathbf{c}_i$ defined by $c_{i,j} = 1$ for $j \in L^{\text{acc}}$ and $c_{i,k} = 0$ for $k \in L^{\text{rej}}$. Since $\sum_{j \in L^{\text{acc}}} x_j \leq \sum_{j=1}^{m} x_j = 1$, this implies $\sum_{j \in L^{\text{acc}}} x_j = 1$, and thus $\sum_{k \in L^{\text{rej}}} x_k = 0$, so $x_k = 0$ for all $k \in L^{\text{rej}}$ For $\mathbf{x} \in \Delta(S)$, we have $\langle \mathbf{c}_i, \mathbf{x} \rangle = \sum_{j \in L^{\text{acc}}} x_j$, so

$$\langle \mathbf{c}_i, \mathbf{x} \rangle \geq 1 \iff \sum_{j \in L^{\text{acc}}} x_j \geq 1 \iff x_k = 0 \text{ for all } k \in L^{\text{rej}},$$

which is exactly the acceptable set.

**Case 2.** There exists some $a \in L^{\text{acc}}$ with $\alpha_{r,a} < 1$ (Line 14), which implies $u_i(s_a) > \tau_i$ by (5). For each $k \in L^{\text{rej}} \setminus \{r\}$, Algorithm 1 computes the unique $\alpha_{k,a} \in (0, 1)$ satisfying $U_i(x_{k,a,\alpha_{k,a}}) = \tau_i$, i.e.,

$$\tau_i = \alpha_{k,a} u_i(s_a) + (1 - \alpha_{k,a}) u_i(s_k),$$

which is unique because $U_i(\mathbf{x}_{k,a,0}) = u_i(s_k) < \tau_i$ and $U_i(\mathbf{x}_{k,a,1}) = u_i(s_a) > \tau_i$. Algorithm 1 sets $c_{i,a} = 1/\alpha_{r,a}$ and then defines for each $k \in L^{\text{rej}} \setminus \{r\}$:

$$c_{i,k} := \frac{1 - \alpha_{k,a} c_{i,a}}{1 - \alpha_{k,a}}.$$

We now show that in this case,

$$c_{i,j} = \frac{u_i(s_j) - u_i(s_r)}{\tau_i - u_i(s_r)} \quad \text{for all } j \in [m]. \tag{6}$$

For $j \in L^{\text{acc}}$, (6) holds by (5), since Algorithm sets $c_{i,j} = 1/\alpha_{r,j}$. Also $c_{i,r} = 0 = (u_i(s_r) - u_i(s_r))/(\tau_i - u_i(s_r))$, so (6) holds for $j = r$.

Fix $k \in L^{\text{rej}} \setminus \{r\}$. From $\tau_i = \alpha_{k,a} u_i(s_a) + (1 - \alpha_{k,a}) u_i(s_k)$, we have

$$u_i(s_k) = \frac{\tau_i - \alpha_{k,a} u_i(s_a)}{1 - \alpha_{k,a}},$$

and hence

$$\frac{u_i(s_k) - u_i(s_r)}{\tau_i - u_i(s_r)} = \frac{1}{1 - \alpha_{k,a}} \cdot \frac{\tau_i - u_i(s_r) - \alpha_{k,a}(u_i(s_a) - u_i(s_r))}{\tau_i - u_i(s_r)} = \frac{1 - \alpha_{k,a} \cdot \frac{u_i(s_a) - u_i(s_r)}{\tau_i - u_i(s_r)}}{1 - \alpha_{k,a}}.$$

But by (5) applied to $a$, we have $c_{i,a} = 1/\alpha_{r,a} = (u_i(s_a) - u_i(s_r))/(\tau_i - u_i(s_r))$, so the last expression equals

$$\frac{1 - \alpha_{k,a} c_{i,a}}{1 - \alpha_{k,a}} = c_{i,k},$$

proving (6) for $k$ and thus for all $j \in [m]$.

Finally, for any lottery $\mathbf{x} \in \Delta(S)$, using (6) we have

$$\langle \mathbf{c}_i, \mathbf{x} \rangle \geq 1 \iff \sum_{j=1}^{m} \frac{u_i(s_j) - u_i(s_r)}{\tau_i - u_i(s_r)} x_j \geq 1 \iff \sum_{j=1}^{m} (u_i(s_j) - u_i(s_r)) x_j \geq \tau_i - u_i(s_r).$$

Since $\sum_{j=1}^{m} x_j = 1$, the left-hand side equals $\sum_{j=1}^{m} u_i(s_j) x_j - u_i(s_r) \sum_{j=1}^{m} x_j = U_i(\mathbf{x}) - u_i(s_r)$, so the inequality is equivalent to $U_i(\mathbf{x}) - u_i(s_r) \geq \tau_i - u_i(s_r)$, i.e., $U_i(\mathbf{x}) \geq \tau_i$.

This proves that whenever Algorithm 1 returns a vector $\mathbf{c}_i$, it satisfies $\langle \mathbf{c}_i, \mathbf{x} \rangle \geq 1 \iff U_i(\mathbf{x}) \geq \tau_i$ for all $\mathbf{x} \in \Delta(S)$, as desired.

## C.5. Proof of Theorem 3.3

We first prove **correctness**. Let $F(C) := \{\mathbf{x} \in \Delta(S) : \langle \mathbf{c}, \mathbf{x} \rangle \geq 1 \text{ for all } \mathbf{c} \in C\}$ be the feasible set determined by the constraints in $C$. By definition, $\text{Select}(C)$ returns NULL if and only if $F(C) = \varnothing$, and otherwise returns a unique lottery $\mathbf{x} \in F(C)$.

Also, by Lemma 3.2, if $\mathbf{c}_i$ is the vector returned by LearnHyperplane (in the non-REJECTALL case), then for every $\mathbf{x} \in \Delta(S)$,

$$\langle \mathbf{c}_i, \mathbf{x} \rangle \geq 1 \iff U_i(\mathbf{x}) \geq \tau_i. \tag{7}$$

Now, at every iteration of the **while** loop (Line 3), immediately after the assignment $\mathbf{x} \leftarrow \text{Select}(C)$ (Line 4), the following invariant holds: For every agent $i \in N \setminus N'$, $C$ contains the constraint $\mathbf{c}_i$ learned for agent $i$, and the current candidate $\mathbf{x}$ satisfies $U_i(\mathbf{x}) \geq \tau_i$.

To see this, observe that if $i \in N \setminus N'$, then $i$ was previously selected as a violator, and removed from $N'$. In that iteration, either

- LearnHyperplane$(i) = $ REJECTALL, in which case the algorithm terminates immediately (so the invariant is only relevant before termination), or

- LearnHyperplane$(i) = \mathbf{c}_i$, then the algorithm adds the returned vector $\mathbf{c}_i$ to $C$.

In later iterations, $\mathbf{x} \leftarrow \text{Select}(C) \in F(C)$, so in particular, $\langle \mathbf{c}_i, \mathbf{x} \rangle \geq 1$. By (7), this implies $U_i(\mathbf{x}) \geq \tau_i$. Note that LearnHyperplane$(i) \neq$ ACCEPTALL, because we only call LearnHyperplane$(i)$ after observing Query$(i, \mathbf{x}) = $ False. If an agent accepted all pure lotteries, linearity would imply they accept all lotteries, so they could not be a violator.

Next, we prove that if Algorithm 2 returns a lottery, it is unanimously acceptable. Suppose the algorithm returns some $\mathbf{x}$ (Line 14). This occurs only if there is no violator, which means that for every unlearned agent $i \in N'$, we observed Query$(i, \mathbf{x}) = $ True, i.e., $U_i(\mathbf{x}) \geq \tau_i$. For any learned agent $i \in N \setminus N'$, the invariant implies $U_i(\mathbf{x}) \geq \tau_i$. Consequently, we get that $U_i(\mathbf{x}) \geq \tau_i$ for all $i \in N$, and $\mathbf{x}$ is unanimously acceptable.

Next, we prove that if Algorithm 2 returns NULL, then no unanimously acceptable lottery exists. There are two ways the algorithm returns NULL:

1. $\text{Select}(C)$ returns NULL (Line 5). Then $F(C) = \varnothing$. Suppose for a contradiction that there exists a lottery $\mathbf{x}^* \in \Delta(S)$ such that $U_i(\mathbf{x}^*) \geq \tau_i$ for all $i \in N$. In particular, for every learned agent $i \in N \setminus N'$, we have $U_i(\mathbf{x}^*) \geq \tau_i$. By (7), this implies $\langle \mathbf{c}_i, \mathbf{x}^* \rangle \geq 1$ for each $\mathbf{c}_i \in C$. Thus, $\mathbf{x}^* \in F(C)$, contradicting $F(C) = \varnothing$. Thus, no unanimously acceptable lottery exists.

2. LearnHyperplane$(i)$ returns REJECTALL (Line 17). By definition of LearnHyperplane, this means that agent $i$ rejects every pure lottery. By linearity of expected utility, if $U_i(\mathbf{e}_j) < \tau_i$ for all $j \in [m]$, then for every $\mathbf{x} \in \Delta(S)$,

$$U_i(\mathbf{x}) = \sum_{j=1}^m x_j U_i(\mathbf{e}_j) < \sum_{j=1}^m x_j \tau_i = \tau_i,$$

and agent $i$ rejects every lottery. Thus, unanimity is impossible and returning NULL is correct.

Finally, we show that Algorithm 2 terminates. In each iteration of the **while** loop, either the algorithm returns, or it finds a violator $i \in N'$, and then removes $i$ from $N'$. Since $N' \subseteq N$ and agents are removed at most once, this can happen at most $n$ times. Thus, the algorithm terminates after at most $n + 1$ iterations.

Next, we prove the **query complexity** by providing an upper bound on the number of queries made by Algorithm 2. Let $\gamma$ be the number of times the algorithm calls LearnHyperplane. Clearly $\gamma \leq n$. By Lemma 3.1, each call to LearnHyperplane makes $\mathcal{O}(m \log(1/\varepsilon))$ queries. Therefore, all invocations of LearnHyperplane together contribute $\mathcal{O}(nm \log(1/\varepsilon))$ queries.

Next, we count the number of "verification" queries. In an iteration where the unlearned set has size $|N'| = k$, the **for** loop queries each agent in $N'$ at most once, so it makes at most $k$ direct queries. Each time a violator is found and learned, $|N'|$

decreases by 1. Thus, in the worst case (violator being last each time), the total number of "verification" queries is at most

$$n + (n-1) + \cdots + 1 = \frac{n(n+1)}{2} = \mathcal{O}(n^2).$$

Thus, the total number of queries made by Algorithm 2 is $\mathcal{O}\left(n^2 + nm \log(1/\varepsilon)\right)$.

### C.6. Proof of Theorem 3.4

Note that we can assume $n \geq 2$ and $m \geq 2$, otherwise the problem is trivial. For integer weights $w_1, \ldots, w_n \geq 1$, let $N_{\mathrm{multi}}(w)$ denote the multiset containing exactly $w_i$ labeled copies of each agent $i$, and so $|N_{\mathrm{multi}}(w)| = \sum_{i=1}^n w_i$.

Let WeightedSample$(w, r')$ return a uniformly random subset $N_{\mathrm{rand}} \subseteq N_{\mathrm{multi}}(w)$ of size $r'$ (i.e., $|N_{\mathrm{rand}}| = r'$) drawn *without replacement*. Write $\mathrm{supp}(N_{\mathrm{rand}}) \subseteq N$ for the set of distinct agents appearing in $N_{\mathrm{rand}}$.

Our algorithm (Algorithm 3) is as follows.

For each agent $i$, let $\mathcal{A}_i \subseteq \Delta(S)$ denote the agent's acceptable region (halfspace). By Lemma 3.2, either $\mathcal{A}_i = \Delta(S)$ (ACCEPTALL), or $\mathcal{A}_i = \varnothing$ (REJECTALL), or $\mathcal{A}_i = \{\mathbf{x} \in \Delta(S) : \langle \mathbf{c}_i, \mathbf{x} \rangle \geq 1\}$ for the vector $\mathbf{c}_i$ returned by LearnHyperplane$(i)$.

Then, for any subset of agents $N' \subseteq N$, define the feasible region

$$P(N') : \bigcap_{i \in N'} \mathcal{A}_i.$$

If $P(N') \neq \varnothing$, define $\mathbf{x}_{N'}$ to be the unique lexicographically maximum point in $P(N')$ (i.e., the output of Select under the deterministic lexicographic tie-breaking rule from Section 3); otherwise define $x_{N'} := \text{NULL}$. When no agent in $N'$ is of type REJECTALL (i.e., when $\mathcal{A}_i \neq \varnothing$ for all $i \in N'$), this coincides with $\mathbf{x}_{N'} = \text{Select}(\{\mathbf{c}_i : i \in N', \mathcal{A}_i \neq \Delta(S)\})$, since ACCEPTALL agents contribute no constraint.

Define the *basis* for a subset of agents $N' \subseteq N$ (where $P(N') \neq \varnothing$) as a minimal subset $B \subseteq N'$ such that $P(B) \neq \varnothing$ and $\mathbf{x}_B = \mathbf{x}_{N'}$. For weight vector $w = (w_1, \ldots, w_n) \in \mathbb{Z}_{\geq 1}^n$ and $N' \subseteq N$, write $w(N') := \sum_{i \in N'} w_i$ and $W := w(N)$.

We begin by proving **correctness**. If Algorithm 3 returns

- $\mathbf{x}$ (Line 21), then every agent accepts $\mathbf{x}$ (i.e., the set of violating agents $V$ is empty), so $\mathbf{x}$ is unanimously acceptable.

- NULL, then either:
    - some call to LearnHyperplane returns REJECTALL (Line 10), in which case that agent rejects every lottery in $\Delta(S)$ and finding a unanimously acceptable lottery is impossible; or
    - Select$(C_{N_{\mathrm{rand}}})$ returns NULL (Line 18), meaning $P(\mathrm{supp}(N_{\mathrm{rand}})) = \varnothing$. Since $P(N) \subseteq P(\mathrm{supp}(N_{\mathrm{rand}}))$, emptiness of $P(\mathrm{supp}(N_{\mathrm{rand}}))$ implies $P(N) = \varnothing$ (note that $P(N') = \cap_{i \in N'} \mathcal{A}_i$ with $\mathcal{A}_i \subseteq \Delta(S)$), i.e., no unanimously acceptable lottery exists. Hence returning NULL is correct.

Next, we will prove the **expected query complexity** using a series of intermediate lemmas.

We first make use of a lemma from Matoušek et al. (1996, Lemma 1(iii)), which highlights a standard property of $d$-dimensional linear programming: under lexicographic (deterministic) selection, every feasible bounded instance has a basis of size at most $d$. Since our feasible region lies in the affine hull of dimension $d = m - 1$, the claim follows (note that $P(N') \subseteq \Delta(S)$ is always bounded).

**Lemma C.1.** *For any subset of agents $N' \subseteq N$, if $P(N') \neq \varnothing$, then $N'$ has a basis $B \subseteq N'$ with $|B| \leq m - 1$.*

This lemma essentially tells us that there exists a "small" set of agents whose constraints determine the final chosen lottery $\mathbf{x}$. This "smallness" is what gives us the $\mathcal{O}(m \log n)$ dependence later.

Next, we prove the following weighted sampling lemma. Intuitively, in each iteration, Algorithm 3 sample $r'$ elements from the multiset $N_{\mathrm{multi}}(w)$ (given a weight vector $w = (w_1, \ldots, w_n)$) and solves the LP for the sampled distinct agents

---

**Algorithm 3** Randomized algorithm that returns a unanimously acceptable lottery if one exists, or NULL otherwise

---

**Input:** Set of agents $N$, and set of alternatives $S = \{s_1, \ldots, s_m\}$
**Output:** Lottery $\mathbf{x} = (x_1, \ldots, x_m) \in \Delta(S)$ such that $U_i(\mathbf{x}) \geq \tau_i$ for all $i \in N$; or NULL
 1: $r \leftarrow 16(m-1)^2$                                               $\triangleright$ sample size
 2: Initialize $w_i \leftarrow 1$, $known_i \leftarrow$ False, and $\mathbf{c}_i \leftarrow$ None for all $i \in N$
 3: **while** True **do**
 4:      $W \leftarrow \sum_{i=1}^{n} w_i$
 5:      $r' \leftarrow \min\{r, W\}$                           $\triangleright$ if $W < r$ we sample the whole multiset
 6:      $N_{\mathrm{rand}} \leftarrow$ WeightedSample$(w, r')$            $\triangleright$ $N_{\mathrm{rand}} \subseteq N_{\mathrm{multi}}(w)$, $|N_{\mathrm{rand}}| = r'$
 7:      **for all** $i \in \mathrm{supp}(N_{\mathrm{rand}})$ **do**
 8:          **if** $known_i =$ False **then**
 9:              $\mathbf{c} \leftarrow$ LearnHyperplane$(i)$
10:              **if** $\mathbf{c} =$ REJECTALL **then return** NULL
11:              **else if** $c =$ ACCEPTALL **then** $known_i \leftarrow$ True and $\mathbf{c}_i \leftarrow$ None
12:              **else** $known_i \leftarrow$ True and $\mathbf{c}_i \leftarrow \mathbf{c}$
13:              **end if**
14:          **end if**
15:      **end for**
16:      $C_{N_{\mathrm{rand}}} \leftarrow \{\mathbf{c}_i : i \in \mathrm{supp}(N_{\mathrm{rand}}) \wedge \mathbf{c}_i \neq$ None$\}$
17:      $\mathbf{x} \leftarrow$ Select$(C_{N_{\mathrm{rand}}})$
18:      **if** $\mathbf{x} =$ NULL **then return** NULL
19:      **end if**
20:      Compute the set $V := \{i \in N : \mathrm{Query}(i, \mathbf{x}) =$ False$\}$       $\triangleright$ identifying violating agents
21:      **if** $V = \varnothing$ **then return** $\mathbf{x}$
22:      **end if**
23:      **for all** $i \in V$ **do**
24:          $w_i \leftarrow 2w_i$                          $\triangleright$ multiplicative weights update on violators
25:      **end for**
26: **end while**

---

$\mathrm{supp}(N_{\mathrm{rand}})$ constraints (if an agent $i$ has not been "learned" yet, then run LearnHyperplane$(i)$ and store $\mathbf{c}_i$), getting a lottery $\mathbf{x}_{\mathrm{supp}(N_{\mathrm{rand}})}$, which is the unique lexicographically maximum feasible lottery for the sampled constraints (i.e., the output of Select on the sampled constraint set). The algorithm then verifies this against every agent by querying every agent on this candidate $\mathbf{x}_{\mathrm{supp}(N_{\mathrm{rand}})}$. Let $V$ be the agents who reject $\mathbf{x}_{\mathrm{supp}(N_{\mathrm{rand}})}$. This lemma gives us an upper bound on the expected *total weight of the violators* $w(V)$ after solving the sampled "subproblem" in each iteration. In particular, we will show that the expected total weight of violators for any iteration is $\mathbb{E}[w(V)] \leq m \cdot \frac{W}{r'+1}$.

Now, we can think of these weights as a sampling distribution (i.e., agent $i$ is more likely to be included in the next sampled subproblem/iteration when $w_i^{(t)}$ is large), and as a penalty counter (if agent $i$ rejects our proposed $\mathbf{x}$, we double $w_i^{(t)}$ for the next iteration). Thus, if the expected total weight of violators is small, then doubling of violator's weight would flesh them out faster, and provide us some progress towards identifying "more constraining" agents/likely violators faster.

**Lemma C.2.** *Let* $V := \varnothing$ *if* $\mathbf{x}_{\mathrm{supp}(N_{\mathrm{rand}})} =$ NULL*; and* $V := \{i \in N : \mathbf{x}_{\mathrm{supp}(N_{\mathrm{rand}})} \notin \mathcal{A}_i\}$ *otherwise. Then*

$$\mathbb{E}[w(V)] \leq m \cdot \frac{W}{r'+1}.$$

*In particular, if* $r' = \min\{16(m-1)^2, W\}$*, then if* $W \leq r$*, then* $r' = W$ *and the sample contains all agents, so* $w(V) = 0$*. If* $W > r$*, then*

$$\mathbb{E}[w(V)] \leq \frac{W}{8(m-1)}.$$

*Proof.* Each element $i \in N_{\mathrm{multi}}(w)$ is a copy of some agent $\widehat{i} \in N$.

For any $N_{\text{rand}} \subseteq N_{\text{multi}}(w)$, define $\mathbf{x}_{N_{\text{rand}}} := \mathbf{x}_{\text{supp}(N_{\text{rand}})}$ and $\pi : N_{\text{multi}}(w) \to N$. Then, define the *violator copy set*

$$V_{\text{copy}}(N_{\text{rand}}) := \{i \in N_{\text{multi}}(w) \setminus N_{\text{rand}} : \mathbf{x}_{N_{\text{rand}}} \neq \text{NULL} \text{ and } \mathbf{x}_{N_{\text{rand}}} \notin \mathcal{A}_{\pi(i)}\}.$$

Intuitively, agents that have been sampled would have had their constraints learned and thus would not be a violator. Thus, it makes sense that violators only make up the agents that have not been learned (and in particular not sampled in this iteration).

If $\mathbf{x}_{N_{\text{rand}}} = \text{NULL}$ then $V_{\text{copy}}(N_{\text{rand}}) = \varnothing$ by definition. If $\mathbf{x}_{N_{\text{rand}}} \neq \text{NULL}$ and an agent $i$ violates $\mathbf{x}_{N_{\text{rand}}}$, then no copy of $i$ can be in $N_{\text{rand}}$ (otherwise $i \in \text{supp}(N_{\text{rand}})$ and $\mathbf{x}_{N_{\text{rand}}} \in \mathcal{A}_i$). Thus, all $w_i$ copies of $i$ is contained in $N_{\text{multi}}(w) \setminus N_{\text{rand}}$ and also $V_{\text{copy}}(N_{\text{rand}})$. This means that

$$|V_{\text{copy}}(N_{\text{rand}})| = w(V), \tag{8}$$

i.e., every violating agent contributes all its copies, and no violating agent can appear in $N_{\text{rand}}$.

For any subset $Q \subseteq N_{\text{multi}}(w)$, with $|Q| = (r' + 1)$, define its *extreme elements*

$$X(Q) := \{i \in Q : i \in V_{\text{copy}}(Q \setminus \{i\})\}.$$

Then, we will show that the number of extreme elements is no more than $m$:

$$|X(Q)| \leq m. \tag{9}$$

We split our analysis into two cases. We first provide an intuition of the proof for the two cases. When $P(\text{supp}(Q)) \neq \varnothing$, then an extreme element corresponds to an agent $\widehat{i}$ such that removing $\widehat{i}$ changes the unique optimum and causes $\widehat{i}$ to be violated. We prove that such an $\widehat{i}$ must lie in every basis of $\text{supp}(Q)$. Since there exists some basis of size $\leq m - 1$, there can be at most $m - 1$ such agents. Also, $X(Q)$ cannot contain two copies of the same agent (because then removing one copy would not remove that agent from the support). Thus, $|X(Q)| \leq m - 1$.

In the case where $P(\text{supp}(Q)) = \varnothing$, then an extreme point corresponds to an agent $\widehat{i}$ such that removing $i$ makes the subproblem feasible (since $Q \setminus \{i\}$ is feasible by definition of a violator). If removing $\widehat{i}$ restores feasibility (i.e., makes the intersection nonempty), then any infeasible subcollection must include $i$ (because every subset of a feasible family is feasible). Then we apply Helly's theorem in dimension $m - 1$: since the family on $\text{supp}(N_{\text{rand}})$ is infeasible, there exists an infeasible subfamily of size $\leq m$; taking a minimal one gives a "minimal infeasible" set $B$ with $|B| \leq m$. Every "essential" agent $\widehat{i}$ (whose removal restores feasibility) must be in all minimal infeasible sets, in particular this one $B$. Thus, there are at most $m$ such agents. Again there is at most one copy per agent in $X(Q)$, so $|X(Q)| \leq m$.

We prove it formally as follows.

**Case 1:** $P(\text{supp}(Q)) \neq \varnothing$ **(feasible).** Let $i \in X(Q)$ (and $\widehat{i}$ be the corresponding agent in $N$). Then $Q \setminus \{i\}$ is such that $\mathbf{x}_{\text{supp}(Q) \setminus \{\widehat{i}\}} = \mathbf{x}_{Q \setminus \{i\}}$ exists and violates $\widehat{i}$. We claim $\widehat{i}$ must belong to every basis of $\text{supp}(Q)$. Indeed, suppose (for contradiction) there is a basis $B \subseteq \text{supp}(Q)$ with $\widehat{i} \notin B$. Then $B \subseteq \text{supp}(Q) \setminus \{\widehat{i}\}$ and $\mathbf{x}_B = \mathbf{x}_{\text{supp}(Q)}$. However, $\mathbf{x}_{\text{supp}(Q) \setminus \{\widehat{i}\}}$ is feasible for $B$ and (since $\mathbf{x}_{\text{supp}(Q) \setminus \{\widehat{i}\}}$ violates $\widehat{i}$) we have $\mathbf{x}_{\text{supp}(Q) \setminus \{\widehat{i}\}} \neq \mathbf{x}_{\text{supp}(Q)}$. Since $\mathbf{x}_{\text{supp}(Q)}$ is feasible for $P(\text{supp}(Q) \setminus \{b_i\})$ and $\mathbf{x}_{\text{supp}(Q) \setminus \{b_i\}}$ is the lexicographically maximum point in $P(\text{supp}(Q) \setminus \{b_i\})$, we have that $\mathbf{x}_{\text{supp}(Q) \setminus \{b_i\}}$ is lexicographically at least $\mathbf{x}_{\text{supp}(Q)}$. Moreover, $\mathbf{x}_{\text{supp}(Q) \setminus \{b_i\}} \neq \mathbf{x}_{\text{supp}(Q)}$ because $\mathbf{x}_{\text{supp}(Q) \setminus \{b_i\}}$ violates $b_i$ while $\mathbf{x}_{\text{supp}(Q)}$ satisfies $b_i$. Hence $\mathbf{x}_{\text{supp}(Q) \setminus \{b_i\}}$ is lexicographically larger than $\mathbf{x}_{\text{supp}(Q)}$. This contradicts the fact that $\mathbf{x}_B = \mathbf{x}_{\text{supp}(Q)}$ is the lexicographically maximum point in $P(B)$, because $\mathbf{x}_{\text{supp}(Q) \setminus \{b_i\}}$ is feasible for $B$. Hence $\widehat{i}$ is in every basis of $\text{supp}(Q)$. By Lemma C.1, $\text{supp}(Q)$ has some basis of size at most $m - 1$, so there can be at most $m - 1$ such distinct agents $\widehat{i}$. Moreover, $X(Q)$ contains at most one copy per agent (if $Q$ contained two copies of $\widehat{i}$, removing one would still leave $\widehat{i} \in \text{supp}(Q \setminus \{i\})$, so $i$ could not be a violator). Thus $|X(Q)| \leq m - 1$ in this case.

**Case 2:** $P(\text{supp}(Q)) = \varnothing$ **(infeasible).** Let $i \in X(Q)$ (and $\widehat{i}$ be the corresponding agent in $N$). Then by definition $Q \setminus \{i\}$ is feasible, hence $P(\text{supp}(Q) \setminus \{\widehat{i}\}) \neq \varnothing$. Therefore, every infeasible subset of $\text{supp}(Q)$ must contain $\widehat{i}$ (otherwise it would remain infeasible inside $\text{supp}(Q) \setminus \{\widehat{i}\}$), so $\widehat{i}$ belongs to every minimal infeasible subset of $\text{supp}(Q)$. By Helly's theorem (Danzer et al., 1963), in the $(m - 1)$-dimensional affine hull, we can choose a minimal infeasible subset $B \subseteq \text{supp}(Q)$ with $|B| \leq m$. Since every $\widehat{i}$ with $P(\text{supp}(Q) \setminus \{\widehat{i}\}) \neq \varnothing$ must lie in every minimal infeasible

subset, we have $\widehat{i} \in B$. Thus there are at most $m$ such distinct agents, and again at most one copy per agent can lie in $X(Q)$. Hence $|X(Q)| \leq m$.

This proves (9). Finally, we count pairs $(N_{\mathrm{rand}}, i)$ such that $|N_{\mathrm{rand}}| = r'$ and $i \in V_{\mathrm{copy}}(N_{\mathrm{rand}})$.

On one hand, the count equals $\sum_{N_{\mathrm{rand}}:|N_{\mathrm{rand}}|=r'} |V_{\mathrm{copy}}(N_{\mathrm{rand}})|$. On the other hand, mapping $(N_{\mathrm{rand}}, i) \mapsto Q := N_{\mathrm{rand}} \cup \{i\}$ gives a bijection to pairs $(Q, i)$ with $|Q| = r' + 1$ and $i \in X(Q)$. Therefore,

$$\sum_{N_{\mathrm{rand}}:|N_{\mathrm{rand}}|=r'} |V_{\mathrm{copy}}(N_{\mathrm{rand}})| = \sum_{Q:|Q|=r'+1} |X(Q)| \leq m \binom{W}{r'+1},$$

where the rightmost inequality follows from the fact that there are $\binom{W}{r'+1}$ possible such subsets $Q$.

Then, dividing both sides by $\binom{W}{r'}$ and using the fact that $\binom{W}{r'+1}/\binom{W}{r'} = (W - r')/(r' + 1)$, we get

$$\mathbb{E}[|V_{\mathrm{copy}}(N_{\mathrm{rand}})|] \leq m \cdot \frac{W - r'}{r' + 1}.$$

Finally, from (8), we know that $|V_{\mathrm{copy}}(N_{\mathrm{rand}})| = w(V)$. Substituting this into the inequality above gives us

$$\mathbb{E}[w(V)] \leq m \cdot \frac{W - r'}{r' + 1} \leq m \cdot \frac{W}{r' + 1},$$

as desired. Then, since Algorithm 3 uses a sample size of $r = 16(m - 1)^2$, for $m \geq 2$, we get that

$$m \cdot \frac{W}{r' + 1} \leq m \cdot \frac{W}{16(m - 1)^2 + 1} \leq \frac{W}{8(m - 1)},$$

as desired. $\qquad\square$

The remainder of the proof can be broken down into four steps:

1. Construct a small "witness set" $B^*$ of agents of size $\leq m$ that can "explain" feasibility or infeasibility.

2. Formalize the random process so we can reason about expectations cleanly.

3. Define a potential function $\Phi_t$ and prove it has positive expected drift until termination, thus giving us $\mathbb{E}[T] = \mathcal{O}(m \log n)$, where $T$ is the number of iterations of the **while** loop.

4. Convert the iteration bound into the final expected query bound.

**Step 1: Constructing the witness set $B^*$.**

Next, we construct the *witness set* $B^* \subseteq N$. Intuitively, this is a set of agents of size at most $m$ that "explains" feasibility (or infeasibility). Intuitively in feasible instances, agents in $B^*$ should be sufficient to "pin down" the final (unanimously acceptable lottery) $\mathbf{x}_N$; or in infeasible instances, it already certifies infeasibility.

Fix a set $B^* \subseteq N$ as follows:

- If $P(N) \neq \varnothing$ (feasible instance), let $B^*$ be any basis of $N$; by Lemma C.1, $|B^*| \leq m - 1$.

- If $P(N) = \varnothing$ (infeasible instance), by Helly's theorem (Danzer et al., 1963) there exists a $B^* \subseteq N$ with $|B^*| \leq m$ such that $P(B^*) = \varnothing$.

Let $b := |B^*| \leq m$. Now, algorithmically, each iteration of the **while** loop (Line 3) samples some constraints from the subset of agents $N' \subseteq N$ and computes $\mathbf{x}_{N'}$ (Line 6). If $\mathbf{x}_{N'}$ is not unanimously acceptable, then we can define a violator set $V(N') := \{i \in N : \mathbf{x}_{N'} \notin \mathcal{A}_i\}$. Then the following lemma proves that if there is a violation, at least one violator must lie in $B^*$.

This lemma is the key that links our weight doubling update to progress (which we will see with the defined potential function later).

**Lemma C.3.** *For any $N' \subseteq N$ with $\mathbf{x}_{N'} \neq \text{NULL}$, if $V(N') \neq \varnothing$ then $V(N') \cap B^* \neq \varnothing$.*

*Proof.* If $P(N) = \varnothing$ (and consequently $P(B^*) = \varnothing$), then for any lottery $\mathbf{x} \in \Delta(S)$, there exists an agent $i \in B^*$ with $\mathbf{x} \notin \mathcal{A}_i$, so in particular $V(N') \cap B^* \neq \varnothing$.

Thus, we assume that $P(N) \neq \varnothing$ and $B^*$ is a basis of $N$. Let $\mathbf{x}^* := \mathbf{x}_N = \mathbf{x}_{B^*}$ be the unique tie-broken optimum for the full instance. Suppose for contradiction that $\mathbf{x}_{N'}$ violates some agent, but none of those violators are in $B^*$, i.e., $V(N') \cap B^* = \varnothing$ but $V(N') \neq \varnothing$. This means that $\mathbf{x}_{N'}$ satisfies every constraint in $B^*$, i.e., $\mathbf{x}_{N'} \in P(B^*)$.

Since $\mathbf{x}^*$ is the unique lexicographically maximum point in $P(B^*)$ and $\mathbf{x}_{N'} \in P(B^*)$, it follows that $\mathbf{x}_{N'}$ is not lexicographically larger than $\mathbf{x}^*$. On the other hand, $\mathbf{x}^*$ is feasible for $P(N')$ (because it satisfies all constraints in $N$ and $N' \subseteq N$), and $\mathbf{x}_{N'}$ is the lexicographically maximum point in $P(N')$, so $\mathbf{x}_{N'}$ is lexicographically at least $\mathbf{x}^*$. Therefore, $\mathbf{x}_{N'} = \mathbf{x}^*$. However, $\mathbf{x}^*$ is feasible for all agents, so it cannot have violators, i.e., $V(N') \neq \varnothing$, which is a contradiction. $\qquad\square$

Thus, any violating candidate lottery must violate at least one agent in the witness set $B^*$.

**Step 2: Defining the random process.**

Note that all randomness in Algorithm 3 comes from the calls to WeightedSample. Let $T$ be the (random) index of the iteration in which the algorithm terminates (i.e., returns $\mathbf{x}$ or NULL). For $t \geq 1$, let:

- $w^{(t)} = (w_1^{(t)}, \dots, w_n^{(t)}) \in \mathbb{Z}_{\geq 1}^n$ be the weight vector at the *start* of iteration $t$, and $W^{(t)} := \sum_{i=1}^n w_i^{(t)}$.

- $r^{(t)} := \min\{r, W^{(t)}\}$ and $N_{\text{rand}}^{(t)} \subseteq N_{\text{multi}}(w^{(t)})$ be the random sample of size $r^{(t)}$.

- Let $\mathbf{x}_t := \mathbf{x}_{\text{supp}(N_{\text{rand}}^{(t)})}$. If $\mathbf{x}_t = \text{NULL}$, then the algorithm terminates at iteration $t$. Otherwise let $V^{(t)} := V(\text{supp}(N_{\text{rand}}^{(t)})) = \{i \in N : \mathbf{x}_t \notin \mathcal{A}_i\}$ and define the *violator weight* (i.e., total weight of violators at iteration $t$) as

$$\omega^{(t)} := \begin{cases} \sum_{i \in V_t} w_i^{(t)} & \text{if } \mathbf{x}_t \neq \text{NULL}, \\ 0 & \text{if } \mathbf{x}_t = \text{NULL}. \end{cases}$$

  If $t < T$, then necessarily $\mathbf{x}_t \neq \text{NULL}$ and $V^{(t)} \neq \varnothing$.

Define the filtration $\{\mathcal{F}_t\}_{t \geq 0}$ by letting $\mathcal{F}_0$ be trivial and $\mathcal{F}_t := \sigma(N_{\text{rand}}^{(1)}, \dots, N_{\text{rand}}^{(t)})$ for each $t \geq 1$.[12] Intuitively, $\mathcal{F}_t$ is the history of samples $N_{\text{rand}}^{(1)}, \dots, N_{\text{rand}}^{(t)}$ up to time $t$. Later on, when we condition on $\mathcal{F}_{t-1}$, it means we freeze everything that happened in rounds $1, \dots, t-1$, and (the random sample in) round $t$ is the only remaining randomness.

Let $(N_{\text{rand}}^{(1)}, \dots, N_{\text{rand}}^{(t)})$ be the sample history. Then, conditioned on $(N_{\text{rand}}^{(1)}, \dots, N_{\text{rand}}^{(t)})$, the algorithm's state (and in particular $w^{(t+1)}$ and $W^{(t+1)}$) is fixed, because all other steps are deterministic. To see this, given the sample history up to $t$, we know (i) which agents were sampled, (ii) which constraints were learned, (iii) which $\mathbf{x}_t$ was computed (since Select is deterministic under the fixed lexicographic tie-breaking rule) and (iv) which agents are violators, and thus their new weight vector.

Furthermore, $T$ is a *stopping time* with respect to this filtration (because whether the algorithm has terminated by the end of iteration $t$ is determined by $(N_{\text{rand}}^{(1)}, \dots, N_{\text{rand}}^{(t)})$). Intuitively, this means that we can tell whether we have stopped by time $t$ using only information revealed up to time $t$; we don't need future samples. We require this because later on, we will have condition expectation terms multiplied by an indicator for whether the algorithm has stopped. In order to correctly pull that indicator outside the conditional expectation, we need this fact (or in other words, this indicator must be measurable with respect to the $\sigma$-field we condition on).

**Step 3: A drift bound for the potential function and $\mathbb{E}[T]$.**

The key idea is that we want to upper bound $\mathbb{E}[T]$, the expected number of iterations. To do so, we will define a potential $\Phi_t \in [0, 1]$ for round $t$, show that before termination, $\log \Phi_t$ increases by at least a constant in expectation each round, and since $\log \Phi_t \leq 0$ always, and we start from a negative value, we can only have about $\mathcal{O}(m \log n)$ rounds.

---

[12] Here, $\sigma(\cdot)$ denotes $\sigma$-algebra generated by the input.

Fix the witness set $B^* \subseteq N$ constructed in Step 1, and recall that $b := |B^*| \le m$. Define the potential at the start of iteration $t$ by

$$\Phi_t := \frac{\prod_{i \in B^*} w_i^{(t)}}{(W^{(t)})^b}.$$

Intuitively, the numerator $\prod_{i \in B^*} w_i^{(t)}$ tracks how big the witnesses weights are (multiplicatively); whereas the denominator $(W^{(t)})^b$ normalizes by the overall scale of weights (since all weights might grow). Raising $W$ to the power of $b$ ensures the potential is always at most 1. Thus, note that $0 < \Phi_t \le 1$ for all $t$, and initially $\Phi_1 = n^{-b}$ since $w_i^{(1)} = 1$ and $W^{(1)} = n$.

For $t < T$, the iteration is nonterminal, so $\mathbf{x}_t \ne$ NULL and the violator set $V^{(t)} := \{i \in N : \mathbf{x}_t \notin \mathcal{A}_i\}$ is nonempty. Recall that the total violator weight is $\omega^{(t)} := \sum_{i \in V^{(t)}} w_i^{(t)}$.

**Lemma C.4.** *Assume $m \ge 2$ and Algorithm 3 uses $r = 16(m-1)^2$. There exists an absolute constant $c_0 > 0$ (e.g., $c_0 = \log 2 - \frac{1}{4}$) such that for every $t \ge 1$,*

$$\mathbb{E}\left[(\log \Phi_{t+1} - \log \Phi_t)\, \mathbf{1}[t < T]\, |\, \mathcal{F}_{t-1}\right] \ge c_0 \cdot \mathbf{1}[t < T].$$

*Proof.* On the event $\{t < T\}$ we have $V^{(t)} \ne \varnothing$ and $x_t \ne$ NULL. By Lemma C.3, $V^{(t)} \cap B^* \ne \varnothing$, so at least one weight in $B^*$ doubles in iteration $t$. Hence $\prod_{i \in B^*} w_i^{(t+1)} \ge 2 \prod_{i \in B^*} w_i^{(t)}$. Also the total weight updates as $W^{(t+1)} = W^{(t)} + \omega^{(t)}$. Therefore, on $\{t < T\}$,

$$\Phi_{t+1} \ge \Phi_t \cdot \frac{2}{\left(1 + \omega^{(t)}/W^{(t)}\right)^b},$$

and so

$$\log \Phi_{t+1} - \log \Phi_t \ge \log 2 - b \log\left(1 + \frac{\omega^{(t)}}{W^{(t)}}\right).$$

Condition on $\mathcal{F}_{t-1}$, so $w^{(t)}$ and $W^{(t)}$ are fixed. Lemma C.2 (with $r = 16(m-1)^2$ and $r^{(t)} = \min\{r, W^{(t)}\}$) gives $\mathbb{E}[\omega^{(t)}\,|\,\mathcal{F}_{t-1}] \le W^{(t)}/(8(m-1))$ for $m \ge 2$. Since $\log(1+u)$ is concave,

$$\mathbb{E}\left[\log\left(1 + \frac{\omega^{(t)}}{W^{(t)}}\right)\Big|\mathcal{F}_{t-1}\right] \le \log\left(1 + \frac{\mathbb{E}[\omega^{(t)}\,|\,\mathcal{F}_{t-1}]}{W^{(t)}}\right) \le \log\left(1 + \frac{1}{8(m-1)}\right).$$

Using $b \le m$ and $\log(1+u) \le u$,

$$b \log\left(1 + \frac{1}{8(m-1)}\right) \le m \cdot \frac{1}{8(m-1)} \le \frac{1}{4}.$$

Thus

$$\mathbb{E}\left[(\log \Phi_{t+1} - \log \Phi_t)\mathbf{1}[t < T]|\mathcal{F}_{t-1}\right] \ge \left(\log 2 - \frac{1}{4}\right)\mathbf{1}[t < T],$$

so we may take $c_0 = \log 2 - \frac{1}{4} > 0$. $\square$

**Lemma C.5.** $\mathbb{E}[T] = \mathcal{O}(m \log n)$.

*Proof.* Fix $K \ge 1$ and let $T_K := \min\{T, K\}$. Telescoping gives

$$\log \Phi_{T_K} - \log \Phi_1 = \sum_{t=1}^{K-1} (\log \Phi_{t+1} - \log \Phi_t)\mathbf{1}[t < T].$$

Take expectations and apply the drift lemma:

$$\mathbb{E}[\log \Phi_{T_K}] - \log \Phi_1 \ge c_0\, \mathbb{E}[\min\{T - 1, K - 1\}].$$

Since $\Phi_{T_K} \le 1$, we have $\mathbb{E}[\log \Phi_{T_K}] \le 0$. Therefore

$$\mathbb{E}[\min\{T - 1, K - 1\}] \le \frac{\log(1/\Phi_1)}{c_0}.$$

Letting $K \to \infty$ and using $\Phi_1 = n^{-b}$ gives

$$\mathbb{E}[T] \leq 1 + \frac{b \log n}{c_0} = \mathcal{O}(m \log n),$$

since $b \leq m$ and $c_0$ is an absolute constant. $\qquad\qquad\qquad\qquad\qquad\qquad\qquad\qquad\qquad\square$

**Step 4: Expected number of** LearnHyperplane **calls and query complexity.**

Finally, we prove the expected number of LearnHyperplane calls and conclude with the query complexity of Algorithm 3.

Each iteration samples $r^{(t)} \leq r$ copies, hence at most $r$ *distinct* agents appear in $\mathrm{supp}(N_{\mathrm{rand}}^{(t)})$. Since each agent's hyperplane is learned at most once, the total number of distinct LearnHyperplane calls is at most $L \leq \min\{n, rT\}$. Since the function $y \mapsto \min\{n, ry\}$ is concave, taking expectations and using Jensen's inequality, we get $\mathbb{E}[L] \leq \min\{n, r\mathbb{E}[T]\} = \mathcal{O}(\min\{n, m^3 \log n\})$, where the equality is using Lemma C.5 and $r = \Theta((m-1)^2)$.

Finally, in any iteration that reaches Line 20, the verification step queries all $n$ agents once; thus each iteration uses at most $n$ verification queries, and the total verification cost is at most $nT$ queries. Each call to LearnHyperplane makes $\mathcal{O}(m \log(1/\varepsilon))$ queries, by Lemma 3.1. Thus the total expected query complexity is

$$\mathcal{O}\left(n\mathbb{E}[T]\right) + \mathcal{O}\left(m \log(1/\varepsilon) \cdot \mathbb{E}[L]\right) = \mathcal{O}(nm \log n + \min\{n, m^3 \log n\} \cdot m \log(1/\varepsilon)),$$

as desired.

### C.7. Remark on finding infeasibility witness set

When our algorithms output NULL, they can also output an explicit infeasibility witness: a subset of agents $B \subseteq N$ such that $\bigcap_{i \in B} \mathcal{A}_i = \varnothing$. This can be done without any additional membership queries.

If LearnHyperplane($i$) returns REJECTALL, we may take $B = \{i\}$. Otherwise, NULL is returned only when Select is infeasible on a set of already learned halfspaces; since these constraints are explicit, Helly's theorem (Helly, 1923) implies there exists an infeasible subcollection of size at most $m$ in the $(m-1)$-dimensional affine hull. Such a witness subset can be extracted offline (e.g., via a standard LP routine) and returned alongside NULL.

## D. Omitted Proofs from Section 4

### D.1. Proof of Theorem 4.1

We prove the lower bound for deterministic algorithms first, and then extend it to randomized algorithms.

Any deterministic query algorithm can be represented as a (possibly infinite) binary decision tree $T$. Each internal node is labeled by a query $\mathrm{Query}(i, \mathbf{x})$, where $i \in N$ and $\mathbf{x} \in \Delta(S)$. The two outgoing edges correspond to the oracle answers True and False. Each leaf is labeled either by an output lottery $\mathbf{x}^* \in \Delta(S)$ or by NULL. For an instance $\mathcal{I}$, let $Q_T(\mathcal{I})$ denote the number of queries made on $\mathcal{I}$, i.e., the depth of the root-to-leaf path followed by $T$ on $\mathcal{I}$. The worst-case query complexity of $T$ is $\sup_{\mathcal{I}} Q_T(\mathcal{I})$.

We call $T$ *correct* if on every instance $\mathcal{I}$ it outputs a unanimously acceptable lottery when one exists, and outputs NULL otherwise. Then, we have the following lemma.

**Lemma D.1.** *Let $T$ be any correct decision tree. Fix any feasible instance $\mathcal{I}$ (i.e., there exists $\mathbf{x} \in \Delta(S)$ such that $U_i(\mathbf{x}) \geq \tau_i$ for all $i \in N$). Then along the (unique) root-to-leaf path traversed by $T$ on $\mathcal{I}$, every agent $i \in N$ is queried at least once. In particular, $Q_T(\mathcal{I}) \geq n$.*

*Proof.* Let $P$ be the root-to-leaf path followed by $T$ on $\mathcal{I}$. Since $\mathcal{I}$ is feasible and $T$ is correct, the reached leaf is labeled by some lottery $\mathbf{x}^*$ (not NULL).

Suppose for contradiction that there exists an agent $j \in N$ that is never queried along $P$. Construct a new instance $\mathcal{I}'$ by changing only agent $j$ as follows: set $u_j'(s) = 0$ for all $s \in S$ and set $\tau_j' = \varepsilon$. Then for every lottery $\mathbf{x} \in \Delta(S)$, we have $U_j'(\mathbf{x}) = 0 < \tau_j'$, so agent $j$ rejects every query. All other agents are unchanged, so every query made along $P$ (which never involves $j$) receives the same answer on $\mathcal{I}'$ as on $\mathcal{I}$. Therefore $T$ follows the same path $P$ on $\mathcal{I}'$ and outputs the same $\mathbf{x}^*$.

However, $\mathcal{I}'$ is infeasible (agent $j$ rejects all lotteries), so correctness requires output NULL, a contradiction. Thus, every agent must be queried at least once along $P$. □

Next, we introduce a grid family of lotteries. Since $1/\varepsilon \in \mathbb{Z}_+$ by our finite-precision assumption, define the $\varepsilon$-grid on the simplex by

$$\Delta_\varepsilon(S) := \Big\{x \in \Delta(S) : x_j \in \{0, \varepsilon, 2\varepsilon, \dots, 1\} \ \forall j \in [m]\Big\}, \quad \text{and} \quad \Delta_\varepsilon^+(S) := \{x \in \Delta_\varepsilon(S) : x_j \geq \varepsilon \ \forall j \in [m]\}.$$

We only use these sets to define a hard family; the algorithm itself may query arbitrary rational lotteries.

**Lemma D.2.** *Fix $m \geq 2$ and $\varepsilon > 0$ such that $1/\varepsilon \geq m$. For each $\mathbf{x} = (x_1, \dots, x_m) \in \Delta_\varepsilon^+(S)$, define an instance $\mathcal{I}_\mathbf{x}$ with $m$ agents as follows: for each agent $i \in [m]$,*

$$u_i(s_i) = 1, \quad u_i(s_j) = 0 \ (j \neq i), \text{ and } \quad \tau_i = x_i.$$

*Then the set of unanimously acceptable lotteries in $\mathcal{I}_\mathbf{x}$ is exactly the singleton $\{\mathbf{x}\}$. Moreover, any correct deterministic decision tree has depth at least $\Omega((m-1)\log(1/\varepsilon))$ on some instance $\mathcal{I}_\mathbf{x}$.*

*Proof.* For any lottery $\mathbf{x}' = (x_1', \dots, x_m') \in \Delta(S)$ and any $i \in [m]$, the utility is $U_i(\mathbf{x}') = x_i'$. Thus agent $i$ accepts $\mathbf{x}'$ if and only if $x_i' \geq \tau_i = x_i$.

First, $\mathbf{x}$ is unanimously acceptable since $x_i = \tau_i$ for all $i$. Now take any unanimously acceptable $\mathbf{x}'$. Then $x_i' \geq x_i$ for all $i \in [m]$. Let $\delta_i := x_i' - x_i \geq 0$. Because both $\mathbf{x}$ and $\mathbf{x}'$ lie in the simplex,

$$\sum_{i=1}^m \delta_i = \sum_{i=1}^m x_i' - \sum_{i=1}^m x_i = 1 - 1 = 0,$$

so $\delta_i = 0$ for all $i$, i.e., $\mathbf{x}' = \mathbf{x}$. Hence the unanimously acceptable set is $\{\mathbf{x}\}$.

Therefore, for distinct $\mathbf{x}, \mathbf{x}' \in \Delta_\varepsilon^+(S)$, a correct algorithm must output different lotteries on $\mathcal{I}_\mathbf{x}$ and $\mathcal{I}_{x'}$. In a decision tree, the output at a leaf is fixed, so $\mathcal{I}_\mathbf{x}$ and $\mathcal{I}_{x'}$ must reach different leaves. Hence any correct tree has at least $|\Delta_\varepsilon^+(S)|$ leaves.

We now lower bound $|\Delta_\varepsilon^+(S)|$. Writing $x_i = k_i\varepsilon$ with integers $k_i \geq 1$, the constraint $\sum_{i=1}^m x_i = 1$ becomes $\sum_{i=1}^m k_i = 1/\varepsilon$. The number of positive integer solutions is $|\Delta_\varepsilon^+(S)| = \binom{1/\varepsilon - 1}{m-1}$. A binary tree of depth $d$ has at most $2^d$ leaves, so $d \geq \log|\Delta_\varepsilon^+(S)|$.

Finally, using the standard inequality $\binom{a}{b} \geq (a/b)^b$ for $a \geq b \geq 1$, with $a = 1/\varepsilon - 1$ and $b = m - 1$, we get

$$\log|\Delta_\varepsilon^+(S)| = \log\binom{1/\varepsilon - 1}{m-1} \geq (m-1)\log\Big(\frac{1/\varepsilon - 1}{m-1}\Big) \geq (m-1)\log\Big(\frac{1}{\varepsilon m}\Big).$$

Then, since $m \leq (1/\varepsilon)^{1-\delta}$ for some fixed $\delta \in (0,1]$ (by our assumption in the preliminaries), we have $\frac{1}{\varepsilon m} \geq (1/\varepsilon)^\delta$, hence $\log(1/(\varepsilon m)) \geq \delta \log(1/\varepsilon)$. Thus $\log|\Delta_\varepsilon^+(S)| = \Omega((m-1)\log(1/\varepsilon))$, proving the lemma. □

We now prove the stated lower bound (our main result) for deterministic algorithms, by splitting into two cases.

**Case 1:** $n \geq m$. For each $\mathbf{x} \in \Delta_\varepsilon^+(S)$, define an instance $\widetilde{\mathcal{I}}_\mathbf{x}$ with $n$ agents as follows. Agents $1, \dots, m$ are defined exactly as in Lemma D.2. The remaining $n - m$ agents are *dummy* agents that accept every lottery: for each $i \in \{m+1, \dots, n\}$ set

$$u_i(s_j) = 1 \text{ for all } j \in [m] \quad \text{and} \quad \tau_i = 1.$$

Then $U_i(\mathbf{x}') = 1$ for every lottery $\mathbf{x}'$, so these agents always answer True. The unanimously acceptable set of $\widetilde{\mathcal{I}}_\mathbf{x}$ is still $\{\mathbf{x}\}$, since the dummy agents impose no constraint and the first $m$ agents force uniqueness by Lemma D.2.

Let $T$ be any correct decision tree and consider its behavior on the family of instances $\mathcal{F} := \{\widetilde{\mathcal{I}}_\mathbf{x} : \mathbf{x} \in \Delta_\varepsilon^+(S)\}$. By Lemma D.1, on every feasible instance $\widetilde{\mathcal{I}}_\mathbf{x}$, the path followed by $T$ queries each of the $n$ agents at least once. In particular, on each $\widetilde{\mathcal{I}}_\mathbf{x}$ it makes at least $n - m$ queries to the dummy agents.

Now prune $T$ by deleting every internal node that queries a dummy agent and contracting to its `True` child (which is the only possible answer on $\mathcal{F}$). Call the pruned tree $T'$. For each instance $\widetilde{\mathcal{I}}_{\mathbf{x}} \in \mathcal{F}$, the pruned execution path in $T'$ ends at a leaf labeled by $\mathbf{x}$ (the same output), so distinct $\mathbf{x}$ still reach distinct leaves of $T'$. Thus, $T'$ has at least $|\Delta_\varepsilon^+(S)|$ leaves, implying that some $\widetilde{\mathcal{I}}_{\mathbf{x}}$ satisfies $Q_{T'}(\widetilde{\mathcal{I}}_{\mathbf{x}}) \geq \log|\Delta_\varepsilon^+(S)|$. For that same instance,

$$Q_T(\widetilde{\mathcal{I}}_{\mathbf{x}}) \geq (n-m) + Q_{T'}(\widetilde{\mathcal{I}}_{\mathbf{x}}) \geq (n-m) + \log|\Delta_\varepsilon^+(S)|.$$

By Lemma D.2, $\log|\Delta_\varepsilon^+(S)| = \Omega((m-1)\log(1/\varepsilon))$, so

$$\sup_{\mathcal{I}} Q_T(\mathcal{I}) \geq \Omega\big((n-m) + (m-1)\log(1/\varepsilon)\big).$$

**Case 2:** $n < m$. Consider lotteries supported on the first $n$ alternatives:

$$\Delta_n^+(S) := \Big\{\mathbf{x} \in \Delta(S) : x_i \in \{\varepsilon, 2\varepsilon, \ldots, 1\} \text{ for all } i \leq n,\ x_i \geq \varepsilon \text{ for all } i \leq n,\ \text{and } x_{n+1} = \cdots = x_m = 0\Big\}.$$

For each $x \in \Delta_n^+(S)$ define an instance $\widehat{\mathcal{I}}_{\mathbf{x}}$ with $n$ agents, where each agent $i \in [n]$ has

$$u_i(s_i) = 1, \quad u_i(s_j) = 0\ (j \neq i), \text{ and } \quad \tau_i = x_i.$$

We claim the unanimously acceptable set of $\widehat{\mathcal{I}}_{\mathbf{x}}$ is $\{\mathbf{x}\}$. Indeed, if a lottery $\mathbf{x}'$ is unanimously acceptable, then for each agent $i \leq n$ we have $U_i(\mathbf{x}') = x_i' \geq \tau_i = x_i$. For $j > n$, $x_j = 0$ and $x_j' \geq 0$, so $x_j' \geq x_j$ as well. Thus, $\mathbf{x}' \geq \mathbf{x}$ coordinate-wise, but since $\mathbf{x}, \mathbf{x}' \in \Delta(S)$ we must have $\sum_j(x_j' - x_j) = 0$, forcing $\mathbf{x}' = \mathbf{x}$. Therefore any correct decision tree must have at least $|\Delta_n^+(S)|$ leaves over the family $\{\widehat{\mathcal{I}}_{\mathbf{x}} : \mathbf{x} \in \Delta_n^+(S)\}$, and consequently, some instance requires at least $\log|\Delta_n^+(S)|$ queries. Moreover, $|\Delta_n^+(S)| = \binom{1/\varepsilon-1}{n-1}$, and under the assumption that $n < m \leq (1/\varepsilon)^{1-\delta}$ we obtain $\log|\Delta_n^+(S)| = \Omega((n-1)\log(1/\varepsilon))$. This gives us

$$\sup_{\mathcal{I}} Q_T(\mathcal{I}) \geq \Omega\big((n-1)\log(1/\varepsilon)\big).$$

Combining the two cases gives us the deterministic lower bound

$$\sup_{\mathcal{I}} Q_T(\mathcal{I}) \geq \Omega\Big((n - \min\{n,m\}) + (\min\{n,m\} - 1)\log(1/\varepsilon)\Big),$$

Now, we detail the extension of this lower bound to randomized algorithms. We interpret the query complexity of a randomized algorithm as worst-case expected number of queries (and the algorithm must be correct on every instance for every realization of its internal randomness).

Formally, a randomized algorithm $R$ is a distribution over deterministic decision trees. Let $T_\sigma$ denote the deterministic tree obtained by fixing the random seed $\sigma$. Define the expected query complexity of $R$ on instance $\mathcal{I}$ as

$$Q_R(\mathcal{I}) := \mathbb{E}_\sigma\big[Q_{T_\sigma}(\mathcal{I})\big],$$

and its worst-case expected query complexity as $\sup_{\mathcal{I}} Q_R(\mathcal{I})$.

We use Yao's minimax principle (Yao, 1977) in the following form.

**Lemma D.3.** *For any distribution $\mu$ over instances,*

$$\inf_R \sup_{\mathcal{I}} Q_R(\mathcal{I}) \geq \min_T \mathbb{E}_{\mathcal{I} \sim \mu}\big[Q_T(\mathcal{I})\big],$$

*where the minimum ranges over correct deterministic decision trees $T$, and the infimum ranges over correct randomized algorithms $R$ (distributions over correct deterministic trees).*

*Proof.* Fix $\mu$ and a randomized algorithm $R$ (a distribution over correct deterministic trees). Then

$$\sup_{\mathcal{I}} Q_R(\mathcal{I}) \geq \mathbb{E}_{\mathcal{I} \sim \mu}[Q_R(\mathcal{I})] = \mathbb{E}_{\mathcal{I} \sim \mu}\mathbb{E}_\sigma[Q_{T_\sigma}(\mathcal{I})] = \mathbb{E}_\sigma \mathbb{E}_{\mathcal{I} \sim \mu}[Q_{T_\sigma}(\mathcal{I})] \geq \inf_T \mathbb{E}_{\mathcal{I} \sim \mu}[Q_T(\mathcal{I})].$$

Taking $\inf_R$ on the left-hand side gives us the claim. $\qquad\square$

We also use a standard average-depth bound.

**Lemma D.4.** *Let $T$ be a binary decision tree. For any leaf $\ell$ of $T$, let $\mathrm{depth}(\ell)$ denote the number of internal nodes on the path from the root to $\ell$. Let $L$ be any finite set of leaves of $T$, and let $\ell \sim \mathrm{Unif}(L)$ be a uniformly random leaf in $L$. Then $\mathbb{E}[\mathrm{depth}(\ell)] \geq \log |L|$.*

*Proof.* By Kraft's inequality, $\sum_{\ell \in L} 2^{-\mathrm{depth}(\ell)} \leq 1$. Let $D := \mathrm{depth}(\ell)$. Since the function $f(y) = 2^{-y}$ is convex, Jensen's inequality gives us

$$2^{-\mathbb{E}[D]} = f(\mathbb{E}[D]) \leq \mathbb{E}[f(D)] = \mathbb{E}[2^{-D}].$$

Because $\ell$ is uniform on $L$, $\mathbb{E}[2^{-D}] = \frac{1}{|L|} \sum_{\ell \in L} 2^{-\mathrm{depth}(\ell)} \leq \frac{1}{|L|}$. Taking $\log$ of both sides gives us $\mathbb{E}[D] \geq \log |L|$. $\quad\square$

Now choose $\mu$ to be the uniform distribution over the hard family from the relevant deterministic case:

- If $n \geq m$, let $\mu$ be uniform over $\mathcal{F} = \{\widetilde{\mathcal{I}}_{\mathbf{x}} : \mathbf{x} \in \Delta_\varepsilon^+(S)\}$.

- If $n < m$, let $\mu$ be uniform over $\{\widehat{\mathcal{I}}_{\mathbf{x}} : \mathbf{x} \in \Delta_n^+(S)\}$.

**Lemma D.5.** *For every correct deterministic decision tree $T$,*

$$\mathbb{E}_{\mathcal{I} \sim \mu}[Q_T(\mathcal{I})] \geq \begin{cases} (n-m) + \log |\Delta_\varepsilon^+(S)|, & \text{if } n \geq m, \\ \log |\Delta_n^+(S)|, & \text{if } n < m. \end{cases}$$

*Proof.* We prove the two cases.

**Case $n \geq m$.** Fix a correct deterministic tree $T$. For any $\mathcal{I} = \widetilde{\mathcal{I}}_{\mathbf{x}} \in \mathcal{F}$, Lemma D.1 implies that the execution path on $\mathcal{I}$ queries each of the $n$ agents at least once, hence makes at least $n - m$ queries to the dummy agents. Prune $T$ into $T'$ by contracting dummy-agent query nodes to their `True` child, exactly as in the deterministic proof. For each $\mathcal{I} \in \mathcal{F}$ we have $Q_T(\mathcal{I}) \geq (n-m) + Q_{T'}(\mathcal{I})$.

Moreover, since each $\mathcal{I} = \widetilde{\mathcal{I}}_{\mathbf{x}}$ has a unique correct output $\mathbf{x}$, the map $\mathcal{I} \mapsto$ (reached leaf of $T'$) is injective over $\mathcal{F}$, and its image is a set $L$ of size $|\Delta_\varepsilon^+(S)|$. Because $\mu$ is uniform over $\mathcal{F}$, the induced distribution on $L$ is uniform. Applying Lemma D.4 to $T'$ gives us $\mathbb{E}_{\mathcal{I} \sim \mu}[Q_{T'}(\mathcal{I})] \geq \log |\Delta_\varepsilon^+(S)|$. Therefore,

$$\mathbb{E}_{I \sim \mu}[Q_T(I)] \geq (n-m) + \log |\Delta_\varepsilon^+(S)|.$$

**Case $n < m$.** The same injectivity argument applies (no pruning needed): distinct $\mathbf{x}$ induce distinct leaves, and $\mu$ is uniform on that family, so Lemma D.4 gives $\mathbb{E}_{\mathcal{I} \sim \mu}[Q_T(I)] \geq \log |\Delta_n^+(S)|$. $\quad\square$

Finally, by Lemma D.3 and Lemma D.5,

$$\inf_R \sup_{\mathcal{I}} Q_R(\mathcal{I}) \geq \begin{cases} (n-m) + \log |\Delta_\varepsilon^+(S)|, & \text{if } n \geq m, \\ \log |\Delta_n^+(S)|, & \text{if } n < m. \end{cases}$$

Using Lemma D.2 (and the same binomial counting bound with $m$ replaced by $n$ in the second case), we have $\log |\Delta_\varepsilon^+(S)| = \Omega((m-1)\log(1/\varepsilon))$ and $\log |\Delta_n^+(S)| = \Omega((n-1)\log(1/\varepsilon))$. Therefore every correct randomized algorithm satisfies

$$\sup_{\mathcal{I}} Q_R(\mathcal{I}) \geq \Omega\Big((n - \min\{n, m\}) + (\min\{n, m\} - 1)\log(1/\varepsilon)\Big),$$

completing the proof for randomized algorithms as well.

## D.2. Proof of Theorem 4.2

Consider the single-agent case $n = 1$. For each $j \in [m]$, define a feasible instance $\mathcal{I}_j$ by setting

$$\tau_1 = 1, \quad u_1(s_j) = 1, \quad u_1(s_k) = 0 \text{ for all } k \in [m] \setminus \{j\}.$$

Also define an infeasible instance $\mathcal{I}_0$ by

$$\tau_1 = 1, \quad u_1(s_k) = 0 \text{ for all } k \in [m].$$

We first characterize the oracle on these instances. Fix any lottery $\mathbf{x} \in \Delta(S)$.

On $\mathcal{I}_j$, we have

$$U_1(\mathbf{x}) = \langle \mathbf{u}_1, \mathbf{x} \rangle = x_j,$$

and therefore

$$\text{Query}(1, \mathbf{x}) = \texttt{True} \iff x_j \geq 1 \iff \mathbf{x} = \mathbf{e}_j,$$

where the last equivalence uses that $\mathbf{x} \in \Delta(S)$ implies $x_j \leq 1$ with equality if and only if $\mathbf{x} = \mathbf{e}_j$. Thus, $\mathcal{I}_j$ is feasible and the acceptable set is exactly $\{\mathbf{e}_j\}$.

On $\mathcal{I}_0$, we have $U_1(\mathbf{x}) = 0 < \tau_1 = 1$ for every $\mathbf{x} \in \Delta(S)$, so $\text{Query}(1, \mathbf{x}) = \texttt{False}$ for every query and the instance is infeasible.

Next, we show the lower bound on deterministic algorithms. Let $\mathcal{F}$ be any deterministic algorithm that is correct on every instance. Run $\mathcal{F}$ on $\mathcal{I}_0$. Since $\mathcal{I}_0$ rejects every lottery, every query made by $\mathcal{F}$ is answered $\texttt{False}$, and (by correctness) $\mathcal{F}$ must eventually halt and output NULL.

Let $q_0$ be the number of queries $\mathcal{F}$ makes on this run, and let

$$J := \{j \in [m] : \mathcal{F} \text{ queried the pure lottery } \mathbf{e}_j \text{ at least once among its first } q_0 \text{ queries}\}.$$

If $|J| < m$, choose some $j^* \in [m] \setminus J$. Consider running $\mathcal{F}$ on the feasible instance $\mathcal{I}_{j^*}$. By our characterization, $\mathcal{I}_{j^*}$ answers $\texttt{True}$ only on the single query $\mathbf{e}_{j^*}$, and by the choice of $j^*$, $\mathcal{F}$ does not query $\mathbf{e}_{j^*}$ among its first $q_0$ queries. Therefore, the first $q_0$ oracle answers received by $\mathcal{F}$ on $\mathcal{I}_{j^*}$ are also all $\texttt{False}$. Since $\mathcal{F}$ is deterministic, it follows that $\mathcal{F}$ receives exactly the same query/answer history on $\mathcal{I}_0$ and on $\mathcal{I}_{j^*}$, and hence produces the same output on both instances.

Now, we show that this is impossible: correctness requires output NULL on $\mathcal{I}_0$, but requires output $\mathbf{e}_{j^*}$ on $\mathcal{I}_{j^*}$ because $\mathbf{e}_{j^*}$ is the only acceptable lottery there. Thus we must have $|J| = m$, meaning that $\mathcal{F}$ queries every pure lottery $\mathbf{e}_1, \ldots, \mathbf{e}_m$ before it can correctly output NULL on $\mathcal{I}_0$. In particular, $q_0 \geq m$, so the worst-case query complexity of any correct deterministic algorithm is at least $m$.

Finally, we prove the lower bound for any randomized algorithm. Let $\mathcal{R}$ be any randomized algorithm in our model (as mentioned in Section 2, such a randomized algorithm is always correct for every realization of internal randomness). Fix any realization of its internal randomness; this gives us a deterministic algorithm that is still correct on every instance. Applying the deterministic lower bound above to this deterministic instantiation shows it makes at least $m$ queries on $\mathcal{I}_0$. Since this holds for every realization, $\mathcal{R}$ makes at least $m$ queries on $\mathcal{I}_0$ in expectation as well. Therefore every correct randomized algorithm has worst-case expected query complexity at least $m$.

# E. Omitted Proofs from Section 5

## E.1. Proof of Theorem 5.1

We prove the statement for an arbitrary permutation $\sigma$; instantiating with $\sigma = \hat{\sigma}$ gives us the result.

Fix any permutation $\sigma$ of $N$, and consider Algorithm 2[$\sigma$]. Recall the definition of record agents: for any permutation $\sigma$, let $R(\sigma)$ be the number of agents on which LearnHyperplane is invoked during the run of Algorithm 2[$\sigma$].

**Correctness** follows from Theorem 3.3, whose proof does not rely on the specific fixed scan order $1, 2, \ldots, n$, only on the facts that (i) in each iteration of the **while** loop (Line 3 of Algorithm 2) the algorithm proposes a candidate $\mathbf{x} \leftarrow \text{Select}(C)$,

(ii) it checks unlearned agents one-by-one until either finding a violator or exhausting the scan, and (iii) upon finding a violator it elicits that agent's halfspace and adds it to $C$ (or returns NULL if LearnHyperplane returns REJECTALL). These properties remain true under any fixed permutation order $\sigma$. Hence Algorithm 2$[\sigma]$ outputs a unanimously acceptable lottery when one exists and outputs NULL otherwise.

Thus, we focus on proving its **query complexity**, bounding the number of queries made by Algorithm 2$[\sigma]$. By Lemma 3.1, each call to LearnHyperplane makes $\mathcal{O}(m \log(1/\varepsilon))$ queries. Since Algorithm 2$[\sigma]$ calls LearnHyperplane exactly for the $R(\sigma)$ agents, the total number of elicitation queries is

$$\mathcal{O}(R(\sigma)\, m \log(1/\varepsilon))$$

Consider one iteration of the outer **while** loop of Algorithm 2$[\sigma]$ (Line 3). After computing $\mathbf{x} \leftarrow \text{Select}(C)$, the algorithm queries agents in the order $\sigma(1), \ldots, \sigma(n)$, skipping learned agents (those not in $N'$), and stops either when it finds the first violator $j$ with $\text{Query}(j, \mathbf{x}) = \text{False}$, or when it finishes the scan and returns $\mathbf{x}$. In a single such iteration, each agent is queried at most once, so each iteration uses at most $n$ membership queries for verification. Let $\gamma$ be the number of **while** iterations executed. Every such iteration (except possibly the last) ends by learning at least one new violating agent (and hence invoking LearnHyperplane at least once), and each agent can be learned at most once because it is removed from $N'$ immediately after being learned. Therefore, $\gamma \leq R(\sigma) + 1$, and the total number of verification queries is at most

$$n\gamma \leq n(R(\sigma) + 1).$$

The total number of membership queries made by Algorithm 2$[\sigma]$ is therefore

$$\mathcal{O}(R(\sigma)\, m \log(1/\varepsilon)) + n(R(\sigma) + 1) = \mathcal{O}((n + m \log(1/\varepsilon))(R(\sigma) + 1)).$$

Since we assumed $R(\sigma) \geq 1$, then $R(\sigma) + 1 \leq 2R(\sigma)$, and so the query complexity is simply

$$\mathcal{O}((n + m \log(1/\varepsilon))\, R(\sigma)).$$

If $R(\sigma) = 0$, the algorithm makes a single verification scan and uses at most $n$ queries, which is consistent with the $(R(\sigma) + 1)$ bound.

### E.2. Proof of Theorem 5.2

We prove the statement for an arbitrary permutation $\sigma$; instantiating it with $\sigma = \hat{\sigma}$ gives us the result.

Fix any permutation $\sigma$ of $N$, and consider Algorithm 3$[\sigma]$. Recall that Algorithm 3$[\sigma]$ is equivalent to running Algorithm 3, except that it initializes the weights as

$$w_i^{(1)} := \left\lceil \frac{n}{\text{rank}_\sigma(i)} \right\rceil \qquad \text{for all } i \in N,$$

and then runs the same sampling and multiplicative updating procedure as in Algorithm 3.

**Correctness** follows from Algorithm 3, whose proof does not rely on how weights are initialized, only on: (i) correctness of LearnHyperplane, (ii) the fact that returning $\mathbf{x}$ when no violators exist is unanimously acceptable, and (iii) the fact that returning NULL on REJECTALL or on infeasible sampled constraints is correct.

Thus, we focus on proving its **query complexity**.

In the proof of Theorem 3.4 (Step 1), a witness set $B \subseteq N$ is:

- if the instance is feasible ($P(N) \neq \varnothing$): any basis $B$ of $N$, with $|B| \leq m - 1$;

- if infeasible ($P(N) = \varnothing$): any Helly witness $B$ with $|B| \leq m$ and $P(B) = \varnothing$.

**Witness parameter.** Let $W$ be the family of valid witness sets used in the analysis of Algorithm 3: $W$ consists of bases when $P(N) \neq \varnothing$ and Helly witnesses when $P(N) = \varnothing$. Define the prefix-witness parameter

$$K := E(\sigma) := \min_{B \in W} \max_{i \in B} \text{rank}_\sigma(i),$$

i.e., there exists some witness set contained in the first $K$ positions of $\sigma$. Fix a witness set $B^* \in W$ achieving this minimum, and let $b := |B^*| \leq m$.

**Potential and drift.** Let $w^{(t)}$ be the weight vector at iteration $t$, $W^{(t)} := \sum_{i=1}^{n} w_i^{(t)}$, and define the potential

$$\Phi_t := \frac{\prod_{i \in B^*} w_i^{(t)}}{(W(t))^b}.$$

As in Lemma C.4 in the proof of Theorem 3.4, on every nonterminal iteration $t < T$, Lemma C.3 implies the violator set intersects $B^*$, hence the numerator multiplies by at least 2, while the denominator increases by a factor $(1 + \omega^{(t)}/W^{(t)})^b$. Using Lemma C.2 (with $r = 16(m-1)^2$) to bound $E[\omega^{(t)} \mid \mathcal{F}_{t-1}]$ and concavity of log, we obtain a constant $c_0 > 0$ (e.g. $c_0 = \log 2 - 1/4$) such that

$$\mathbb{E}\big[(\log \Phi_{t+1} - \log \Phi_t)\mathbf{1}[t < T] \mid \mathcal{F}_{t-1}\big] \geq c_0 \mathbf{1}[t < T].$$

**Bounding $\mathbb{E}[T]$.** For $M \geq 1$, let $T^{(M)} := \min\{T, M\}$. Telescoping and the drift bound gives us

$$\mathbb{E}[\min\{T-1, M-1\}] \leq \frac{\log(1/\Phi_1)}{c_0}.$$

Letting $M \to \infty$ (monotone convergence) gives us

$$\mathbb{E}[T] \leq 1 + \frac{\log(1/\Phi_1)}{c_0} = \mathcal{O}(\log(1/\Phi_1)).$$

**Bounding $\Phi_1$ with predictions.** Under the initialization inferred from our predicted permutation,

$$W^{(1)} = \sum_{k=1}^{n} \left\lceil \frac{n}{k} \right\rceil \leq \sum_{k=1}^{n} \left( \frac{n}{k} + 1 \right) = n \left( \sum_{k=1}^{n} \frac{1}{k} + 1 \right).$$

Moreover, every $i \in B^*$ has $\operatorname{rank}_\sigma(i) \leq K$, thus

$$w_i^{(1)} = \left\lceil \frac{n}{\operatorname{rank}_\sigma(i)} \right\rceil \geq \frac{n}{K} \implies \prod_{i \in B^*} w_i^{(1)} \geq (n/K)^b.$$

Therefore,

$$\Phi_1 \geq \frac{(n/K)^b}{(n(\sum_{k=1}^{n} \frac{1}{k} + 1))^b} = \frac{1}{((\sum_{k=1}^{n} \frac{1}{k} + 1)K)^b},$$

so

$$\log(1/\Phi_1) \leq b \log\left( \left( \sum_{k=1}^{n} \frac{1}{k} + 1 \right) K \right) \leq m \log\left( \mathcal{O}(\log n) K \right).$$

**Query complexity.** Each iteration uses at most $n$ membership queries in the verification step, so verification costs $\mathcal{O}(n\mathbb{E}[T])$. Let $L$ be the number of distinct agents whose hyperplanes are learned. Each iteration samples at most $r = \mathcal{O}(m^2)$ agents, and each agent is learned at most once, so $L \leq \min\{n, rT\}$ and by concavity

$$\mathbb{E}[L] \leq \min\{n, r\mathbb{E}[T]\} = \mathcal{O}(\min\{n, m^3 \log(K \log n)\}).$$

Each learned hyperplane costs $\mathcal{O}(m \log(1/\varepsilon))$ queries (by Lemma 3.1), giving total expected queries

$$\mathcal{O}\left( nm \log(K \log n) \right) + m \log(1/\varepsilon) \cdot \min\{n, m^3 \log(K \log n)\}.$$

Substituting $E(\sigma) = K$ and setting $\mu = \log(K \log n) = \log(E(\sigma) \log n) = \log E(\sigma) + \log \log n$, we get

$$\mathcal{O}\left( nm\mu + \min\{n, m^3 \mu\} \cdot m \log(1/\varepsilon) \right),$$

as desired.

### E.3. Lower Bound with Permutation Predictions

**Proposition E.1.** *Fix any $n, m$. Then, any (deterministic or randomized) algorithm) that is correct for every instance $\mathcal{I}$ and permutation prediction $\hat{\sigma}$ (i.e., outputs a unanimously acceptable lottery whenever one exists and* NULL *otherwise) has a worst-case (expected) number of queries is $\Omega((n - \min\{n, m\}) + (\min\{n, m\} - 1)\log(1/\varepsilon))$.*

*Proof.* Fix $n, m$. Let $\mathcal{A}$ be any deterministic (or randomized) algorithm that is correct for *every* pair $(\mathcal{I}, \hat{\sigma})$ where $\mathcal{I}$ is an instance of the problem and $\hat{\sigma}$ is an arbitrary permutation of $N$ (i.e., $\mathcal{A}$ outputs a unanimously acceptable lottery whenever one exists and outputs NULL otherwise).

We will prove our result by showing, for each case $n \geq m$ and $n < m$, a hard family of feasible instances together with a fixed permutation advice $\hat{\sigma}$ whose prefix witness parameter is small. Since $\mathcal{A}$ must be correct for *every* permutation advice, lower bounding its query complexity under this specific $\hat{\sigma}$ is without loss of generality.

We first show the following lemma.

**Lemma E.2.** *Fix any permutation advice $\hat{\sigma}$. Let $T$ be any* correct *deterministic decision tree for the problem under advice $\hat{\sigma}$ (i.e., $T$ outputs a unanimously acceptable lottery on every feasible instance and outputs* NULL *on every infeasible instance, when run with input $(\mathcal{I}, \hat{\sigma})$). Then on every feasible instance $\mathcal{I}$, along the root-to-leaf path followed by $T$ on input $(\mathcal{I}, \hat{\sigma})$, every agent $i \in N$ is queried at least once. In particular, $T$ makes at least $n$ membership queries on $\mathcal{I}$.*

*Proof.* Let $P$ be the root-to-leaf path followed by $T$ on input $(\mathcal{I}, \hat{\sigma})$. Because $\mathcal{I}$ is feasible and $T$ is correct, the reached leaf is labeled by some lottery $\mathbf{x}^* \in \Delta(S)$ (not NULL). Suppose for contradiction that some agent $j \in N$ is never queried along $P$. Construct a new instance $\mathcal{I}'$ by changing only agent $j$ as follows: set $u'_j(s) = 0$ for all $s \in S$ and set $\tau'_j = \varepsilon$. Then $U'_j(\mathbf{x}) = 0 < \tau'_j$ for every lottery $\mathbf{x} \in \Delta(S)$, so agent $j$ rejects every query and $\mathcal{I}'$ is infeasible. However, since $j$ is never queried along $P$, every query asked along $P$ receives the same answer on $(\mathcal{I}', \hat{\sigma})$ as on $(\mathcal{I}, \hat{\sigma})$, and so $T$ follows the same path $P$ and outputs the same non-NULL lottery $\mathbf{x}^*$. This contradicts correctness on $(\mathcal{I}', \hat{\sigma})$, which would require output NULL. $\square$

We split our analysis into two cases.

**Case 1:** $n \geq m$. Let $\Delta^+(S)$ denote the positive $\varepsilon$-grid on the simplex, i.e.,

$$\Delta^+(S) := \{\mathbf{x} \in \Delta(S) : x_j \in \{\varepsilon, 2\varepsilon, \ldots, 1\} \text{ for all } j \in [m]\}.$$

Note that under $\varepsilon$-precision and $m \leq (1/\varepsilon)^{1-\delta} \leq 1/\varepsilon$, this set is nonempty.

For each $\mathbf{x} = (x_1, \ldots, x_m) \in \Delta^+(S)$ define an instance $\mathcal{I}_{\mathbf{x}}$ with $n$ agents as follows:

- For each $i \in [m]$, set $u_i(s_i) = 1$, $u_i(s_j) = 0$ for all $j \neq i$, and set $\tau_i = x_i$.

- For each dummy agent $i \in \{m+1, \ldots, n\}$, set $u_i(s_j) = 1$ for all $j \in [m]$ and $\tau_i = 1$ (so $i$ accepts every lottery).

Fix the permutation advice $\hat{\sigma}$ so that agents $1, \ldots, m$ occupy the first $m$ positions (e.g. $\hat{\sigma}(k) = k$ for all $k \in [n]$). Since all nontrivial constraints are contained in agents $1, \ldots, m$, there exists a witness set contained in the first $m$ positions; thus $E(\hat{\sigma}) \leq m$. Moreover, since each dummy agent has $\mathcal{A}_i = \Delta(S)$, no minimal witness set (basis) can include a dummy agent, so some valid witness set lies entirely in $\{1, \ldots, m\}$.

We claim that the unanimously acceptable set of $\mathcal{I}_{\mathbf{x}}$ is the singleton $\{\mathbf{x}\}$. Indeed, for any lottery $\mathbf{y} \in \Delta(S)$ and any $i \in [m]$, we have $U_i(\mathbf{y}) = y_i$. Thus agent $i$ accepts $\mathbf{y}$ if and only if $y_i \geq \tau_i = x_i$. If $\mathbf{y}$ is unanimously acceptable, then $y_i \geq x_i$ for all $i \in [m]$. Let $d_i := y_i - x_i \geq 0$. Because $\mathbf{x}, \mathbf{y} \in \Delta(S)$, we have $\sum_{i=1}^m d_i = \sum_{i=1}^m y_i - \sum_{i=1}^m x_i = 1 - 1 = 0$, forcing $d_i = 0$ for all $i$, i.e., $\mathbf{y} = \mathbf{x}$.

Now, fix any correct deterministic decision tree $T$ and consider its behavior on the family $\mathcal{F} := \{\mathcal{I}_{\mathbf{x}} : \mathbf{x} \in \Delta^+(S)\}$ with the advice fixed to $\hat{\sigma}$. By Lemma E.2, on each feasible instance $\mathcal{I}_{\mathbf{x}}$ the execution path queries every agent at least once. In particular, it makes at least $n - m$ queries to dummy agents. Over the family $\mathcal{F}$, every dummy-agent query is answered

`True`. Prune $T$ into a tree $T'$ by contracting every internal node that queries a dummy agent to its `True` child. Then for every $\mathbf{x} \in \Delta^+(S)$,

$$Q_T(\mathcal{I}_\mathbf{x}, \hat{\sigma}) \geq (n - m) + Q_{T'}(\mathcal{I}_\mathbf{x}, \hat{\sigma}).$$

By uniqueness, the correct output on $\mathcal{I}_\mathbf{x}$ is exactly $\mathbf{x}$. Therefore distinct $\mathbf{x} \neq \mathbf{x}'$ must reach distinct leaves of $T'$ (a leaf has a fixed output label), so $T'$ has at least $|\Delta^+(S)|$ leaves. Hence $\sup_\mathbf{x} Q_{T'}(\mathcal{I}_\mathbf{x}, \hat{\sigma}) \geq \log |\Delta^+(S)|$, and thus for some $\mathbf{x}$,

$$Q_T(\mathcal{I}_\mathbf{x}, \hat{\sigma}) \geq (n - m) + \log |\Delta^+(S)|.$$

Let $x_i = k_i \varepsilon$ with integers $k_i \geq 1$. The simplex constraint $\sum_{i=1}^m x_i = 1$ becomes $\sum_{i=1}^m k_i = 1/\varepsilon$. Thus, $|\Delta^+(S)|$ is the number of positive integer solutions to this equation, namely

$$|\Delta^+(S)| = \binom{1/\varepsilon - 1}{m - 1}.$$

Using $\binom{a}{b} \geq (a/b)^b$ for $a \geq b \geq 1$ with $a = 1/\varepsilon - 1$ and $b = m - 1$ gives us

$$\log |\Delta^+(S)| \geq (m - 1) \log \left( \frac{1/\varepsilon - 1}{m - 1} \right) \geq (m - 1) \log \left( \frac{1}{\varepsilon m} \right).$$

By the standing growth condition $m \leq (1/\varepsilon)^{1-\delta}$, we have $1/(\varepsilon m) \geq (1/\varepsilon)^\delta$ and hence

$$\log |\Delta^+(S)| = \Omega((m - 1) \log(1/\varepsilon)).$$

Combining, for $n \geq m$ we obtain

$$\sup_{\mathbf{x} \in \Delta^+(S)} Q_T(\mathcal{I}_\mathbf{x}, \hat{\sigma}) \geq \Omega((n - m) + (m - 1) \log(1/\varepsilon)).$$

**Case 2: $n < m$.** Define the positive $\varepsilon$-grid supported on the first $n$ alternatives: $\Delta_n^+(S) := \{x \in \Delta(S) : x_i \in \{\varepsilon, 2\varepsilon, \ldots, 1\}$ for all $i \leq n, x_{n+1} = \cdots = x_m = 0\}$. For each $\mathbf{x} \in \Delta_n^+(S)$ define an instance $\mathcal{I}_\mathbf{x}$ with $n$ agents: for each $i \in [n]$, set $u_i(s_i) = 1$, $u_i(s_j) = 0$ for $j \neq i$, and $\tau_i = x_i$. Fix any permutation advice $\hat{\sigma}$ (e.g. identity); trivially $E(\hat{\sigma}) \leq n$.

The same coordinate-wise argument shows that $\mathcal{I}_\mathbf{x}$ has a unique unanimously acceptable lottery $\mathbf{x}$. Thus, any correct deterministic decision tree under advice $\hat{\sigma}$ must have at least $|\Delta_n^+(S)|$ leaves over this family, so some instance forces at least $\log |\Delta_n^+(S)|$ queries. Counting as above gives us $|\Delta_n^+(S)| = \binom{1/\varepsilon - 1}{n - 1}$, so $\log |\Delta_n^+(S)| = \Omega((n - 1) \log(1/\varepsilon))$, using again that $n < m \leq (1/\varepsilon)^{1-\delta}$.

Combining the two cases gives us the deterministic lower bound

$$\sup_\mathcal{I} Q_T(\mathcal{I}, \hat{\sigma}) \geq \Omega((n - \min\{n, m\}) + (\min\{n, m\} - 1) \log(1/\varepsilon)).$$

We interpret a randomized algorithm as a distribution over correct deterministic decision trees. For a (random seed-indexed) deterministic tree $T_\omega$, define $Q_\mathcal{A}(\mathcal{I}, \hat{\sigma}) := \mathbb{E}_\omega[Q_{T_\omega}(\mathcal{I}, \hat{\sigma})]$.

We use two standard facts.

**Lemma E.3** (Yao's minimax principle). *For any distribution $\mu$ over instances,*

$$\inf_\mathcal{A} \sup_\mathcal{I} Q_\mathcal{A}(\mathcal{I}, \hat{\sigma}) \geq \min_T \mathbb{E}_{\mathcal{I} \sim \mu}[Q_T(\mathcal{I}, \hat{\sigma})],$$

*where the infimum ranges over correct randomized algorithms $\mathcal{A}$ and the minimum ranges over correct deterministic trees $T$.*

*Proof.* Fix $\mu$ and a randomized algorithm $\mathcal{A}$ (a distribution over deterministic trees). Then

$$\sup_\mathcal{I} Q_\mathcal{A}(\mathcal{I}, \hat{\sigma}) \geq \mathbb{E}_{\mathcal{I} \sim \mu}[Q_\mathcal{A}(\mathcal{I}, \hat{\sigma})] = \mathbb{E}_{\mathcal{I} \sim \mu} \mathbb{E}_{T \sim \mathcal{A}}[Q_T(\mathcal{I}, \hat{\sigma})] = \mathbb{E}_{T \sim \mathcal{A}} \mathbb{E}_{\mathcal{I} \sim \mu}[Q_T(\mathcal{I}, \hat{\sigma})] \geq \min_T \mathbb{E}_{\mathcal{I} \sim \mu}[Q_T(\mathcal{I}, \hat{\sigma})].$$

Taking $\inf_\mathcal{A}$ on the left-hand side proves the claim. $\square$

**Lemma E.4** (Average depth bound). *Let $T$ be a binary decision tree and let $L$ be any finite set of leaves of $T$. If $\ell$ is drawn uniformly from $L$, then $\mathbb{E}[\text{depth}(\ell)] \geq \log |L|$.*

*Proof.* By Kraft's inequality, $\sum_{\ell \in L} 2^{-\text{depth}(\ell)} \leq 1$. Let $D := \text{depth}(\ell)$ where $\ell \sim \text{Unif}(L)$. Then

$$\mathbb{E}[2^{-D}] = \frac{1}{|L|} \sum_{\ell \in L} 2^{-\text{depth}(\ell)} \leq \frac{1}{|L|}.$$

Since $2^{-x}$ is convex, Jensen's inequality gives $2^{-\mathbb{E}[D]} \leq \mathbb{E}[2^{-D}] \leq 1/|L|$, hence $\mathbb{E}[D] \geq \log |L|$. $\qquad \square$

Now choose $\mu$ to be uniform over the corresponding hard family: $\{\mathcal{I}_\mathbf{x} : x \in \Delta^+(S)\}$ when $n \geq m$, and $\{\mathcal{I}_\mathbf{x} : x \in \Delta_n^+(S)\}$ when $n < m$. Fix any correct deterministic tree $T$. In each case, the correct output is unique for every instance in the support of $\mu$. Therefore distinct instances in the support must reach distinct output leaves of the (possibly pruned) tree, so the set $L$ of leaves reached under $\mu$ has size $|L| = |\Delta|$. Because $\mu$ is uniform on instances, the induced distribution on $L$ is uniform. By Lemma E.4,

$$\mathbb{E}_{\mathcal{I} \sim \mu}[Q_T(\mathcal{I}, \hat{\sigma})] \geq \log |L| = \log |\Delta|,$$

and in the $n \geq m$ case the pruning argument contributes the additive $(n-m)$ term exactly as above. Thus the same asymptotic lower bound holds for $\min_T \mathbb{E}_{\mathcal{I} \sim \mu}[Q_T(\mathcal{I}, \hat{\sigma})]$, and Lemma E.3 implies the same lower bound for $\inf_{\mathcal{A}} \sup_{\mathcal{I}} Q_{\mathcal{A}}(\mathcal{I}, \hat{\sigma})$. Equivalently, every correct randomized algorithm has worst-case expected query complexity

$$\Omega\big((n - \min\{n, m\}) + (\min\{n, m\} - 1) \log(1/\varepsilon)\big),$$

even with permutation advice $\hat{\sigma}$ with $E(\hat{\sigma}) \leq \min\{n, m\}$. $\qquad \square$

### E.4. Proof of Theorem 5.3

We first define the warm-started threshold routine used by LearnHyperplane[$\hat{\mathbf{x}}$].

---

**Algorithm 4** ExactThresholdPred($i, k, k', \widehat{\alpha}$)

---

**Input:** agent $i \in N$, indices $k, k' \in [m]$ with Query$(i, e_k) = $ False, Query$(i, e_{k'}) = $ True, and a warm start $\widehat{\alpha} \in [0, 1]$.
**Output:** the exact turning point $\alpha^*_{i;k,k'} \in (0, 1]$ with Query$(i, \mathbf{x}_{k,k',\alpha}) = $ True iff $\alpha \geq \alpha^*_{i;k,k'}$.

1:   $step \leftarrow \varepsilon^2/2$.
2:   $b \leftarrow$ Query$(i, \mathbf{x}_{k,k',\widehat{\alpha}})$.
3:   **if** $b = $ True **then**                                                    $\triangleright \widehat{\alpha} \geq \alpha^*_{i;k,k'}$
4:       $upper \leftarrow \widehat{\alpha}$.
5:       **while** $upper - step > 0$ and Query$(i, \mathbf{x}_{k,k',upper-step}) = $ True **do**
6:            $upper \leftarrow upper - step$;    $step \leftarrow 2 \times step$.
7:       **end while**
8:       $lower \leftarrow \max\{0, upper - step\}$.
9: **else**                                                              $\triangleright \widehat{\alpha} < \alpha^*_{i;k,k'}$
10:      $lower \leftarrow \widehat{\alpha}$.
11:      **while** $lower + step < 1$ and Query$(i, \mathbf{x}_{k,k',lower+step}) = $ False **do**
12:           $lower \leftarrow lower + step$;    $step \leftarrow 2 \times step$.
13:      **end while**
14:      $upper \leftarrow \min\{1, lower + step\}$.
15: **end if**
16: Perform bisection on $[lower, upper]$ until $upper - lower < \varepsilon^2/2$, maintaining Query$(i, \mathbf{x}_{k,k',lower}) = $ False and Query$(i, \mathbf{x}_{k,k',upper}) = $ True.
17: Apply the same rational reconstruction step as ExactThreshold (Appendix C.1) to recover $\alpha^*_{i;k,k'}$ uniquely from $[lower, upper]$, and return it.

---

Then, we prove the following lemma.

**Lemma E.5.** *Fix $i, k, k'$ as in Algorithm 4, and let $\alpha^* := \alpha^*_{i;k,k'}$. For any warm start $\widehat{\alpha} \in [0, 1]$, ExactThresholdPred returns $\alpha^*$ exactly. Moreover, it has a query complexity of*

$$\mathcal{O}\left(1 + \log\left(1 + \frac{|\widehat{\alpha} - \alpha^*|}{\varepsilon^2}\right)\right).$$

*Proof.* Correctness follows from monotonicity of $\mathrm{Query}(i, \mathbf{x}_{k,k',\alpha})$ on $\alpha$ and the same denominator/uniqueness argument used for ExactThreshold in Appendix C.1: $\alpha^*$ is a rational with reduced denominator at most $1/\varepsilon$, so any bracket of width $< \varepsilon^2/2$ contains *at most one* such rational, and rational reconstruction returns $\alpha^*$ uniquely.

It remains to show that the algorithm maintains a valid bracket and to bound its length and query complexity. Let $d := |\widehat{\alpha} - \alpha^*|$.

If $b = \texttt{True}$, then $\widehat{\alpha} \geq \alpha^*$ and the algorithm decreases *upper* in steps of size *step* that double geometrically until either (i) the next candidate point falls below 0 or (ii) the query at $upper - step$ flips to $\texttt{False}$. In either case, by construction $\mathrm{Query}(i, \mathbf{x}_{k,k',upper}) = \texttt{True}$ and $\mathrm{Query}(i, \mathbf{x}_{k,k',lower}) = \texttt{False}$, and monotonicity implies $\alpha^* \in (lower, upper]$. The case $b = \texttt{False}$ is symmetric (searching to the right), giving us a bracket $\mathrm{Query}(i, \mathbf{x}_{k,k',lower}) = \texttt{False}$, $\mathrm{Query}(i, \mathbf{x}_{k,k',upper}) = \texttt{True}$, and $\alpha^* \in (lower, upper]$.

Next, we derive the bracket width after the exponential phase. Consider the case $b = \texttt{True}$ (the other case is identical). Let $step_0 := \varepsilon^2/2$ be the initial step size and suppose the **while** loop performs $t \geq 0$ successful iterations (i.e., the queried point remains accepting). Then the step size at the end of the exponential phase is $step_t = 2^t \times step_0$, and

$$\widehat{\alpha} - upper = step_0(1 + 2 + \cdots + 2^{t-1}) = (2^t - 1) \times step_0.$$

Since *upper* remains accepting throughout, we have $upper \geq \alpha^*$, and thus $d = \widehat{\alpha} - \alpha^* \geq \widehat{\alpha} - upper = (2^t - 1) \times step_0$. Rearranging gives $2^t \times step_0 \leq d + step_0$, i.e.,

$$step_t \leq d + \varepsilon^2/2.$$

At termination of the exponential phase, the algorithm sets $lower = \max\{0, upper - step_t\}$, so the resulting bracket has width at most $step_t$ (and at most *upper* if clipped at 0). Thus, the bracket width after the exponential phase is at most $d + \varepsilon^2/2$.

Now, we prove the query complexity. The exponential phase uses one query to evaluate $b$ plus one query per loop test, and the loop doubles *step* each successful iteration until *step* reaches the scale of $d$ (or until clipped at an endpoint). Thus the number of queries in the exponential phase is

$$\mathcal{O}\left(1 + \log\left(1 + \frac{d}{\varepsilon^2}\right)\right).$$

After this phase, the bracket width is at most $d + \varepsilon^2/2$. The bisection phase reduces the width to $< \varepsilon^2/2$, requiring

$$\mathcal{O}\left(\log\left(\frac{d + \varepsilon^2/2}{\varepsilon^2/2}\right)\right) = \mathcal{O}\left(\log\left(1 + \frac{d}{\varepsilon^2}\right)\right)$$

additional membership queries. Rational reconstruction uses no membership queries. Summing the phases proves the lemma. $\square$

Now, fix an agent $i \in N$ and a prediction $\hat{\mathbf{x}} \in \Delta(S)$.

Note that $\mathrm{LearnHyperplane}[\hat{\mathbf{x}}](i)$ is identical to $\mathrm{LearnHyperplane}(i)$ except that every invocation of $\mathrm{ExactThreshold}(i, k, k')$ is replaced by $\mathrm{ExactThresholdPred}(i, k, k', \widehat{\alpha}_{k,k'}(\hat{\mathbf{x}}))$, where $\widehat{\alpha}_{k,k'}(\hat{\mathbf{x}})$ is the pairwise projection coordinate defined in Section 5.2. All other steps (including the construction of $\mathbf{c}_i$ from the recovered turning points) are unchanged.

We first show **correctness**. $\mathrm{LearnHyperplane}(i)$'s correctness (Lemma 3.2) relies only on the fact that each threshold routine returns the exact turning point $\alpha^*_{i;k,k'}$ on the queried edge. By Lemma E.5, ExactThresholdPred returns the same turning point as ExactThreshold for every queried edge. Therefore $\mathrm{LearnHyperplane}[\hat{\mathbf{x}}](i)$, computes the same

values $\alpha^*_{i;k,k'}$ and returns exactly the same output (ACCEPTALL, REJECTALL, or an equivalent halfspace vector $\mathbf{c}_i$) as LearnHyperplane($i$).

Next, we prove the **query complexity**. Moreover, as in LearnHyperplane, LearnHyperplane[$\hat{\mathbf{x}}$] first queries all $m$ pure lotteries $\mathbf{e}_1, \ldots, \mathbf{e}_m$ once. If it returns ACCEPTALL or REJECTALL at this stage, the query complexity is exactly $m$. Otherwise, it invokes the threshold routine on at most $|\mathcal{E}_i| \leq m - 1$ simplex edges. For each $(k, k') \in \mathcal{E}_i$, the warm start used is $\widehat{\alpha}_{k,k'}(\hat{\mathbf{x}})$, so the distance to the true turning point is precisely $\delta_{i;k,k'}(\hat{\mathbf{x}})$. Lemma E.5 therefore implies that the number of queries used on edge $(k, k')$ is $\mathcal{O}\left(1 + \log\left(1 + \delta_{i;k,k'}(\hat{\mathbf{x}})/\varepsilon^2\right)\right)$. Summing over $(k, k') \in \mathcal{E}_i$ and adding the $m$ pure-lottery queries gives us

$$\mathcal{O}\left(m + \sum_{(k,k')\in\mathcal{E}_i} \log\left(1 + \frac{\delta_{i;k,k'}(\hat{\mathbf{x}})}{\varepsilon^2}\right)\right).$$

Finally, since $|\mathcal{E}_i| \leq m - 1$ and $\delta_{i;k,k'}(\hat{\mathbf{x}}) \leq \delta_i^{\max}(\hat{\mathbf{x}})$ for every $(k, k') \in \mathcal{E}_i$, we obtain the simplified bound

$$\mathcal{O}\left(m + m\log\left(1 + \frac{\delta_i^{\max}(\hat{\mathbf{x}})}{\varepsilon^2}\right)\right),$$

completing the proof.

### E.5. Lottery Predictions: Why violator count (of the lottery prediction) is not a useful error model

A tempting way to quantify the quality of a predicted lottery $\hat{\mathbf{x}} \in \Delta(S)$ is by its *violator count*

$$|V(\hat{\mathbf{x}})| \quad \text{where} \quad V(\hat{\mathbf{x}}) := \{i \in N : \text{Query}(i, \hat{\mathbf{x}}) = \texttt{False}\}.$$

While this quantity is relevant for the trivial "verify and stop" use of the lottery advice, it is generally a weak error model for learning-augmented elicitation because it evaluates advice at a *single* point and discards the geometric information that determines how many queries are required to learn constraints.

In particular, $|V(\hat{\mathbf{x}})|$ does not distinguish between (i) a prediction that lies deep inside many agents' acceptable regions and (ii) a prediction that is accepted by many agents but sits arbitrarily close to their acceptance boundaries. In case (ii), even if $|V(\hat{\mathbf{x}})|$ is small, a subsequent candidate lottery produced by an elicitation algorithm (e.g., Algorithm 2 or Algorithm 3) after learning a new constraint and recomputing $\mathbf{x} \leftarrow \text{Select}(C)$) can move by a small amount that flips many previously accepting agents into rejecting. As a result, the total number of encountered violators (and hence the number of costly hyperplane learning steps) can still be large despite a small violator count at $\hat{\mathbf{x}}$.

In contrast, the dominant cost inside LearnHyperplane is a collection of one-dimensional searches along simplex edges $\mathbf{x}_{k,k',\alpha}$, whose difficulty is governed by the corresponding turning points $\alpha^*_{i;k,k'}$. Therefore, the algorithmically relevant prediction quality is inherently *local to the edge*: the only useful information that a full lottery $\hat{\mathbf{x}}$ provides for queries restricted to the edge (the convex hull of $\mathbf{e}_k$ and $\mathbf{e}_{k'}$: $\text{conv}\{\mathbf{e}_k, \mathbf{e}_{k'}\}$) is its pairwise projection coordinate $\widehat{\alpha}_{k,k'}(\hat{\mathbf{x}})$, and the right notion of prediction error is the turning point projection error

$$\delta_{i;k,k'}(\hat{\mathbf{x}}) := \left|\widehat{\alpha}_{k,k'}(\hat{\mathbf{x}}) - \alpha^*_{i;k,k'}\right|.$$

This quantity directly controls how quickly a warm-started threshold search can bracket $\alpha^*_{i;k,k'}$ before the final bisection/reconstruction phase, giving us the smooth bounds in Theorem 5.3.

### E.6. Proof of Proposition 5.4

Let $Q := 1/\varepsilon \in \mathbb{Z}_{>0}$. If $Q < 4$, then $\delta/\varepsilon^2 \leq 1/\varepsilon^2 = Q^2 \leq 9$, so $\log(1 + \delta/\varepsilon^2) = \mathcal{O}(1)$. On the other hand, on any feasible instance every correct algorithm must query each agent at least once (the proof of Lemma D.1 applies with the advice $\hat{\mathbf{x}}$ held fixed), so it makes at least $n = 2$ queries. Thus the claimed $\Omega(\log(1 + \delta/\varepsilon^2))$ lower bound holds after adjusting constants. Henceforth, we assume $Q \geq 4$.

Define $M := \min\{\lfloor Q/2 \rfloor, \lfloor \delta Q^2 \rfloor + 1\}$. Since $\delta \geq \varepsilon^2$, we have $\delta Q^2 \geq 1$, hence $\lfloor \delta Q^2 \rfloor + 1 \geq 2$; also $\lfloor Q/2 \rfloor \geq 2$ because $Q \geq 4$. Therefore $M \geq 2$.

For each $t \in \{0, 1, \ldots, M - 1\}$, set

$$q_t := Q - t \quad \text{and} \quad \alpha_t := \frac{1}{q_t} \in (0, 1).$$

Note $1 \le q_t \le Q$, so $q_t \varepsilon \in \{0, \varepsilon, 2\varepsilon, \dots, 1\}$ and $\alpha_t$ is a rational with reduced denominator $q_t \le Q$.

Define the prediction coordinate as the midpoint

$$\hat{\alpha} := \frac{1}{2}(\alpha_0 + \alpha_{M-1}) \quad \text{and} \quad \hat{\mathbf{x}} := \mathbf{x}_{1,2,\hat{\alpha}} \in \Delta(S).$$

We claim that for every $t \in \{0, \dots, M-1\}$,

$$|\alpha_t - \hat{\alpha}| \le \delta. \tag{10}$$

Indeed, $\alpha_t$ is increasing in $t$, so

$$|\alpha_t - \hat{\alpha}| \le \frac{\alpha_{M-1} - \alpha_0}{2} = \frac{1}{2}\left(\frac{1}{Q - (M-1)} - \frac{1}{Q}\right) = \frac{M-1}{2Q(Q - (M-1))}.$$

Because $M \le \lfloor Q/2 \rfloor$, we have $Q - (M-1) \ge Q/2$, thus

$$|\alpha_t - \hat{\alpha}| \le \frac{M-1}{Q^2}.$$

Finally, since $M - 1 \le \lfloor \delta Q^2 \rfloor \le \delta Q^2$ (because $M - 1 = \min\{\lfloor Q/2 \rfloor - 1, \lfloor \delta Q^2 \rfloor\}$), we obtain $|\alpha_t - \hat{\alpha}| \le \delta$, proving (10).

For each $t \in \{0, \dots, M-1\}$, define an instance $\mathcal{I}_t$ with $n = 2$ agents and $m = 2$ alternatives $S = \{s_1, s_2\}$ as follows:

- Agent 2 has $u_2(s_1) = 0$, $u_2(s_2) = q_t \varepsilon$, and $\tau_2 = \varepsilon$.

- Agent 1 has $u_1(s_1) = q_t \varepsilon$, $u_1(s_2) = 0$, and $\tau_1 = (q_t - 1)\varepsilon$.

All utilities and thresholds are integer multiples of $\varepsilon$ in $[0, 1]$, and $\tau_1 > 0$ since $q_t \ge Q - (M-1) \ge Q/2 \ge 2$.

For any $\alpha \in [0, 1]$, write $\mathbf{x}_{1,2,\alpha} = (1 - \alpha)\mathbf{e}_1 + \alpha \mathbf{e}_2$. Then

$$U_2(\mathbf{x}_{1,2,\alpha}) = \alpha \cdot q_t \varepsilon, \quad \text{so} \quad U_2(\mathbf{x}_{1,2,\alpha}) \ge \tau_2 \iff \alpha \ge \frac{1}{q_t} = \alpha_t,$$

and

$$U_1(\mathbf{x}_{1,2,\alpha}) = (1 - \alpha) \cdot q_t \varepsilon, \quad \text{so} \quad U_1(\mathbf{x}_{1,2,\alpha}) \ge \tau_1 \iff \alpha \le \frac{1}{q_t} = \alpha_t.$$

Thus, the unanimously acceptable set in $\mathcal{I}_t$ is exactly the singleton $\{\mathbf{x}_{1,2,\alpha_t}\}$; denote this unique feasible lottery by $\mathbf{x}_t^*$.

Moreover, in each $\mathcal{I}_t$ we have $\mathrm{Query}(2, \mathbf{e}_1) = \texttt{False}$ and $\mathrm{Query}(2, \mathbf{e}_2) = \texttt{True}$, so the turning point for agent 2 on the edge $(\mathbf{e}_1, \mathbf{e}_2)$ is $\alpha_{2;1,2}^* = \alpha_t$ (by the definition in Section 5.2). Since $m = 2$, the pairwise projection coordinate satisfies $\alpha_{b1,2}(\hat{\mathbf{x}}) = \hat{x}_2 = \hat{\alpha}$, and thus,

$$\delta_{2;1,2}(\hat{\mathbf{x}}) = |\alpha_{b1,2}(\hat{\mathbf{x}}) - \alpha_{2;1,2}^*| = |\hat{\alpha} - \alpha_t| \le \delta,$$

where $\alpha_{2;1,2}^* = \alpha_t$ in instance $\mathcal{I}_t$.

Now, fix any correct deterministic decision tree $T$, and let $Q_T(\mathcal{I})$ be the number of queries made by $T$ on instance $\mathcal{I}$ (with the fixed advice $\hat{\mathbf{x}}$). Each $\mathcal{I}_t$ is feasible and has the unique correct output $\mathbf{x}_t^* = \mathbf{x}_{1,2,\alpha_t}$, and these outputs are distinct over $t$. Therefore the instances $\mathcal{I}_0, \dots, \mathcal{I}_{M-1}$ reach $M$ distinct leaves of $T$. Let $\mu$ be the uniform distribution over $\{\mathcal{I}_0, \dots, \mathcal{I}_{M-1}\}$; since the mapping $\mathcal{I}_t \mapsto$ reached leaf is injective, the induced distribution over these $M$ leaves is uniform. By the average-depth bound (Lemma D.4),

$$\mathbb{E}_{I \sim \mu}[Q_T(\mathcal{I})] \ge \log M. \tag{11}$$

In particular, $\max_t Q_T(\mathcal{I}_t) \ge \log M$.

Now consider randomized algorithms. Apply Lemma D.3 to the distribution $\mu$ above. Since (11) shows that $\mathbb{E}_{\mathcal{I} \sim \mu}[Q_T(\mathcal{I})] \ge \log M$ for every correct deterministic decision tree $T$, Lemma D.3 gives us

$$\inf_{\mathcal{A}} \sup_{\mathcal{I}} Q_{\mathcal{A}}(\mathcal{I}) \ge \log M,$$

where the infimum ranges over correct randomized algorithms. In particular, every correct randomized algorithm $\mathcal{A}$ satisfies $\sup_{\mathcal{I}} Q_{\mathcal{A}}(\mathcal{I}) \geq \log M$.

Let $\lambda := \delta Q^2 = \delta/\varepsilon^2 \geq 1$. Recall $M = \min\{\lfloor Q/2 \rfloor, \lfloor \lambda \rfloor + 1\}$. If $M = \lfloor \lambda \rfloor + 1$, then $M \geq 2$ and $M \geq \lambda$, so $M \geq \max\{2, \lambda\}$. For $\lambda \in [1, 2)$ we have $\log M \geq \log 2 \geq (\log 2 / \log 3) \log(1 + \lambda)$, while for $\lambda \geq 2$,

$$\log M \geq \log \lambda \geq \tfrac{1}{2} \log(1 + \lambda),$$

since $\log(1 + \lambda) \leq \log(2\lambda) = \log 2 + \log \lambda \leq 2 \log \lambda$. Thus in this case $\log M = \Omega(\log(1 + \lambda))$.

If $M = \lfloor Q/2 \rfloor$, then $M \geq Q/4$ (since $Q \geq 4$), so $\log M = \Omega(\log Q)$. Moreover $\lambda \leq Q^2$ (as $\delta \leq 1$) implies $\log(1 + \lambda) \leq \log(1 + Q^2) = O(\log Q)$, and hence again $\log M = \Omega(\log(1 + \lambda))$.

Therefore $\log M = \Omega(\log(1 + \lambda)) = \Omega(\log(1 + \delta/\varepsilon^2))$.

