_{\text{rand}}})$ returns NULL (Line 18), meaning $P(\text{supp}(N_{\text{rand}})) = \varnothing$. Since $P(N) \subseteq P(\text{supp}(N_{\text{rand}}))$, emptiness of $P(\text{supp}(N_{\text{rand}}))$ implies $P(N) = \varnothing$ (note that $P(N') = \cap_{i \in N'} \mathcal{A}_i$ with $\mathcal{A}_i \subseteq \Delta(S)$), i.e., no unanimously acceptable lottery exists. Hence returning NULL is correct.

Next, we will prove the **expected query complexity** using a series of intermediate lemmas.

We first make use of a lemma from Matoušek et al. (1996, Lemma 1(iii)), which highlights a standard property of $d$-dimensional linear programming: under lexicographic (deterministic) selection, every feasible bounded instance has a basis of size at most $d$. Since our feasible region lies in the affine hull of dimension $d = m - 1$, the claim follows (note that $P(N') \subseteq \Delta(S)$ is always bounded).

**Lemma C.1.** *For any subset of agents $N' \subseteq N$, if $P(N') \neq \varnothing$, then $N'$ has a basis $B \subseteq N'$ with $|B| \leq m - 1$.*

This lemma essentially tells us that there exists a "small" set of agents whose constraints determine the final chosen lottery $\mathbf{x}$. This "smallness" is what gives us the $\mathcal{O}(m \log n)$ dependence later.

Next, we prove the following weighted sampling lemma. Intuitively, in each iteration, Algorithm 3 sample $r'$ elements from the multiset $N_{\mathrm{multi}}(w)$ (given a weight vector $w = (w_1, \ldots, w_n)$) and solves the LP for the sampled distinct agents $\mathrm{supp}(N_{\mathrm{rand}})$ constraints (if an agent $i$ has not been "learned" yet, then run LearnHyperplane($i$) and store $\mathbf{c}_i$), getting a lottery $\mathbf{x}_{\mathrm{supp}(N_{\mathrm{rand}})}$, which is the unique lexicographically maximum feasible lottery for the sampled constraints (i.e., the output of Select on the sampled constraint set). The algorithm then verifies this against every agent by querying every agent on this candidate $\mathbf{x}_{\mathrm{supp}(N_{\mathrm{rand}})}$. Let $V$ be the agents who reject $\mathbf{x}_{\mathrm{supp}(N_{\mathrm{rand}})}$. This lemma gives us an upper bound on the expected *total weight of the violators* $w(V)$ after solving the sampled "subproblem" in each iteration. In particular, we will show that the expected total weight of violators for any iteration is $\mathbb{E}[w(V)] \leq m \cdot \frac{W}{r'+1}$.

Now, we can think of these weights as a sampling distribution (i.e., agent $i$ is more likely to be included in the next sampled subproblem/iteration when $w_i^{(t)}$ is large), and as a penalty counter (if agent $i$ rejects our proposed $\mathbf{x}$, we double $w_i^{(t)}$ for the next iteration). Thus, if the expected total weight of violators is small, then doubling of violator's weight would flesh them out faster, and provide us some progress towards identifying "more constraining" agents/likely violators faster.

**Lemma C.2.** *Let $V := \varnothing$ if $\mathbf{x}_{\mathrm{supp}(N_{\mathrm{rand}})} = \mathrm{NULL}$; and $V := \{i \in N : \mathbf{x}_{\mathrm{supp}(N_{\mathrm{rand}})} \notin \mathcal{A}_i\}$ otherwise. Then*

$$\mathbb{E}[w(V)] \leq m \cdot \frac{W}{r'+1}.$$

*In particular, if $r' = \min\{16(m-1)^2, W\}$, then if $W \leq r$, then $r' = W$ and the sample contains all agents, so $w(V) = 0$. If $W > r$, then*

$$\mathbb{E}[w(V)] \leq \frac{W}{8(m-1)}.$$

*Proof.* Each element $i \in N_{\mathrm{multi}}(w)$ is a copy of some agent $\widehat{i} \in N$.

For any $N_{\mathrm{rand}} \subseteq N_{\mathrm{multi}}(w)$, define $\mathbf{x}_{N_{\mathrm{rand}}} := \mathbf{x}_{\mathrm{supp}(N_{\mathrm{rand}})}$ and $\pi : N_{\mathrm{multi}}(w) \to N$. Then, define the *violator copy set*

$$V_{\mathrm{copy}}(N_{\mathrm{rand}}) := \{i \in N_{\mathrm{multi}}(w) \setminus N_{\mathrm{rand}} : \mathbf{x}_{N_{\mathrm{rand}}} \neq \mathrm{NULL} \text{ and } \mathbf{x}_{N_{\mathrm{rand}}} \notin \mathcal{A}_{\pi(i)}\}.$$

Intuitively, agents that have been sampled would have had their constraints learned and thus would not be a violator. Thus, it makes sense that violators only make up the agents that have not been learned (and in particular not sampled in this iteration).

If $\mathbf{x}_{N_{\mathrm{rand}}} = \mathrm{NULL}$ then $V_{\mathrm{copy}}(N_{\mathrm{rand}}) = \varnothing$ by definition. If $\mathbf{x}_{N_{\mathrm{rand}}} \neq \mathrm{NULL}$ and an agent $i$ violates $\mathbf{x}_{N_{\mathrm{rand}}}$, then no copy of $i$ can be in $N_{\mathrm{rand}}$ (otherwise $i \in \mathrm{supp}(N_{\mathrm{rand}})$ and $\mathbf{x}_{N_{\mathrm{rand}}} \in \mathcal{A}_i$). Thus, all $w_i$ copies of $i$ is contained in $N_{\mathrm{multi}}(w) \setminus N_{\mathrm{rand}}$ and also $V_{\mathrm{copy}}(N_{\mathrm{rand}})$. This means

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

$$|\alpha_t - \hat{\alpha}| \leq \frac{M-1}{Q^2}.$$

Finally, since $M - 1 \leq \lfloor \delta Q^2 \rfloor \leq \delta Q^2$ (because $M - 1 = \min\{\lfloor Q/2 \rfloor - 1, \lfloor \delta Q^2 \rfloor\}$), we obtain $|\alpha_t - \hat{\alpha}| \leq \delta$, proving (10).

For each $t \in \{0, \ldots, M-1\}$, define an instance $\mathcal{I}_t$ with $n = 2$ agents and $m = 2$ alternatives $S = \{s_1, s_2\}$ as follows:

- Agent 2 has $u_2(s_1) = 0$, $u_2(s_2) = q_t \varepsilon$, and $\tau_2 = \varepsilon$.

- Agent 1 has $u_1(s_1) = q_t \varepsilon$, $u_1(s_2) = 0$, and $\tau_1 = (q_t - 1)\varepsilon$.

All utilities and thresholds are integer multiples of $\varepsilon$ in $[0, 1]$, and $\tau_1 > 0$ since $q_t \geq Q - (M - 1) \geq Q/2 \geq 2$.

For any $\alpha \in [0, 1]$, write $\mathbf{x}_{1,2,\alpha} = (1 - \alpha)\mathbf{e}_1 + \alpha\mathbf{e}_2$. Then

$$U_2(\mathbf{x}_{1,2,\alpha}) = \alpha \cdot q_t\varepsilon, \quad \text{so} \quad U_2(\mathbf{x}_{1,2,\alpha}) \geq \tau_2 \iff \alpha \geq \frac{1}{q_t} = \alpha_t,$$

and

$$U_1(\mathbf{x}_{1,2,\alpha}) = (1 - \alpha) \cdot q_t\varepsilon, \quad \text{so} \quad U_1(\mathbf{x}_{1,2,\alpha}) \geq \tau_1 \iff \alpha \leq \frac{1}{q_t} = \alpha_t.$$

Thus, the unanimously acceptable set in $\mathcal{I}_t$ is exactly the singleton $\{\mathbf{x}_{1,2,\alpha_t}\}$; denote this unique feasible lottery by $\mathbf{x}_t^*$.

Moreover, in each $\mathcal{I}_t$ we have $\text{Query}(2, \mathbf{e}_1) = \text{False}$ and $\text{Query}(2, \mathbf{e}_2) = \text{True}$, so the turning point for agent 2 on the edge $(\mathbf{e}_1, \mathbf{e}_2)$ is $\alpha_{2;1,2}^* = \alpha_t$ (by the definition in Section 5.2). Since $m = 2$, the pairwise projection coordinate satisfies $\alpha_{b1,2}(\widehat{\mathbf{x}}) = \widehat{x}_2 = \widehat{\alpha}$, and thus,

$$\delta_{2;1,2}(\widehat{\mathbf{x}}) = |\alpha_{b1,2}(\widehat{\mathbf{x}}) - \alpha_{2;1,2}^*| = |\widehat{\alpha} - \alpha_t| \leq \delta,$$

where $\alpha_{2;1,2}^* = \alpha_t$ in instance $\mathcal{I}_t$.

Now, fix any correct deterministic decision tree $T$, and let $Q_T(\mathcal{I})$ be the number of queries made by $T$ on instance $\mathcal{I}$ (with the fixed advice $\hat{x}$). Each $\mathcal{I}_t$ is feasible and has the unique correct output $\mathbf{x}_t^* = \mathbf{x}_{1,2,\alpha_t}$, and these outputs are distinct over $t$. Therefore the instances $\mathcal{I}_0, \ldots, \mathcal{I}_{M-1}$ reach $M$ distinct leaves of $T$. Let $\mu$ be the uniform distribution over $\{\mathcal{I}_0, \ldots, \mathcal{I}_{M-1}\}$; since the mapping $\mathcal{I}_t \mapsto$ reached leaf is injective, the induced distribution over these $M$ leaves is uniform. By the average-depth bound (Lemma D.4),

$$\mathbb{E}_{I \sim \mu}\big[Q_T(\mathcal{I})\big] \geq \log M. \tag{11}$$

In particular, $\max_t Q_T(\mathcal{I}_t) \geq \log M$.

Now consider randomized algorithms. Apply Lemma D.3 to the distribution $\mu$ above. Since (11) shows that $\mathbb{E}_{\mathcal{I} \sim \mu}[Q_T(\mathcal{I})] \geq \log M$ for every correct deterministic decision tree $T$, Lemma D.3 gives us

$$\inf_{\mathcal{A}} \sup_{\mathcal{I}} Q_{\mathcal{A}}(\mathcal{I}) \geq \log M,$$

where the infimum ranges over correct randomized algorithms. In particular, every correct randomized algorithm $\mathcal{A}$ satisfies $\sup_{\mathcal{I}} Q_{\mathcal{A}}(\mathcal{I}) \geq \log M$.

Let $\lambda := \delta Q^2 = \delta/\varepsilon^2 \geq 1$. Recall $M = \min\{\lfloor Q/2 \rfloor, \lfloor \lambda \rfloor + 1\}$. If $M = \lfloor \lambda \rfloor + 1$, then $M \geq 2$ and $M \geq \lambda$, so $M \geq \max\{2, \lambda\}$. For $\lambda \in [1, 2)$ we have $\log M \geq \log 2 \geq (\log 2/\log 3) \log(1 + \lambda)$, while for $\lambda \geq 2$,

$$\log M \geq \log \lambda \geq \tfrac{1}{2} \log(1 + \lambda),$$

since $\log(1 + \lambda) \leq \log(2\lambda) = \log 2 + \log \lambda \leq 2 \log \lambda$. Thus in this case $\log M = \Omega(\log(1 + \lambda))$.

If $M = \lfloor Q/2 \rfloor$, then $M \geq Q/4$ (since $Q \geq 4$), so $\log M = \Omega(\log Q)$. Moreover $\lambda \leq Q^2$ (as $\delta \leq 1$) implies $\log(1 + \lambda) \leq \log(1 + Q^2) = O(\log Q)$, and hence again $\log M = \Omega(\log(1 + \lambda))$.

Therefore $\log M = \Omega(\log(1 + \lambda)) = \Omega(\log(1 + \delta/\varepsilon^2))$.